# Premature birth changes wiring constraints in neonatal structural brain networks

Alexa Mousley [1] ✉, Danyal Akarca[1,2,3] & Duncan E. Astle [1,4]

Structural brain organization in infancy is associated with later cognitive, behavioral, and educational outcomes. Due to practical limitations, such as technological advancements and data availability of fetal MRI, there is still much we do not know about the early emergence of topological organization. We combine the developing Human Connectome Project's large infant dataset with generative network modeling to simulate the emergence of network organization over early development. Preterm infants had reduced connectivity, shorter connection lengths, and lower network efficiency compared to term-born infants. The models were able to recapitulate the organizational differences between term and preterm networks and revealed that preterm infant networks are better simulated under tighter wiring constraints than term infants. Tighter constraints for preterm models resulted in shorter connection lengths while preserving vital, long-range rich club connections. These simulations suggest that preterm birth is associated with a renegotiation of the cost-value wiring trade-off that may drive the emergence of different network organization.

The human brain has a characteristic structural organization that relates to cognition and behavior[1–7]. To explore neural connectivity, maps of brain networks, called connectomes, can be constructed from diffusion-weighted magnetic resonance imaging (dMRI) and used to explore the brain's organizational properties. dMRI captures the direction of water diffusion, enabling the location and direction of white matter fibers in the brain to be traced[8,9]. Connectomes can be built using fiber tracking methods, which results in a map of the brain with nodes representing neural elements (e.g., neuron or brain region) and edges depicting connections (e.g., axonal projections or white matter tracts)[10,11]. These simplified representations of neural networks enable the exploration of organization of individual brain networks[8,9].

Connectomics researchers often apply graph theory measurements to quantify and characterize the organization of the brain's complex networks[10,12–22]. A typical human brain is organized into modules – nodes that are more densely connected to each other than to other parts of the network[23–25]. Modules are connected by rich clubs[17,26,27], which are connections between hubs regions (i.e., regions

with a high number of total connections)[10,28]. In addition, network efficiency can be calculated by the length of paths connecting the regions, providing information about the cost-effectiveness of information transfer throughout the network[17,24]. Overall, the application of graph theory principles has facilitated the development of a common framework of whole-brain network properties that can be used to explore universal and individual differences in structural brain organization[10,11,17].

Many of the organizational hallmarks of human brains that are associated with cognitive outcomes in childhood are developed during the prenatal period. Adult-like hub distribution and rich clubs are present by the time a full-term infant is born[7,12,13,20,24,27,29–32]. Specifically, major rich club organization and connections are present by 30 weeks gestational age (GA) and small-worldness properties (i.e., highly clustered with short path lengths) are reported as early as 20 weeks GA[27,31]. The study of the early emergence of these fundamental organizational properties has practical challenges, such as limited in-utero imaging and comorbidities associated with premature birth.

[1]MRC Cognition and Brain Sciences Unit, University of Cambridge, Cambridge, UK. [2]Department of Electrical and Electronic Engineering, Imperial College London, London, UK. [3]Imperial-X, Imperial College London, London, UK. [4]Department of Psychiatry, University of Cambridge, Cambridge, UK.
✉e-mail: alexa.mousley@mrc-cbu.cam.ac.uk

Generative network modeling (GNM) is a computational framework that can provide insight into the principles of topological organization because these models achieve a compressed representation of the principle constraints upon brain networks[33–36]. The purpose of these models is not to recapitulate edge locations, instead binary GNMs aim to replicate topological patterns of real networks and thus allow for exploration of what conditions are required to yield these properties. The earliest GNMs were motivated by Kaiser and Hilgetag (2004) who demonstrated it was possible to emulate empirical connection length distributions of the animal connectomes via a simple spatial exponential penalty model[37]. Vértes et al. (2012) further explored multiple equations for incorporating spatial constraints and added topological value as a driving factor for forming connections beyond cost alone[38]. The best preforming equation, which accurately modeled topological organization in the human brain, multiplies the 'cost' (i.e., Euclidean distance between nodes) by a topological 'value' of making a connection[38]. If the cost outweighs the value of the connection, the probability that the connection will be made is low and vice versa. This equation has been explored with multiple different metrics for 'value', such as the difference between clustering coefficients of both nodes[39]. A homophily rule, which quantifies how similar both nodes are via proportion or number of shared neighbors, yields the most accurate simulation of structural and functional brain networks of humans, mice, and neural cells[22,39–41]. Homophilic GNMs have been validated in healthy adults and adults with schizophrenia as well as typical and neurodiverse children[22,34,39,42].

This study explores the development of neonatal structural network organization, and its relation to premature birth, by using computational models – namely GNMs – to simulate the formation of those networks. Specifically, (1) we explored observed topological organization across a diverse sample of term and preterm infants; (2) we optimized and fit accurate simulations to infant brain networks; and (3) we examined network simulations to develop insight into altered economic wiring constraints related to the timing of birth.

## Results

We explored the relationship between binarized structural brain networks and two age-related metrics. Postmenstrual age (PMA) refers to the time since the mother's last menstrual period (including both gestation and time since birth), and gestational age (GA) at birth refers to the gestational week of the infant when they were born. GA at birth defines their term: an infant is *preterm* if born less than 37 weeks gestation and *term* if born 37 weeks or later. For example, two infants scanned at 42 PMA have had the same amount of time to develop, however, one of these infants was born at 23 weeks GA (i.e., preterm), while the other was born at 41 weeks GA (i.e., term) (Fig. 1a). Therefore, while the two are developmentally comparable, one has been exposed to the extra-uterine environment for 19 weeks, while the other has only been in the extra-uterine environment for one week (Fig. 1a). For plots, each neonate will have two points: one depicting their GA at birth and the other depicting their PMA at the time of the scan. We considered both age metrics' effects on macroscopic neural connectivity by statistically controlling for PMA when exploring GA at birth and vice versa using generalized additive models (see "Methods"; "Statistics" for detail).

Our cross-sectional sample consisted of neonates ($N = 630$) scanned at 26 to 45 weeks PMA ($n = 183$ preterm, $n = 447$ term; see "Methods"; "Participants" for detail; Supplementary Fig. 1). The infants were scanned on average 1.84 weeks following birth ($SD = 2.38$), with a significant negative correlation between GA at birth and the number of weeks between birth and scan ($r_{pearson} = -0.53$, $p = 6.76 \times 10^{-48}$). As scanning was completed at a maximum from 44 PMA, infants born at term must be scanned within a short time frame, whereas infants born early have a longer age range in which scanning can be conducted.

## PMA and GA at birth relationships with the macroscopic neonatal connectome

Creating GNMs is only possible with sparse connectomes[22,38,39,41]. This requires that we threshold our connectomes (see "Methods"; "Connection construction"), so we first explored how these sparse networks change according to PMA and GA at birth. To disentangle topological structure from network density, we conducted two analyses: (1) a variable density analysis for which an absolute threshold was applied (325 streamlines) that yielded an average density of 10% across the sample, and (2) a density-controlled analysis for which each network thresholded individually so that every network in the sample was exactly 10% density. Finally, because the modeling requires sparse networks, we also explored the impact of that sparsity on topology across networks with 8–24% density (Supplementary Fig. 2).

Network density (i.e., the number of connections compared to the number of possible connections) increased across both PMA and GA at birth (Fig. 1c). A rapid increase in density was observed from about 26 to 35 PMA, followed by a decline which established a mild inverted U curve (Fig. 1c; $F_{density,PMA} = 30.23$, estimated df = 7.82, $p < 2.00 \times 10^{-16}$). A more linear relationship was observed across GA at birth, with density steadily increasing across preterm ages and flattening around the time of term birth (Fig. 1c; $F_{density,GA\ at\ birth} = 6.68$, estimated df = 3.13, $p = 3.56 \times 10^{-5}$). This result was replicated in a term-equivalent age analysis in which only infants scanned at or above 37 weeks PMA were included (Supplementary Fig. 3a; $F_{density,GA\ at\ birth} = 2.71$, estimated df = 3.02, $p = 0.026$). Together, these results are consistent with the expected rapid increase in connectivity across development, represented by PMA, as well as a relative reduction in connectivity being associated with premature birth.

The global organization of the binarized networks with variable density demonstrated a shallow U curve trend for network segregation (for definitions of network measures see "Methods"; "Graph theory"). Network modularity significantly decreased across PMA, reaching a minimum around 36 weeks followed by a minor increase (Fig. 2a; $F_{modularity,PMA} = 4.55$, estimated df = 3.99, $p = 4.10 \times 10^{-4}$). This finding is congruent with network density increases because as more connections are added to the network, fewer non-overlapping groups of nodes will be present. However, the density-controlled analysis revealed that modularity significantly increased across PMA, particularly rapidly before about 34 PMA (Fig. 2a; $F_{modularity,PMA} = 8.15$, estimated df = 5.35, $p < 2.00 \times 10^{-16}$). This suggests that while the overall increase in density is related to a less modular network, the *type* of connections that are forming at older PMAs are driving an increase in modular structure. In contrast, both variable and density-controlled analyses demonstrated that modularity significantly decreases across GA at birth (Fig. 2a; variable: $F_{modularity,GA\ at\ birth} = 7.96$, estimated df = 3.08, $p = 6.74 \times 10^{-6}$; controlled: $F_{modularity,GA\ at\ birth} = 8.93$, estimated df = 1.00, $p = 2.90 \times 10^{-4}$). This result was replicated in the term-equivalent age analysis (Supplementary Fig. 3b; $F_{modularity,GA\ at\ birth} = 9.42$, estimated df = 1.00, $p = 2.25 \times 10^{-3}$). In addition, preterm infants had significantly more modular networks compared to term infants ($t(216.96) = 12.16$, $p = 2.20 \times 10^{-16}$) in the propensity-matched analysis where the two groups are identically matched on sex and PMA (Supplementary Fig. 3b; see "Methods"; "Participants"). These findings suggest that in addition to less dense, more modular networks overall, infants born at early GA have a fundamentally more modular topology.

Network integration in variable density connectomes increased across PMA, forming an inverted U curve that peaks around 36 postmenstrual weeks. We observed a significant increase in global efficiency and decrease in characteristic path length across PMA (Fig. 2a; $F_{global\ efficiency,PMA} = 18.92$, estimated df = 5.92, $p < 2.00 \times 10^{-16}$; $F_{characteristic\ path\ length,PMA} = 17.94$, estimated df = 5.71, $p < 2.00 \times 10^{-16}$). However, when density is controlled, global efficiency significantly decreases while characteristic path length increases (Fig. 2a; $F_{global\ efficiency,PMA} = 10.99$, estimated df = 3.91, $p < 2.00 \times 10^{-16}$; $F_{characteristic\ path}$

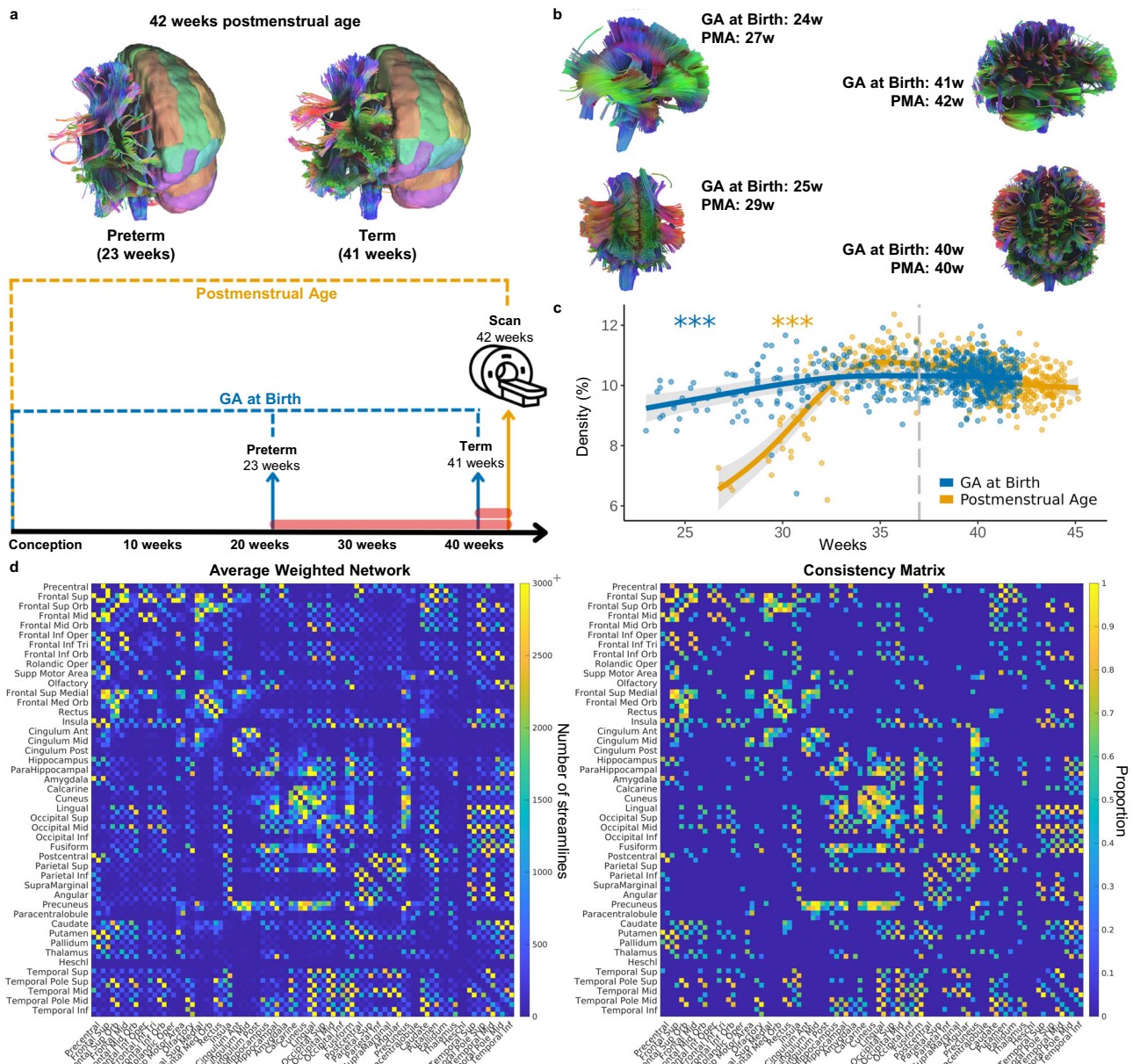

**Fig. 1 | Neonatal connectivity across PMA and GA at birth. a** Tractograms highlighting the reduced connectivity between preterm and term infants who were both scanned at 42 weeks PMA. The left hemispheres of tractograms depict region segmentation adapted from AAL90 atlas in Shi F, et al. (2011) Infant Brain Atlases from Neonates to 1- and 2-Year-Olds. PLoS ONE 6(4): e18746. doi:10.1371/journal.pone.0018746. Below the tractograms, a graphic depicts the difference between PMA (orange) and GA at birth (blue) for these infants. The red bars indicate the time infants were exposed to the *extra-uterine* environment before being scanned. **b** Representative tractograms of four neonates born and scanned at different PMA and GA at birth. **c** Connectome density significantly increased across PMA ($p < 2.00 \times 10^{-16}$) and GA at birth ($p = 3.56 \times 10^{-5}$) (represented as predicted density from generalized additive model). The grey dotted line indicates the cut-off for term birth (37 weeks GA or later is term-born). The shaded area indicates 95% confidence intervals. **d** Average connectivity weighted matrix (average number of streamlines) and binarized matrix (consistency matrix), where the value indicates the proportion of participants that have that edge. *** indicates $p < 0.001$, ** indicates $p < 0.01$, * indicates $p < 0.05$.

$length,PMA = 10.17$, estimated df = 3.66, $p < 2.00 \times 10^{-16}$). Together, these results suggest that increasing network efficiency across PMA can be explained by increases in density – when density is controlled for, the topological structure of networks becomes *less* efficient across PMA. Similar to PMA, with variable density networks global efficiency significantly increased while characteristic path length decreased across GA at birth (Fig. 2a; $F_{global\ efficiency,GA\ at\ birth} = 6.12$, estimated df = 3.66, $p = 3.82 \times 10^{-5}$; $F_{characteristic\ path\ length,GA\ at\ birth} = 5.69$, estimated df = 3.16, $p = 1.79 \times 10^{-4}$). This result was replicated in the term-equivalent age analysis (Supplementary Fig. 3c; $F_{global\ efficiency,GA\ at\ birth} = 2.71$, estimated df = 3.60, $p = 0.029$) and is consistent with the propensity-

matched analysis in which preterm infants had significantly lower global efficiency compared to term infants ($t(203.35) = -7.24$, $p = 9.19 \times 10^{-12}$). When controlling for density, however, no significant relationships were found between GA at birth and global efficiency (Fig. 2a; $p = 0.462$) or characteristic path length ($p = 0.918$). In other words, the less efficient networks for infants born at early GA can primarily be explained by less dense networks, not a fundamental change in underlying topology. Multiple local organization measures of variable density networks displayed significant changes across PMA and GA at birth after false discovery rate correction and are in the supplementary materials (Supplementary Fig. 4 & 5).

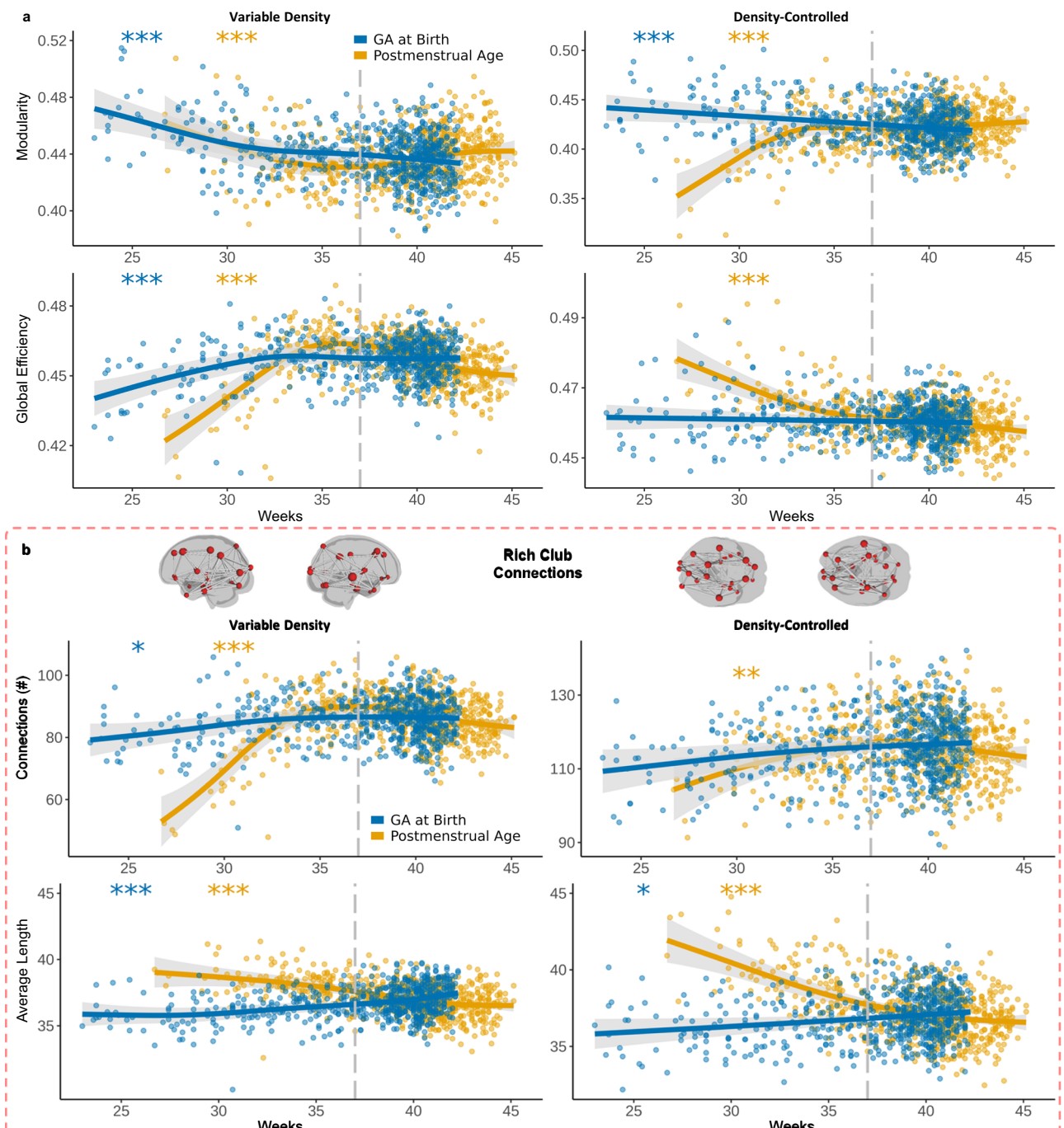

**Fig. 2 | Organizational differences across PMA and GA at birth. a** Modularity significantly decreased across PMA ($p = 4.10 \times 10^{-4}$) and GA at birth ($p = 6.74 \times 10^{-6}$) in variable density networks, but modularity significantly increased across PMA ($p < 2.00 \times 10^{-16}$) and decreased across GA at birth ($p = 2.90 \times 10^{-4}$) in density-controlled networks. Global efficiency significantly increased across both PMA ($p < 2.00 \times 10^{-16}$) and GA at birth ($p = 3.82 \times 10^{-5}$) in variable density networks. However, with density-controlled networks global efficiency significantly decreased across PMA ($p < 2.00 \times 10^{-16}$) but no significant effect was found across GA at birth ($p = 0.462$). **b** The number of rich club connections significantly increased across PMA in both variable density ($p < 2.00 \times 10^{-16}$) and density-controlled networks ($p = 2.31 \times 10^{-3}$). The number of rich club connections also

significantly increased across GA at birth in variable density networks ($p = 0.022$), but no significant effect was found with density-controlled networks ($p = 0.059$). In addition, the average length of rich club connections (Euclidean distance) significantly decreased across PMA in both variable density ($p < 2.00 \times 10^{-16}$) and density-controlled networks ($p < 2.00 \times 10^{-16}$). The average length of rich club connections significantly increased across GA at birth in both variable density ($p = 1.06 \times 10^{-4}$) and density-controlled networks ($p = 0.029$). All results presented are generalized additive models. The shaded area indicates 95% confidence intervals. The grey dotted line indicates the cut-off for term birth (37 weeks GA or later is term-born). *** indicates $p < 0.001$, ** indicates $p < 0.01$, * indicates $p < 0.05$.

We also analyzed rich club connections by defining 27 rich club nodes in the consensus network (Supplementary Fig. 6; see "Methods"; "Graph theory"). Across PMA and GA at birth, the total number of rich club connections significantly increased in variable density networks,

as would be expected given increasing density (Fig. 2b; $F_{rich\ connections, PMA} = 16.62$, estimated df = 5.92, $p < 2.00 \times 10^{-16}$; $F_{rich\ connections, GA\ at\ birth} = 3.05$, estimated df = 2.69, $p = 0.022$). For the term-equivalent age analysis, however, no significant relationship was found

between the number of rich club connections and GA at birth (Supplementary Fig. 3d; $p = 0.412$). Indeed, the density-controlled network analysis demonstrated that while the total number of rich club connections still significantly increased across PMA (Fig. 2b; $F_{rich\ connections,PMA} = 4.05$, estimated df = 3.42, $p = 2.31$ x $10^{-3}$), no significant effect was found across GA at birth (Fig. 2b; $p = 0.059$). Put simply, the increase in rich club connections across PMA occurred regardless of changes in density, but differences in rich club connections across GA at birth were density dependent.

We also explored the lengths of rich club connections. Across PMA, average length of rich club connections significantly decreased in both variable and density-controlled networks (Fig. 2b; variable: $F_{rich\ lengths,PMA} = 11.92$, estimated df = 3.04, $p < 2.00$ x $10^{-16}$; density-controlled: $F_{rich\ lengths,PMA} = 15.58$, estimated df = 3.12, $p < 2.00$ x $10^{-16}$). These results suggest that early PMA rich club formation consists of longer-range connections compared to later PMAs. In addition, the average length of rich club connections significantly increased across GA at birth in both variable and density-controlled networks (Fig. 2b; variable: $F_{rich\ lengths,GA\ at\ birth} = 5.37$, estimated df = 3.82, $p = 1.06$ x $10^{-4}$; density-controlled: $F_{rich\ lengths,GA\ at\ birth} = 3.45$, estimated df = 1.37, $p = 0.029$) as well as in the term-equivalent age analysis (Supplementary Fig. 3e; $F_{rich\ connection\ length,GA\ at\ birth} = 8.10$, estimated df = 2.81, $p = 2.34$ x $10^{-5}$). Contrasting that of PMA, these results indicate that early GA at birth is related to shorter rich club connections than infants born at later GA, unrelated to total connectivity differences. Plots of connection numbers and lengths for rich, feeder and local connections in variable density networks are in the supplementary materials (Supplementary Fig. 7).

To summarize, these findings suggest a shallow U curve trajectory for early brain development. During the early stages of neonatal development (approximately before 35 weeks PMA), structural brain networks rapidly undergo rapid formation of new connections. This process fosters increased integration and efficiency across PMA which masks the fundamental topological trends towards less integrated and efficient networks. Changes across GA at birth are more nuanced, rapid increases in density relate to increasing integration, though when density is controlled, we observed decreasing modularity but no differences in efficiency. Nevertheless, connectivity between long-range, highly connected regions (i.e., rich clubs) is relatively preserved across GA at birth and suggests that premature birth is related to a biased wiring selection towards shorter rich club connections. Overall, preterm networks exhibit reduced integration and efficiency while also displaying higher segregation compared to term infants.

## A simple generative network model can simulate diverse neonatal networks
Next, we tested whether or how organizational differences are shaped by economic wiring constraints, by simulating the formation of those networks. These simulations are produced by a simple model which iteratively adds connections between nodes (i.e., regions) which are in a known physical space (i.e., Euclidean distances between nodes). In these models, binary networks are generated by a probabilistic equation that determines if any pair of unconnected nodes ($i$ and $j$) will form a connection (Fig. 3):

$$p_{i,j} \propto (d_{i,j})^{\eta} (k_{i,j})^{\gamma} \qquad (1)$$

$p_{i,j}$ is the probability that nodes $i$ and $j$ will connect and is determined by a 'cost' measure ($d_{i,j}$) multiplied by a 'value' measure ($k_{i,j}$). $d_{i,j}$ is the Euclidean distance between nodes and is raised to the parameter $\eta$, which, when negative, favors the formation of short-range connections. Euclidean distance, which is highly correlated with fiber length[40], is used because a model requires a full distance matrix, including regions that have not been connected yet. Therefore, $d_{i,j}$ is predefined

by the Euclidean distances between regions based on the atlas (see "Methods"; "Generative network modeling") and is consistent across participants and generative timescale (i.e., the iterative addition of edges). $k_{i,j}$ is a 'value' measure that can be calculated by a variety of methods based on different measures of value such as homophily, clustering, or degree[22,39]. $k_{i,j}$ is raised to the parameter $\gamma$, which weighs the strength of the value measure. Because $k$ is based on topology, every participant starts with the same $k$ value which will diversify through the generative process depending on the status of the network's topology (Fig. 3a). We tested multiple 'value' measures and found that using a homophily measure produced the most accurate networks (Fig. 4a; see "Methods"; "Generative network modeling"; Supplementary Fig. 8), which is consistent with GNM work in animals, cell cultures, and humans[22,34,39,40,42]. After optimizing the GNMs for infant simulations, we produced simulations for each participant and found their best-fit model (Fig. 3b; see "Methods"; "Generative network modeling").

Optimized GNMs replicated multiple aspects of diverse neonatal network topology. Models captured the local arrangements of topological properties of observed networks, termed their 'topological fingerprint'[40] (see "Methods"; "Generative network modeling"). Topological fingerprints demonstrated that the simulations' local measures show similar relationships to each other as the observed networks' local measures (Fig. 4d; TF$_{dissimilarity}$: min = 0.17, max= 1.96, $M = 0.91$, $SD = 0.36$). Furthermore, at the global level simulations accurately recapitulated network integration and segregation measures (Fig. 4b, c). Global organization measures in observed networks were significantly correlated with simulated network measures for number of rich club connections ($r_{pearson} = 0.23$, $p = 6.98$ x $10^{-9}$), average length of rich club connections ($r_{pearson} = 0.20$, $p = 3.84$ x $10^{-7}$), characteristic path length ($r_{pearson} = 0.69$; $p = 3.76$ x $10^{-91}$), global efficiency ($r_{pearson} = 0.82$; $p = 6.58$ x $10^{-155}$), and modularity ($r_{pearson} = 0.31$; $p = 2.49$ x $10^{-15}$) (Fig. 4e).

In addition, GNMs also replicated spatially embedded organizational structure (i.e., local organization measures) of cortical regions. For individually fit GNMs, spatial embedding was assessed with linear mixed effects models (LMEM) with node and participant as random effects. This analysis showed that local organization of the simulated networks was significantly related to that of the observed networks for edge length (estimate = 3.24 x $10^{-2}$, $t(59.98) = 3.52$, $p = 8.42$ x $10^{-4}$), degree (Fig. 4f; estimate = 0.04, $t(78.46) = 4.30$, $p = 4.96$ x $10^{-5}$), local efficiency (estimate = 0.02, $t(60.56) = 3.17$, $p = 2.40$ x $10^{-3}$), clustering coefficient (estimate = 0.02, $t(53.37) = 2.85$, $p = 6.13$ x $10^{-3}$), and matching (estimate = 0.06, $t(89.26) = 4.61$, $p = 1.33$ x $10^{-5}$) but not betweenness centrality ($p = 0.500$) (Supplementary Fig. 9). It is important to note that while we see significant spatial embedding across the sample with LMEM, individual-level spatial embedding appears variable given correlational plots[43] (Fig. 4f; Supplementary Fig. 9).

## Simulations of neonatal networks capture changes in economic wiring conditions across PMA and GA at birth
While GNMs are well-fit given low energy, we saw differences in model fit across both PMA and GA at birth. Energy significantly decreased across PMA and GA at birth (Fig. 5a; $F_{energy,PMA} = 25.65$, estimated df = 8.97, $p < 2.00$ x $10^{-16}$; $F_{energy,GA\ at\ birth} = 7.19$, estimated df = 1.00, $p = 7.00$ x $10^{-3}$). While GA at birth and model fit were linearly related, with model fit improving across GA at birth ($r_{partial\ correlation} = -0.16$; $p = 6.60$ x $10^{-5}$), model fit rapidly improved from about 26 to 33 weeks PMA before plateauing (Fig. 5a). The density-controlled analysis indicates a significant improvement of model fit across PMA but no significant difference in model fit across GA at birth (Supplementary Fig. 10).

We also found that model parameters, which capture the models' economic wiring constraints, were significantly related to both PMA and GA at birth. The 'cost' parameter, $\eta$, significantly decreased across PMA but increased across GA at birth (Fig. 5b; $F_{\eta,PMA} = 31.10$, estimated df =

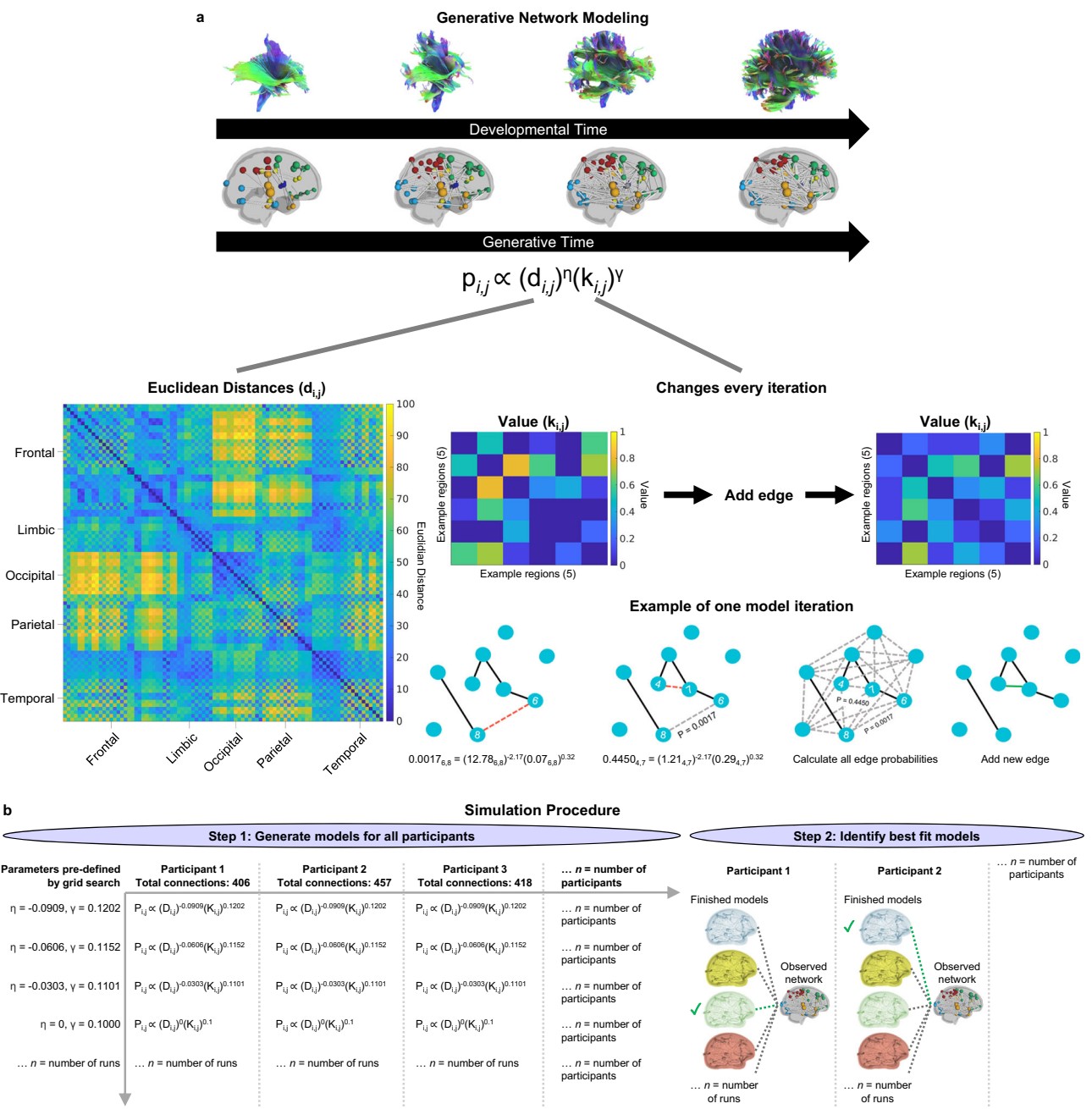

**Fig. 3 | Generative network modeling schematic. a** Diagram depicting how models form over time by adding connections in the network, based on a trade-off between parameterized cost ($d_{i,j}$) and value ($k_{i,j}$), to mimic network growth. Cost is a static metric defined by the Euclidean distances between nodes. Value is a topological metric that changes every time a new connection is added. The model is generated by iteratively by calculating the wiring probability of all potential connections and selecting a single connection based on the probability score. **b** The process of creating simulations starts by performing a grid search to identify the parameter combinations for each run, for example, 10,000 runs = 10,000 parameter combinations. Each participant's generative models were created with the same parameter combinations; however, the models grew to the size of that individual's network based on the total connections present in the participant's observed network. Then, for each participant, we compared their finished models to their observed network and identified the single best-fit model for that individual. Node plots in both panels are adapted from AAL90 atlas in Shi F, et al. (2011) Infant Brain Atlases from Neonates to 1- and 2-Year-Olds. PLoS ONE 6(4): e18746. doi:10.1371/ journal.pone.0018746.

df = 4.09, $p$ < 2.00 x $10^{-16}$; $F_{\eta, GA\ at\ birth}$ = 7.21, estimated df = 3.29, $p$ = 8.61 x $10^{-6}$). This suggests that across PMA, the strength of selection for short-range connection increases. However, early GA at birth is related to a significant acceleration of this effect, resulting in weaker penalization of long-range connections for infants born early. On the other hand, the 'value' parameter, $\gamma$, significantly increased across PMA but decreased across GA at birth (Fig. 5b; $F_{\gamma, PMA}$ = 14.80, estimated df = 5.84, $p$ < 2.00 x $10^{-16}$; $F_{\gamma, GA\ at\ birth}$ = 3.45, estimated df = 2.01, $p$ = 0.018).

The importance of topological information for wiring likely increases across PMA. GA at birth appears to be associated with an acceleration in this process, demonstrated by a higher reliance on topological value for infants born early than at term. Together, these results indicate early GA at birth is associated with tighter constraints (i.e., stronger model parameters for both wiring cost and value). These results are confirmed through density-controlled, term-equivalent age, and propensity-matched analyses. However, in the propensity-matched

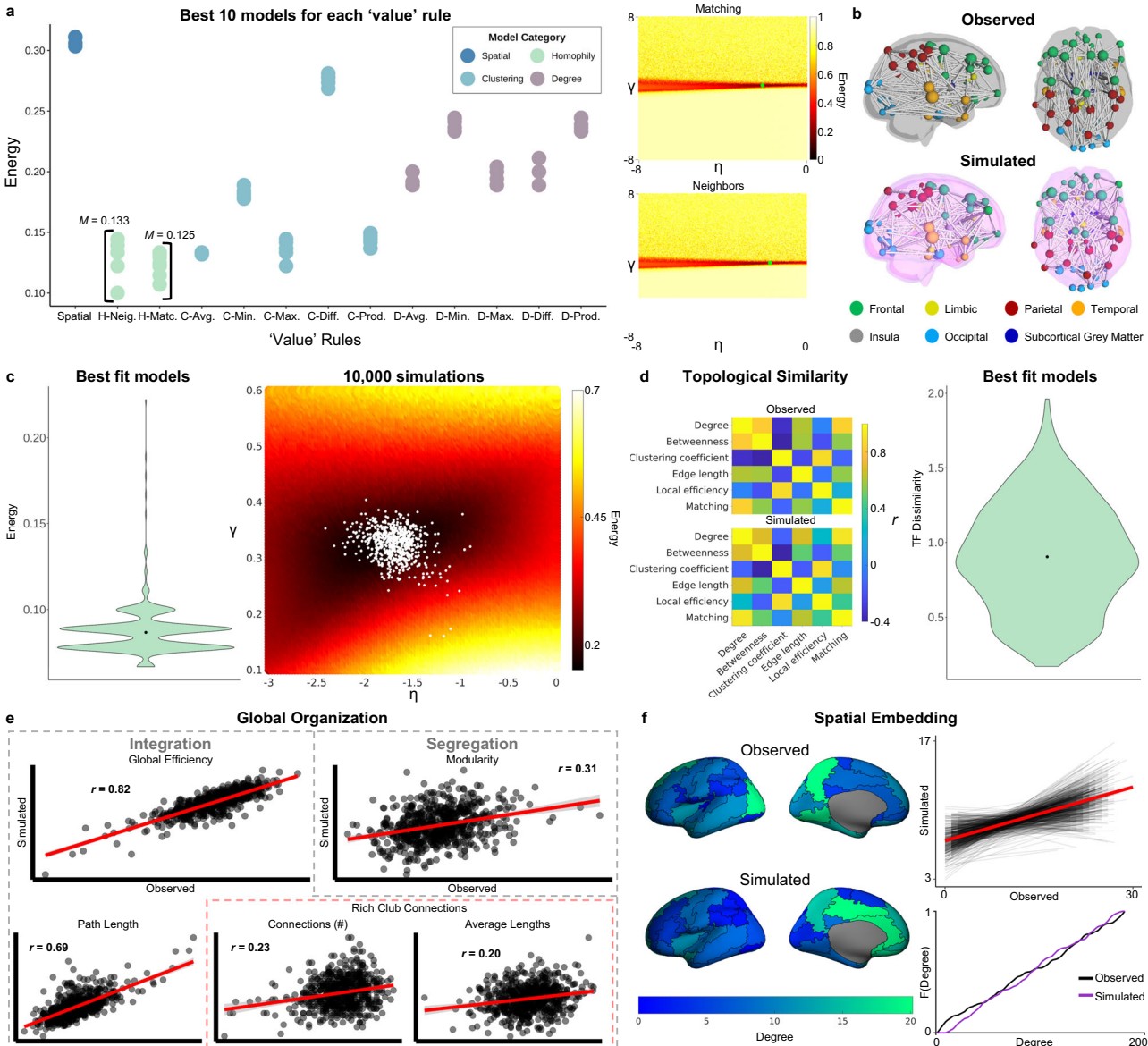

**Fig. 4 | A generative network model of neonatal networks. a** Energies from 90,000 simulations fit to the consensus network. The scatter plot contains the energies of the best 10 models for each 'value' rule. The average energy landscapes of the two best-performing 'value' rules, both of which are homophily principles, depict the energies for all simulations in the parameter space. Landscapes include a green point indicating the location of the best-fitting model. While 'neighbors' had the lowest overall energy, 'matching' had the lowest energy averaged over the top 10 models. **b** Representative well-fit observed network and simulated 'matching' model network. **(c)** The average energy landscape of the individually fit 'matching' models. The white dots indicate the location of the best-fit models for all participants. The violin plot demonstrates the distribution of energies of the best-fit models (min = 0.07, max = 0.22, M = 0.09, SD = 0.01). **d** The topological fingerprints of observed and simulated networks depict the average correlations between local organization measures. **e** Global organization of observed and simulated networks

were significantly correlated (using Pearson's correlation) for global efficiency ($p = 6.58 \times 10^{-155}$), number of rich club connections ($p = 6.98 \times 10^{-9}$), average length of rich club connections ($p = 3.84 \times 10^{-7}$), characteristic path length ($p = 3.76 \times 10^{-91}$), and modularity ($p = 2.49 \times 10^{-15}$). **f** Degree displayed significant spatial embedding using LMEM ($p = 4.96 \times 10^{-5}$). The surface plot depicts the average degree in the left hemisphere for the observed and simulated networks in all regions. The top right graph depicts correlational data between observed and simulated degree for every participant (grey) and the sample average (red). The bottom graphs show the cumulative density function of the average observed (purple) and simulated (orange) networks. Shaded area around best fit lines represent 95% confidence intervals. *** indicates $p < 0.001$, ** indicates $p < 0.01$, * indicates $p < 0.05$. The node plots in panel (b) and surface plots in panel (f) are adapted from AAL90 atlas in Shi F, et al. (2011) Infant Brain Atlases from Neonates to 1- and 2-Year-Olds. PLoS ONE 6(4): e18746. doi:10.1371/ journal.pone.0018746.

analysis, γ is smaller in the preterm group, though the difference is not significant (Supplementary Fig. 10).

To explore how these altered wiring constraints drive the emergence of organizational characteristics, we ran two GNMs with selected parameters determined by the mean of the optimal GA at birth-predicted parameters for preterm (η = -1.85; γ = 0.34) and term neonates (η = −1.74; γ = 0.32) (Fig. 6a). On average, small but significant differences were observed in the number of connections being added

for all connection types between term and preterm models (Fig. 6b). While the number of rich connections was significantly lower for preterm models (group effect: estimate = -0.20, $p < 2.00 \times 10^{-16}$), a larger difference in connections was found in local and feeder connections – with preterm models displaying on average fewer connections across all types (local: group effect: estimate = -0.69, $p < 2.00 \times 10^{-16}$; feeder: group estimate = -0.51, $p < 2.00 \times 10^{-16}$). This suggests a relative conservation of the number of rich club connections compared to local

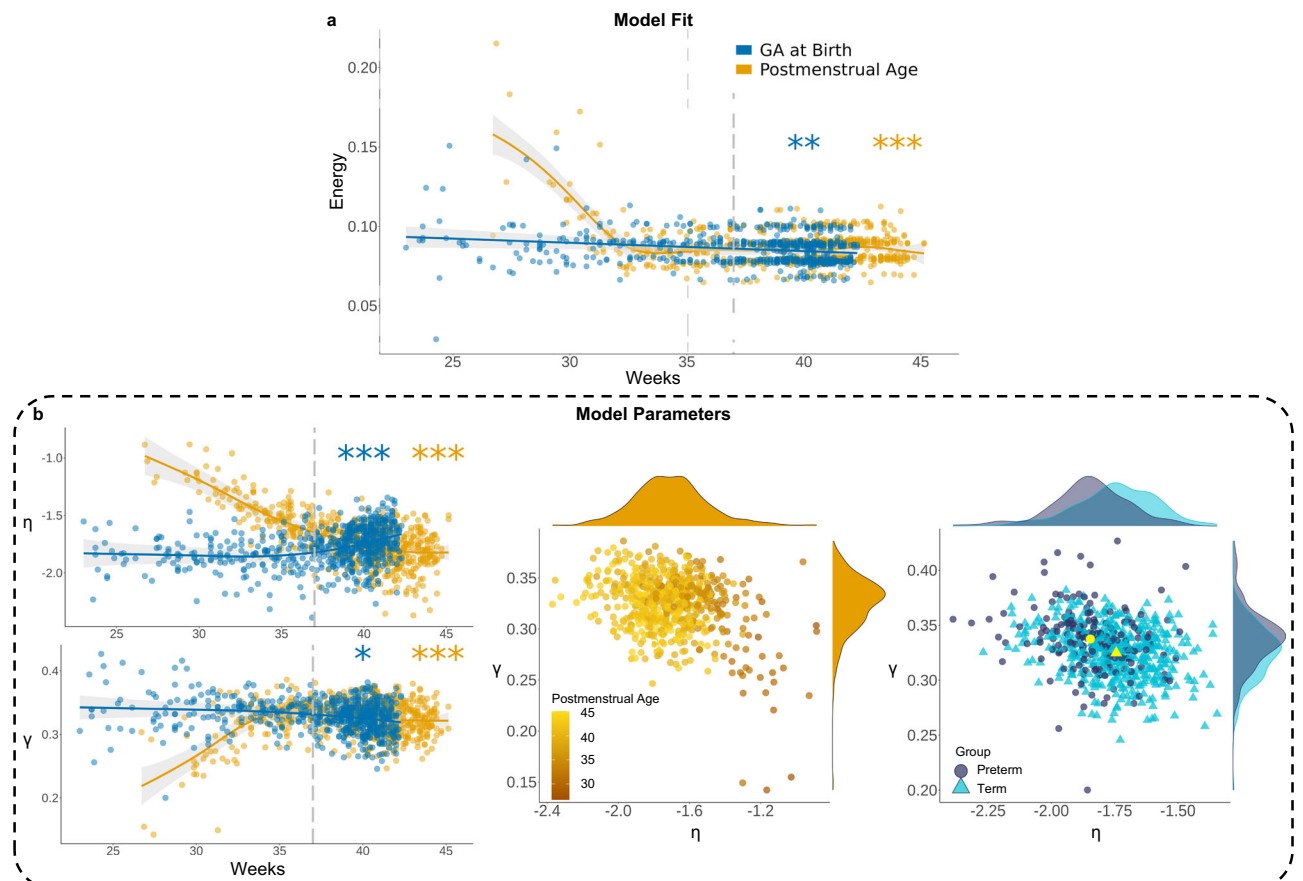

**Fig. 5 | Simulations highlight changes in economic wiring constraints across development. a** Energy significantly decreased across PMA ($p < 2.00 \times 10^{-16}$) and GA at birth ($p = 7.00 \times 10^{-3}$). The relationship between energy and GA at birth was linear. **b** η significantly decreased across PMA ($p < 2.00 \times 10^{-16}$) but increased across GA at birth ($p = 8.61 \times 10^{-6}$). γ significantly increased across PMA ($p < 2.00 \times 10^{-16}$) but decreased across GA at birth ($p = 0.018$). Model parameters predicted by PMA become more constrained (higher γ, lower η) as age increases. Model parameters predicted by GA at birth indicate that preterm infants have higher γ, but lower η compared to term infants. Yellow points indicate mean parameters for preterm (circle) and term (triangle) groups. All results presented are generalized additive models. Shaded area around best fit lines represent 95% confidence intervals. The grey dotted line indicates the cut-off for term birth (37 weeks GA or later is term-born). *** indicates $p < 0.001$, ** indicates $p < 0.01$, * indicates $p < 0.05$.

and feeder connections with the altered wiring conditions of preterm models. A more distinct difference between models, however, was observed in connection *lengths*. The preterm models had significantly shorter connection lengths compared to term models (Fig. 6c; $F_{connection\ length,iteration} = 267599$, estimated df = 8.96, $p < 2.00 \times 10^{-16}$, group effect: estimate = -0.50, $p < 2.00 \times 10^{-16}$). This difference in wiring lengths is consistent across all connection types (Fig. 6d; rich connections: $F_{length,iteration} = 29487$, estimated df = 36.22, $p < 2.00 \times 10^{-16}$, group effect: estimate = -0.30, $p < 2.00 \times 10^{-16}$; feeder connections: $F_{length,iteration} = 26233$, estimated df = 28.31, $p < 2.00 \times 10^{-16}$, group effect: estimate = −0.51, $p < 2.00 \times 10^{-16}$; local connections: $F_{length,iteration} = 18282$, estimated df = 33.47, $p < 2.00 \times 10^{-16}$, group effect: estimate = −0.69, $p < 2.00 \times 10^{-16}$). Despite preterm models forming shorter connections, rich club connections remained the longest connections to be formed, suggesting that the priority of forming 'valuable' connections was preserved. This analysis demonstrates that tight wiring constraints required for simulations of preterm networks is related to a preservation of vital, long-range connections but with overall shorter connection lengths.

## Discussion

### Rapid structural network development that occurs across PMA is altered in premature infants

Across neonatal development, networks rapidly increased in connection number and integration while segregation decreased. These findings are consistent with past work demonstrating that, across the gestational period, networks increase in density, rich clubs, and global efficiency, while the decrease in characteristic path length[44–47]. Our results suggest that increased efficiency is largely explained by network density. Relative to this increase in density, the networks are not actually becoming more efficient per se, but networks do become more modular across PMA. We observed fewer topological differences across GA at birth when controlling to density than with variable density networks, with modularity significantly decreasing and average rich club connection length significantly increasing. This contrasts with some previous research, which has shown increases in modularity across gestation[12,19,29,48]. Given our various sensitivity analyses, we think this apparent discrepancy is unlikely to be caused by network sparsity. We can think of two other possibilities: first, the present study differs from some previous work by including extremely preterm infants. Conflicting modularity results could reflect time points that are earlier in human development than were previously included[12]. Secondly, previous work has not included density-controlled analyses, making it hard to isolate topological modularity during this early developmental period.

Infants born at early GA had reduced network density relative to those born at term, resulting in less efficient and more modular networks. Increasing network density across GA at birth is consistent with existing evidence that preterm infants have reduced connectivity compared to term-born infants[14,27,49–51]. Longer path length and lower

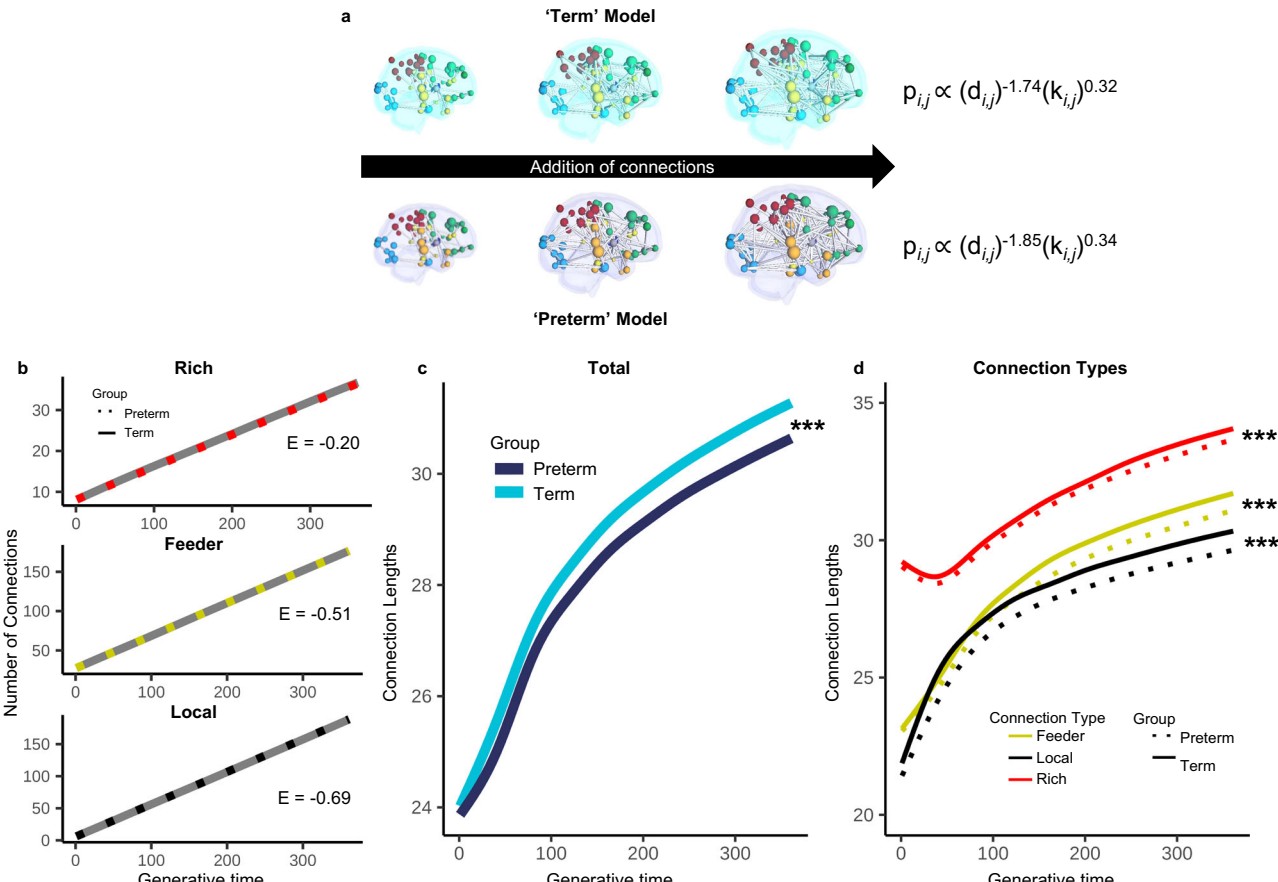

**Fig. 6 | Term and preterm representative models. a** Average term and preterm parameters were selected from birth-age predicted parameters and developmental GNMs were run with every step in the iteration process (the addition of a single connection) saved for comparison. Node plots are adapted from AAL90 atlas in Shi F, et al. (2011) Infant Brain Atlases from Neonates to 1- and 2-Year-Olds. PLoS ONE 6(4): e18746. doi:10.1371/ journal.pone.0018746. **b** Across the generative time (the addition of connections), preterm models formed on average fewer connections across all types (rich: $p < 2.00 \times 10^{-16}$; feeder: $p < 2.00 \times 10^{-16}$; local: $p < 2.00 \times 10^{-16}$), with the largest estimated differences being in local and feeder connections. The difference between term and preterm models was smallest for rich connections. 'E' indicates the estimated group difference. **c** The average length of connections significantly increased throughout generative time ($p < 2.00 \times 10^{-16}$). The term models formed significantly longer connection lengths compared to the preterm models across the entire generative time ($p < 2.00 \times 10^{-16}$). **d** This relationship was consistent across rich ($p < 2.00 \times 10^{-16}$), feeder ($p < 2.00 \times 10^{-16}$), and local connections ($p < 2.00 \times 10^{-16}$). Rich club connections had the longest lengths compared to feeder and local connections. Significant values represent the group difference in generalized additive models. *** indicates $p < 0.001$, ** indicates $p < 0.01$, * indicates $p < 0.05$.

global efficiency in preterm compared to term infants have also been found previously[12,19,21,27,52], though we show here that these trends are primarily explained by changes in density. Indeed, our results indicate that when controlling for density, infants born at early GA do not have significantly less efficient networks but do have significantly more modular networks.

Unsurprisingly, the number of rich club connections rapidly increases across both PMA and GA at birth with variable density networks. Crucially, this increase was primarily explained by increases in density across GA at birth, but not so across PMA. Put simply, preterm infants do not have inherently fewer rich club connections compared to term infants with density-controlled networks. This aligns with the theory that premature infants have conserved connectivity in core connections but reduced connectivity in peripheral regions[21,27]. However, our results also demonstrate that regardless of density, while the length of rich club connections significantly decreases across PMA, the opposite effect is found across GA at birth. In other words, long-range rich club connections form first[15,27]. Our results extend this by highlighting reduced rich club connection lengths for infants born at early GA. This difference implies an acceleration of short-range rich club connection preference in preterm infants compared to infants born at term.

## Network simulations demonstrate changing wiring constraints across PMA and GA at birth

This work is the first to apply generative network modeling to neonatal imaging data, but like studies in other populations, we show that trading off 'cost', defined as Euclidean distance, with 'value', defined as homophily, can accurately simulate structural networks[22,34,39,40,42]. Despite not being aimed to provide edge-level fits, these homophily GNMs with narrow parameters uniquely recapitulate realistic neonatal network properties. These properties are not captured by a plurality of networks and therefore the exploration of how homophily GNMs generate networks could provide new theories into economic constrains of networks. *Why homophily?* The homophily modeling principle emerges consistently as a necessary ingredient when simulating networks across different scales and species[22,34,38–41]. This consistency speaks to its capacity to capture some fundamental, locally computable, network property that simulates the formation of topologically desirable characteristics. One emerging theory is that homophily is a heuristic for efficient communicability – the traversability of information through a network between two points. In a recent paper utilizing spatially-embedded recurrent neural networks, researchers used a regularization term that incorporated both distance and communicability. The researchers found that a homophily rule is required to

fit the generative model to the resulting networks[53]. Thus, one possibility is that homophily reflects a locally computable metric that allows a network to form connections based on minimizing redundant communicability and that this in turn drives the characteristic topology of biological networks.

Importantly, even when controlling for model fit, we observed an increase in parameter constraints across PMA and a decrease across GA at birth. Network wiring becomes more selective across PMA – forming shorter, more valuable connections. However, this is accelerated if an infant is born prematurely. Tighter 'cost' parameters relate to forming shorter connections, while a corresponding increase in the 'value' parameter enables the formation of long connections if they are sufficiently important. Of course, the GNM does not provide mechanistic insight into how these changes in wiring constraint occur, but this shift in trade off, resulting the formation of different types of connections, provides a compressed account of why changing trade-offs shape the emerging topology. The increased constraint for preterm networks results in the preservation of vital rich club connections, despite an overall decrease in connection lengths.

### Generative network modeling interpretability and limitations

When interpreting the GNM parameters, it is important to consider what these models do and do not tell us. A well-fit generative model tells us about the global trade-off between the influence of connection cost (i.e., distance) and connection value (i.e., homophily) needed to recreate the high-level distributional properties of the observed networks. If a GNM does a good job of recapitulating these distributional properties then not only should the global distributions of simulated network properties be very close to those of the observed networks, but so too should the local spatial arrangements of those properties (i.e., the 'topological fingerprint') – a node with high betweenness centrality should have a low clustering coefficient, and so on. There is something specific about the characteristics of GNMs that can do this. Replace the homophily term with a different property (e.g., nodal strength), or change the parameters outside relatively narrow bounds, and the GNM's ability to replicate these properties of networks is much reduced. We included this type of modeling alongside our graph theory analyses because we wanted to test with the *economic constraints* needed to recapitulate these network properties change across the preterm to term timeframe. It is in this sense that these models are compressive: with just two calibrated global parameters, networks can be simulated that capture many of the complex distributional hallmarks of observed networks. These contextual details about the conditions necessary to best replicate network properties are only ascertainable through generative analyses. But it is important to consider what these models do not tell us.

Firstly, these models do poor job of placing edges in the right locations. The models are tuned to achieve distributions of high-level network properties, but the location of the connections themselves will vary and are often poorly matched to those in the observed networks. Likewise, these models produce some spatial embedding – high degree nodes in the simulations are also significantly more likely to be high degree nodes in observed networks – but this embedding is far from perfect. There are many likely reasons for this, including that we are working with very sparse binarized networks. Needless to say, the current application of GNMs offer insights into the trade-offs needed to yield realistic network features. They are not designed for, and should not be used for, edge-level exploration of network formation.

The second obvious way in which these models fall short is that they are not biologically mechanistic. These models are not attempting to simulate a biological process, so whilst they can predict the types of connections that form under different conditions, they do not explain why a given connection has formed. In other words, they are not biologically compressive. There are current methodological barriers

that this type of modeling that must be addressed before the field can move in this direction. We are currently simulating binarized networks without the capability to increase or decrease connection strength or prune connections. This not only limits the biological accuracy of simulations but also requires the observed networks to be binary, which is considered less reliable for representing topology than weighted networks[54]. Weighted GNMs that update connection strength over time, in addition to connection formation, have recently been established although are not currently placed to simulate large samples[55]. Secondly, during the pre- and neonatal phases, the actual volume of the brain changes rapidly; however, we are constrained to calculating the distance between regions with a single atlas. Recent work suggests that differences in cortical geometry may influence connectome topology[43]. Thus, the addition of a biological informed measure into the GNMs likely would improve the accuracy of simulations. While we see substantial replication of topological metrics, GNMs are not identical to observed connectomes and thus cannot be interpreted as exact replications. Thirdly, GNMs use Euclidean distance between regions as the 'cost' metric. A limitation of this metric, as compared to fiber lengths, is that it does not consider distances between regions in different hemispheres being larger than those within the same hemisphere. For GNMs, however, fiber lengths cannot be used as the 'cost' metric because they don't exist between all regions, and thus must be interpolated. It is of note, however, that fiber length and Euclidean distance are correlated – as strongly as $r = 0.77$[55].

### Project design and neurobiological limitations

Macroscale binarized connectomes are compressed representations of neurobiology. A connection being present or absent between two regions could reflect the strength or developmental stage of that connection – not the actual existence of such a connection. Thus, the rapid increase in network density across postmenstrual ages should be interpreted within binarization limitations, specifically in consideration of increasing density. Premature birth is associated not just with reduced connectivity, but also an impairment to the migration and maturation of neural cells[56]. The preterm period overlaps with the pre-myelination stage, during which there is rapid movement and maturation of glial cells[57]. Disruption during initial stages of glial cell development become more apparent in infancy in the myelination process[58]. However, topological analysis of macroscale networks is likely influenced by these early cell-level difference, as it relates to decreasing anisotropy and associated metrics during the preterm period[57]. Thus, our topological analysis of the macroscale neonatal network cannot be interpreted as solely connection-level differences and is likely, if not definitely, encompassing the disruption of pre-myelination development in infants born at early GA. In addition, we used thresholded networks which has important limitations to consider. By thresholding, we remove potential false positive connections from tracking errors[59] but also likely remove real, weak connections. While we have provided variable density, controlled density (Fig. 2), and multiple density analyses (Supplementary Fig. 2) to demonstrate the effects of these choices, it is important to note that there is no 'right' thresholding method, and each has important benefits and limitations.

Furthermore, caution must be taken when interpreting extremely early PMA scans and the effects of preterm birth. Firstly, early PMA fiber tracking is challenging and thus likely includes more errors than tracking of infants closer to term age. Secondly, it cannot be overlooked that the differences observed in this study are potentially related to factors that precede the event of a preterm birth. Indeed, recent advancements in fetal fMRI have led to the ability to identify infants that will be born preterm from reduced fetal fMRI signal[60]. These results, along with the high number of environmental factors associated with preterm birth (e.g., low socio-economic status)[61] suggests the presence of differences for those born early that may

precede the premature birth itself. Thus, it is important to not interpret the present study as causal.

## Methods

### Participants

This sample of infants (n = 758, 26–45 postmenstrual age) was collected by the Developing Human Connectome Project (dHCP). The dHCP is a collaborative effort between King's College London, Imperial College London, and Oxford University that collects neuroimaging data from neonates[62]. To obtain a cross-sectional sample, we only included the first scan for each neonate. Neighboring DWI correlation identified n = 34 subjects with poor scan quality who were removed from the sample[63]. An additional n = 94 subjects were removed for repeat scans, resulting in n = 630 participants analyzed in this study (n = 447 term-born). A secondary analysis was completed with a subsample of term-equivalent age neonates (n = 487, preterm: n = 40, term: n = 447). This secondary analysis serves to validate the statistical controlling of PMA and GA at birth (see "Statistics" below for detail) as well as to address realistic concerns about the tractography quality of early PMA scans. To provide further confidence in term-equivalent age interpretations, we have also included a propensity-matched analysis with additional preterm infants scanned at term-equivalent age (n = 80). With this data, we create two groups (each n = 120) matched in sex and PMA – one group with infants born preterm and the other with infants born at term. There is no significant difference in head circumference between these groups (p = 0.282). Average network matrices of early PMA networks compared to late PMA networks can be found in the supplementary materials (Supplement Fig. 11d).

### MRI acquisition and preprocessing

dHCP MRI acquisition protocol was developed specifically for imaging neonates and included neonatal-specific methods, such as neonatal head coil and positioning devices[62]. Diffusion imaging was performed with multiband EPI acquisition, four-phase encoding directions, gradient demand optimization, and a multi-shell diffusion sensitization scheme[64,65]. The data has been preprocessed by the dHCP with their diffusion SHARD pipeline which denoises and corrects for Gibbs ringing, motion, eddy currents, and susceptibility artefacts (for details see Christiaens et al., 2021)[66].

### Connectome construction

The data used in this study was accessed in semi-processed format from DSI Studio[67]. Restricted diffusion imaging and reconstruction via generalized q-sampling imaging using a sampling length ratio of 1.25 was performed by Dr Yeh (accessed at: https://brain.labsolver.org/hcp_d2.html)[67]. Following previous neonatal studies, fiber tracking was performed in DSI Studio with a turning angle of 35°, step size of 1 mm, maximum fiber count of 5,000,000, minimum fiber length of 30 mm, maximum fiber length of 250 mm, and a random initial seed propagation direction[51]. Connectivity type was count-end, indicating that streamlines will be counted as present between two regions when the streamline ends within those regions. The scans were registered to a neonatal template[68] and parcellated using the Automated Anatomical Labeling 90 (AAL90) atlas for neonates[69]. The AAL90 adult atlas was constructed with anatomical localization and from which neonatal templates, tissue probability maps (grey matter, white matter, and cerebral spinal fluid), and brain parcellation were created[69].

As a minimum fiber length of 30 mm could be too restrictive for neonates, an additional analysis was added to explore the effects of minimum fiber lengths 5 mm, 10 mm, 20 mm and 30 mm during tractography. This analysis demonstrated that the 30 mm tracked networks were denser than 5 mm and 20 mm – indicating non-linear relationship between minimum fiber length in tractography and density (Supplementary Fig. 11a). Crucially for the present study, the local topological arrangement of the networks is similar, indicated by

topological fingerprints (Supplementary Fig. 11b). In addition, Spearman rank order correlations reveals that the order of connectivity of regions is highly similar (5 mm to 10 mm $rs$ = 0.75; 10 mm to 20 mm $rs$ = 0.74; 20 mm to 30 mm $rs$ = 0.74). A full organizational analysis, connectivity matrices and topological fingerprints can be found in the supplementary materials (Supplementary Fig. 11).

As we are utilizing binary networks, we chose a strict thresholding approach to allow for high levels of confidence in the surviving connections in addition to conducting many sensitivity analyses to explore the effects of this thresholding. To reduce the rate of false positive and negative detection in tractography, a 60% consensus threshold was initially applied[59]. This thresholding results in streamlines between regions that are present in 60% of the sample being retained. Following this threshold, we conducted two density-based analyses. First, following previous generative network modeling methods, we performed a 325 binarization threshold to yield an average network density of 10% across the sample[22,39]. This thresholding technique was repeated for the propensity-matched analysis (n = 240) with a 390 binarization threshold to yield an average network density of 10%. Second, to explore the effects of density on topology, we completed a density-controlled analysis where each individual had a proportion binarization threshold that yielded an exactly 10% dense network. By applying these two thresholding techniques, we can conduct a standard GNM analysis with varied density, followed by an exploration of how controlling for density effects topological interpretations. An additional analysis explored how progressively more sparse networks (by increasing the absolute threshold) resulted in networks ranging from 24%–8% density. The analysis reveals how density relates to changes in topology, topological fingerprints and representative connectivity matrices can be found in the supplementary materials (Supplementary Fig. 2). The propensity-matched sample was included in this exploration of multiple densities and showed that global efficiency and modularity differ significantly between groups across 5–24% average density networks (Supplementary Fig. 2e). Together these additional analyses demonstrated that while more dense networks would yield overall more efficient, less modular topology, that this does not change significant effects between term and preterm groups.

For GNM selection, a group-representative consensus network was generated with an algorithm that conserves the connection-length distribution of individual participants (code access: https://www.brainnetworkslab.com/coderesources)[70]. This consensus network was utilized to run multiple types of GNMs to select the optimized 'value' rule, rather than running multiple GNMs on individual participants.

### Graph theory

All graph theory network measures were calculated from binarized networks using the Brain Connectivity Toolbox (BCT) in MATLAB 2020b[10]. This study employed local statistical measures of degree (i.e., the number of connections), betweenness (i.e., participation in shortest paths), clustering coefficient (i.e., the proportion of neighbors that are also connected to each other), edge length (i.e., the sum of the Euclidean distances of all nodal connections), local efficiency (i.e., the inverse of the shortest path length for a neighborhood of nodes) and matching (i.e., the amount of overlap in connectivity patterns between two nodes). Global statistical measures of maximum modularity (i.e., separating the network into nonoverlapping groups using Newman's spectral community detection), characteristic path length (i.e., the average shortest path length), and global efficiency (i.e., the average inverse shortest path length) were also used[28]. Rich club nodes were defined in the consensus network utilizing the normalized rich-club coefficients[71]. Significant rich club nodes were identified by having a degree at which the rich coefficient was significantly higher than chance, calculated from 1000 null models using randmio_und() from the BCT. Two significant rich clubs were identified, one including 27

nodes and the second including only two nodes. For the results of this analysis, see supplementary materials (Supplementary Fig. 6). The 27 rich club nodes from the consensus network were used to define connection types for all participants as rich (rich node to rich node connection), feeder (rich node to non-rich node connection), and local (non-rich node to non-rich node). Average length of connections was also explored across each connection type, defined by the Euclidean distances between connected nodes.

## Generative network modeling

Generative modeling was performed in three steps. First, the most accurate 'value' rule for simulating neonatal networks was identified. Second, the optimized generative modeling equation was run to find every participant's best-fit model. Third, term group-associated parameters were used to run two representative models where connections were assessed at iteration.

We assessed the accuracy of 'value' rules by examining the performance of 13 $K$ rules and matching them to a group-representative consensus network (Supplementary Fig. 8)[39]. Once the most accurate $K$ rule was identified by using three distinct model fitting methods that have been developed in prior research (see *model fitting* below), we fit individual models to every participant.

Models were grown from a sparse seed network (10% of the whole network) calculated by the edges present in 95% of all participants (Supplementary Fig. 12). To determine parameter pairs, a grid search was performed to evenly space parameters within a given range[22,41]. Initial GNMs were run with a wide parameter range (-8 ≤ η ≤ 0 and -8 ≤ γ ≤ 8) for 90,000 simulations to identify the best 'value' rule. For individually fit models, we narrowed the parameter space (-3 ≤ η ≤ 0 and 0.1 ≤ γ ≤ 0.6) based on the energy landscape (Fig. 4a) and ran 10,000 simulations. For the 'term' and 'preterm' representative parameters were calculated by the group average of GA at birth-predicted parameters while controlling for PMA in GAMs. Group representative models were run with 1000 simulations with a single parameter combination.

*Model fitting*. The primary assessment of generative model fit is with the energy function. The energy function calculates how dissimilar each simulated network is to observed networks based on the Kolmogorov-Smirnov statistic (KS) across degree (k), clustering coefficient (c), betweenness centrality (b), and edge length (e) distributions[22,34,38–43]:

$$E = \max(KS_k, KS_c, KS_b, KS_e) \qquad (2)$$

A good fit between simulated and observed networks results in the minimization of energy. A lower output value of the energy equation indicates small differences between simulated and observed networks in degree, clustering, betweenness centrality, and edge length. Out of all value rules, the homophily rules, 'neighbors' and 'matching', had the lowest energies (Fig. 4a). These rules are fundamentally very similar: 'neighbors' is calculated by the number of overlapping connections between neighboring nodes and 'matching' is simply the normalized version of the 'neighbors' rule[22,39,43]. Thus, for both rules, if two nodes have a greater proportion of shared neighbors, a connection is more likely to form than if the proportion of shared neighbors is low.

Most previous research relies solely on the application of the energy function to assess model fit[22,34,38–43]. In addition to this step, we considered topological dissimilarity and spatial embedding of the top-performing models to determine which model best captures network organization beyond the four local measures included in the energy equation.

Topological fingerprints (TFs) were used to calculate dissimilarity in the relationship between local measures (e.g., clustering and betweenness) between observed and simulated networks[40]. TFs are correlation matrices of local statistics within a network while TF dissimilarity is the Euclidean norm of the difference between observed and simulated TFs demonstrated by the following equation[40]:

$$TF_{dissimilarity} = \sqrt{\sum i \sum j (TF_{observed_{i,j}} - TF_{simulated_{i,j}})^2} \qquad (3)$$

Thus, TF dissimilarity calculates the degree to which simulated models accurately replicate the relationship between local measures found in the observed network[40]. Minimization of TF dissimilarity, therefore, indicates a well-fit model. Out of the four best-performing models based on energy, 'neighbors' performed the best followed by 'matching' for TF dissimilarity (Supplementary Fig. 8c).

Spatial embedding is the relationship between the simulated and observed networks' local statistics at each node[22]. Thus, spatial embedding is a measure of the accuracy to which the GNM is simulating local organization. Significant positive relationships across multiple local statistics would indicate a well-fit model. Out of the top four performing models based on energy, 'matching' yielded the most spatial embedding with four out of the six local statistics significantly correlated between the observed and simulated networks, while neighbors had only one significant relationship (Supplementary Fig. 8d).

## Statistics

To assess variation in measures across PMA and GA at birth, generalized additive models (GAMs) with penalized splines were performed with the *mgcv* package in R v4.1.2[72–74]. Models were fit with the restricted maximum likelihood (REML) method[72]. Cubic regression splines were used as the basis function[72]. The curvature of the model was assessed by fitting the model with cubic shrinkage splines[72]. PMA, sex, head circumference, translation, and rotation were controlled for in GAMs on observed data when exploring GA at birth. When looking at PMA, the same variables were controlled for, in addition to GA at birth. For local statistics, $p$-values were corrected using Benjamini-Hochberg false discovery rate correction[75]. Spatial embedding, which is replication of local organization, was calculated in 80 cortical regions. For consensus network GNMs, spatial embedding was assessed with Pearson correlations. For individually-fit GNMS, spatial embedding was assessed with linear mixed effects models with participants and nodes as random effects with the *lme4* package in R[76]. For the propensity-matched analysis, Welch two sample t-Tests were performed in R to examine group differences in modulatory, global efficiency, and GNM parameters.

## Reporting summary

Further information on research design is available in the Nature Portfolio Reporting Summary linked to this article.

## Data availability

The derived data generated in this study is available at: https://osf.io/ng43c/. Source data are provided with this paper. The semi-processed version of dHCP data used in this publication is available at: https://brain.labsolver.org/hcp_d2.html. The dHCP original data is available on their public repository: https://data.developingconnectome.org/app/template/Login.vm. Source data are provided with this paper.

## Code availability

The analysis was completed in MATLAB and R. All code is available at https://github.com/alexamousley/neonatal_generative_network_modeling or https://doi.org/10.5281/zenodo.13913488.

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

## Acknowledgements

Data were provided by the developing Human Connectome Project, KCL-Imperial-Oxford Consortium funded by the European Research Council under the European Union Seventh Framework Program (FP/2007-2013) / ERC Grant Agreement no. 319456. We are grateful to the families who generously supported this trial. We want to thank Dr Fang-Cheng Yeh for sharing fiber tracking files and adding neonatal tools to DSI Studio. We thank Dr Stuart Oldham for developing the code to visualize region-specific organizational measures. We also thank our collaborators, Dr Petra Vértes, Dr Agoston Mihalik, and Dr Giacomo Bignardi for their intellectual contributions. A.M. is supported by the Gates Cambridge Foundation. D.E.A. is supported by Medical Research Council Program Grant MC-A0606-5PQ41 and by the Gnodde Goldman Sachs endowed Professorship in Neuroinformatics awarded to the University of Cambridge. D.A. and D.E.A. are both supported by the James S. McDonnell Foundation Opportunity Award and the Templeton World Charity Foundation, Inc. (funder DOI 501100011730) under the grant TWCF-2022-30510. All research at the Department of Psychiatry at the University of Cambridge is supported by the National Institute for Health and Care Research Cambridge Biomedical Research Center (NIHR203312) and the NIHR Applied Research Collaboration East of England. D.A. was also supported by the Imperial College Research Fellowship with support from Schmidt Sciences and a Nature Computes Better Opportunity Seed with the Advanced Research + Invention Agency (ARIA). This publication does not necessarily reflect the views of the funding agencies.

## Author contributions

All authors conceived the analysis. A.M. constructed connectomes, ran the simulations, conducted the analysis, and drafted the manuscript. D.E.A. and D.A. provided critical manuscript reviews and edits.

## Competing interests

The authors declare no competing interests.
