## [Transparent Peer Review file · Nature Communications]

Premature birth changes wiring constraints in neonatal structural brain networks

Corresponding Author: Ms Alexa Mousley

Version 0:

Reviewer comments:

Reviewer #1

(Remarks to the Author)

In this manuscript Mousley and colleagues analyse a publicly available data of neonatal MRI (the Developing Human Connectome Project, dHCP). They perform a “classic” graph theory characterisation of the neonatal brain, using diffusion MRI tractography from the dHCP, preprocessed by Dr Fang-Cheng Yeh (not an author of this manuscript), and then they fit generative network models to their data, assessing optimal model parameter fits that differ between term and preterm infants.

This is a timely and interesting study, making the most of the new resources generated for the neonatal brain, and publicly available data. However, I also think the article has some flaws; I am unsure about some of the methodological decisions taken, which I think require further justification. In particular, I have some concerns about how some elements of the first part of the manuscript (graph theory analysis) are performed, and I think more details should be provided to reassure readers that structural connectivity networks for very preterm babies included in the analysis are anatomically valid. I think the generative models approach used in the second part of the manuscript is novel and interesting, providing a useful way of interpreting results and going beyond just superficial graph theory analysis.

Please find below a few suggestions and constructive criticisms to some of your methodological and study design choices. I hope these may help you improve the manuscript.

In my opinion the analysis design has the following major potential issues:

1) Networks obtained are strongly dependant on the tractography algorithm used and the arbitrary choices made (e.g., 60% consensus thresholding, minimum 30mm fibre length, minimum 325 streamlines). I am uncertain of very low densities obtained overall (less than 12% for all subjects), and in particular for subjects scanned at less than 32 weeks postconceptional age. I would suggest the authors consider an alternative tractography algorithm that is more flexible to the peculiarities of the neonatal brain, such as mtrix’s MSMT-CSD, which has been used by previous work using this data (see for instance <https://doi.org/10.1016/j.neuroimage.2018.10.060> and <https://doi.org/10.1073/pnas.2023598118>). Tractographies shown in Figure 3A are a bit concerning. Do they correspond to neonatal brain WM anatomy? Apparently just a few central streamlines have been reconstructed.

2) In the data presented by the authors, there is a strong link between network density and age at birth and age at scan. Since network density is, by definition, massively driving measures of integration and segregation, it is unclear whether measures presented in Figure 2 are adding much.

3) Postconceptional age at scan and age at birth are strongly related in these data, but this relationship is particularly true for subjects scanned at <37 weeks postconceptional age, making very difficult to disentangle one factor from the other.

4) At >37 weeks postconceptional age the authors have a mixture of term and preterm born babies. Any conclusion about brain maturation (postconceptional age at scan) will be confounded by the effect of preterm birth.

I think the authors should be very careful with their statistical design in order to address these issues. For points 3 and 4, perhaps they can address specific hypotheses by using the appropriate sample, for instance, typical development with term subjects only; effect of preterm birth with term and preterm babies scanned at term equivalent age (so they are comparable in

terms of postconceptional age at scan); and early network development (<37 weeks) with babies scanned <37 weeks only.

Please see below a list of detailed comments:

-L118-119: Is the sample size corresponding to the number of participants or the number of sessions included? Note that most preterm babies were scanned twice as part of the dHCP. In the methods section the authors mention they are excluding repeated scans, but it is unclear what sessions were excluded (i.e., preterm scanned soon after birth, or preterm at term equivalent age)? Why exclude sessions, instead of include all subjects and perform sub-analyses? Or use LME to deal with repeated measures.

-L135-136: "This is to be expected; children born premature have to wait longer before they can be scanned." -> I find this is a strange statement, what do the authors mean by "wait"? Preterm babies in the dHCP are scanned twice, soon after birth and at term equivalent age. Preterm babies can be scanned in a larger range due to being ex utero for longer. Due to dHCP design (scanning up to 44 weeks PMA), preterm babies can be scanned between 24-44 weeks PMA, while term can only be scanned between 37-44 weeks PMA, causing a negative correlation between age at birth and age at scan. This may also be driven by their exclusion criteria.

-L141-150. I am unconvinced about what can actually be concluded from the authors' assessment of network density. This is extremely dependant of tractography algorithm selected, and ad-hoc parameters chosen. Were the same number of streamlines generated for all subjects? With the same number of seeds? Where were seeds placed? Tractography is particularly tricky for subjects scanned at less than 32 weeks postconceptional age. It would be reassuring to see a couple of examples of such young infants' tractographies, perhaps it could be added to Figure 1A.

-L138-150. It is unclear if association between density and postconceptional age / GA at birth are corrected by one another statistically. Here and elsewhere, the effect of preterm-birth may be confounding the results associated to postconceptional age, i.e., the authors are assessing the combined effect of preterm-birth and post-conceptional age, rather than postconceptional age in typically developing individuals.

-L138-150 In a similar note, I am not sure whether the authors can conclude much about the topology of typically developing babies when mixing term and preterm individuals. It is expected that they will find a shallow U around 36-37 weeks, since this is the age where they are starting to see term- (or nearly term) born babies. I would suggest the authors perhaps could look at associations with postconceptional age and gestational age at birth independently for term and preterm babies, and make clearer whether they are correcting for the appropriate confounders.

-L152-202. Please clarify if assessing binary or weighted graph theory metrics.

-L152-202. Authors have previously shown massive variability in density, which, by definition, massively influence graph theory metrics of integration and segregation. I wonder whether the results shown would hold if comparing brain topology independently of differences in density, for instance with a proportional thresholding approach.

-Figure 3A. I am not sure how to interpret tractographies at 30 and 35 weeks shown here. These tractographies look a bit strange, not corresponding to perinatal WM anatomy.

-Figure 4E. I wonder how much of global efficiency's variance is explained just by network density, and hence, how much is added by the other topological aspects of the generative model. In L311-315 and Supp Fig 7 the authors suggest that there are no differences in model fit due to density. If I understand correctly in methods section, it is stated that a proportional threshold at 10% is applied in order to control for differences in network density. But in Figure 1C it is clear that network density is below 10% for many subjects already, so a 10% proportional threshold won't make a difference. I would suggest to try a 5% threshold instead.

-Perhaps it would be good if it could be clarified that binary networks are being modelled (so readers can interpret the results without having to read methods section in detail). Also, please clarify how are long-/short-range connection defined; and clarify how distance is calculated, e.g., Euclidian distance between ROIs, or streamline (WM) distance between ROIs? If the former, how do you take into account that the distance between regions of different hemispheres (or lobes) that are next to each other is much larger than the Euclidean distance between their centroids?

-Figure 6. Is the preterm model representative of preterm at term equivalent age, or preterm during preterm period?

-L505-516. Please provide more details about the number of streamlines generated, where were the seeds placed? How was WM and GM parcellated? Please provide more information about the 60% consensus thresholding performed, and its justification. How was the "325" streamlines threshold selected?

-L524. Please clarify if using weighted or binary graph theory metrics.

-L510. Isn't 30mm minimum fibre length too restrictive for preterm babies?

OTHER MINOR ISSUES:

-Abstract: why is in utero imaging "poor"? Perhaps this is a bit too ambiguous and unspecific. Fetal MRI has advanced a lot

recently, and ultrasound is fit for purpose.

-The term "birth age" is a bit unusual in the neonatal literature. I would suggest to change "birth age" for "gestational age at birth" or "GA at birth"?

-In Figure 1C it is a bit unclear what is significant at $p < 0.001$. Please include in the legend the statistical model tested.

-Figure 2A why association with postconceptional age only starts at ~32 weeks in this panel?

-L205. No fetal data assessed in this manuscript, so please be careful referring to "prenatal" development.

-L208. It is unclear if differences are due to ex utero exposure, or if there is something different in the brain of preterm born individuals, which may be associated to the reason why they are born preterm.

-Methods, L512. The authors refer to Fitzgibbon et al for the neonatal space. Do they mean to reference Schuh et al. template instead?

Reviewer #2

(Remarks to the Author)

In this paper, the authors present a network analysis of structural connectivity data acquired from a large cohort of neonates (dHCP). Through a series of descriptive analyses performed on binarized brain networks, they demonstrate an increase in network density with consequent decreases in modularity, path length and increases in efficiency. Using a simple generative procedure, the authors generate simulated networks that share similar properties with the empirical connectomes and use the parameters of these models to infer putative constraints on early network development. Overall, the observed network changes replicate previous findings, and the generative network modelling approach adds some interesting novel aspects, however, as written, the paper is difficult to follow and I have some methodological concerns that limit my enthusiasm for the study. I have listed some specific comments below.

1. Using a consensus thresholding approach, the authors find that network density is lower in younger infants, with a preservation of longer connections. The Methods state that a minimum of 30mm cutoff was used for tractography. This strikes me as very high, considering at 25 weeks the occipital-frontal diameter is only around 75-80mm. Presumably, in the youngest infants, many shorter fibres would thus be rejected prior to construction of the network matrices? The authors state that head circumference was considered in some statistical analyses but this would not account for the absence of connections prior to analysis.

2. Network models are necessarily an abstraction from the neurobiology of the brain, however, no commentary is provided on how these network-based parameters may relate to the well-described differences in white matter development following preterm birth. Significant differences in white matter anisotropy have been observed across most of the developing white matter, largely driven by disruptions to (pre)myelination processes. There is extensive literature on the establishment, timing and location of white matter tracts across cortical areas during this period – see work by Emi Takahashi amongst others. The authors should consider their findings in light of this large body of existing work.

3. In the Results section (para 2.) the sample is described as scanned from 20 to 44 weeks and through the Results, trends are described from '20 weeks to..'. No infants were scanned at 20 weeks, the youngest age MRI was acquired in the dHCP is 26 weeks, with the youngest age at birth at 23 weeks. Please correct.

4. On this, 'postconceptional' age is a poorly defined term. In this context, postmenstrual age is preferred as a more precise estimation of age— see <https://publications.aap.org/pediatrics/article/114/5/1362/67715/Age-Terminology-During-the-Perinatal-Period>. Similarly, 'ex-uterine' is typically 'extra-uterine'.

5. Throughout, results are split based on 'post-conception' and 'birth' age. I find this makes the results difficult to follow. Digging into the methods, it appears that postconceptional age were controlled for in GAM models and although this isn't stated anywhere in the results section, I assume that all the 'birth age' results are corrected for age at scan? No information is provided on how this correction was performed, was postconceptional age included as an additional smooth term in the birth age models? Were the postconceptional age models also corrected for birth age? These two parameters are obviously closely related, it is unclear why, when using the flexible GAM framework, a unified model wasn't presented including additive smooth terms for both age at birth and age at scan?

6. How was the significance of the spatial embedding results determined? There are several associations with highly significant results ($p = 10^{-4} - 10^{-6}$) that do not appear well-represented by the accompanying plots. Figure S6, for example, where highly variable positive and negative correlations between empirical and predicted clustering coefficients are displayed with no apparent average effect and little correspondence between surface plots are presented as $p < 0.001$? Are the statistics suitable considering spatially correlated datapoints?

Reviewer #3

(Remarks to the Author)

In this submission, the authors fit generative models to neonatal structural connectomes. They report changes in network

properties of connectomes and that their models fit the data reasonably well.

The results are interesting, but I have a suspicion that much of the findings can be explained by age-related changes in density (effects that the authors describe) but also data-quality issues associated with the acquisition and subsequent processing of neonate imaging data.

There's also a tendency to treat the models as "mechanistic". That is, any changes in estimated parameters are interpreted as changes in the trajectory of the real network. "We see an increase in η with age? Well, older brains must be subjected to stronger spatial constraints." The generative models are simply compressive. They take data and estimate the best fit given the model. But even great fits do not imply that the model is a plausible explanation of the network's trajectory.

In short, I think the submission is fine, but it has clear flaws that, in my opinion, will be very difficult to address.

Line 64 – Why the different citation style for the reviews?

Line 78 – Generative models are useful but they sure are not mechanistic. The models used here and by others are simply *not* developmental models. They are compressive. They identify a set of parameterized wiring rules that create synthetic networks. The process by which the network is generated has *nothing*, not even stylistically, in common with the process by which connectomes develop.

Showing that the model parameters vary with age or are correlated with different demographic/behavioral measures is an interesting exercise, but it doesn't imply different mechanisms of brain growth.

Most damning, the generated networks do not look like connectomes. They match a bunch of high-level distributional traits – e.g. degree, clustering, betweenness, edge length distribution. But the edge-level matches are just not there. That is, the actual wiring diagram, despite "baking into the model" the spatial locations of nodes, does not match what we measure in reality.

Line 103 – what sort of data quality assessments are performed on the MRI data? It's not unreasonable to imagine that data quality for smaller brains in which white-matter is still developing have poorer signal than, say, an adult brain for which most dMRI acquisitions are optimized. It seems possible that many of the reported developmental effects could, in fact, reflect changes in data quality with age.

Figure 1, panel D. The matrices are not particularly informative w/o node/region labels. For the average binarized network, the colorbar label is missing.

Line 142 – Does density really double during this period? That is, does the number of white-matter links actually increase that much? I know there's a period of myelin refinement, but the idea that there are *brand new* tracts forming does not seem plausible. It seems more likely to me that there are issues with dMRI and the acquisition (and maybe tractography) that make reconstructing some of these tracts difficult and in a way that the difficulty of reconstruction varies systematically with age.

Line 145 – How do you estimate a fractional degree of freedom? Not a serious complain, but it's not obvious to me.

Line 152 – Of course network modularity decreases, it's density dependent. If you create random networks with gradually ramped up density, their modularity will decrease monotonically. Careful comparison against null models is absolutely needed to support these claims. It's not obvious that the reported effects have to do with density or true changes in organization of the networks. Frankly, the same is expected for the other network metrics; density biases their values. I haven't seen it demonstrated, but I'm sure there's a similar density effect for rich clubs.

Figure 2 – It's not surprising that mean RC edge length (in ED) increased with age, right? Even if the exact same edges and RC members are preserved at every age, growing brains/volumes mean that edges will tend to grow.

Line 271 – I'm genuinely curious if the authors can shed some light on why homophily does so well as the "value" term. The explanations in earlier generative modeling papers were never fully satisfying. I also feel like homophily is kind of the same thing as the distance penalty. If two nodes are near one another, we'll connect them together. Therefore, nearby nodes also have similar connectivity profiles, making it even more likely that they connect. How does the homophily rule promote long connections? I'm still curious about that and have been since our first generative modeling papers. But it must be real, there's a dozen or so papers that have all replicated the initial findings.

Version 1:

Reviewer comments:

Reviewer #1

(Remarks to the Author)

In this revised version of their manuscript, Mousley and colleagues have clarified or addressed many of my initial concerns. They have performed tractographies with different parameters (including minimum streamline length), and assessed and

clarified the effect of network density on their results, which I think greatly improves their study.

1) My main remaining concern is that given that one of the aims of the study is to characterise the effect of preterm birth in brain development, I do not understand the decision to exclude all the valuable scans of preterm babies scanned at term equivalent age (i.e. GA at birth < 37 with PMA>37 weeks) which are publicly available as part of the dHCP dataset.

In their response to my previous comments they justify this decision based on the idea of avoiding confounding factors: "As the main purpose of this project was to perform generative network modelling, we chose to select a cross-sectional sample to avoid potential confounding factors in the simulation analysis". However, I have the opposite impression, by not including preterm at term scans they are massively impairing the capacity of their models to deal with the confounding effect of GA at birth and PMA at scan. Preterm at term scans would help them to better tune GAM models allowing the authors to actually disentangle the associations with preterm birth from the associations with brain maturation. Without the preterm at term scans I am not sure the authors can hold any interpretation regarding the effect of preterm birth in brain development, and as far as I understand, they are "only" characterising different aspects of brain maturation. Since they are not including comparable data (e.g. term babies vs preterm babies scanned at term equivalent age), I don't see how their GAMs can model what they are not trained to model. In other words, the authors cannot be sure to what extent the effects that they attribute to preterm birth (including their GNM "preterm" model) would also be observed at the same GA in utero, in babies that will then be born at term (if that data was available, which of course it isn't). If they included preterm at term scans, they could directly compare babies with the same postconceptional/PMA age, where the only difference between the two groups (term at term vs preterm at term) is GA at birth, effectively disentangling the effect of GA at birth from PMA at scan.

I also have other suggestions and/or issues that I think the authors should consider addressing:

2) Figure 1C: I am not sure the authors have appropriately described what is plotted here. Since the authors are plotting PMA at scan and GA at birth simultaneously in the x-axis, there should be two datapoints for each given density (one blue dot for GA at birth and one orange/yellow dot for PMA at scan). In other words, each density existing in the y-axis should always have at least two corresponding data points in the x-axis, as all scans (and therefore connectomes) have one age at birth and one age at scan. I struggle to see for instance the GA at birth (blue dots) corresponding to data with PMA at scan between 25 and 30 weeks PMA (with low densities, below 8%, yellow dots). These data don't seem to be taken into account when plotting GAMs model for GA at birth (with much fewer data points below 8% density).

Is the y-axis perhaps not really representing "density" but a corrected version of density given specific GA at birth / PMA at scan (and then labelling and legend are misleading)? Or are there missing datapoints? Similar effect can be seen in Figure 2 and Figure 5 for other variables, please clarify.

3) Figure 1D: Clarify what is edge weight representing (average number of streamlines?). Perhaps the authors want to consider whether streamline count is a valuable metric to use at all (see for instance the "6th sin" in <https://doi.org/10.3390/diagnostics9030115>), and consider the effect of using such an unreliable measure as input to create the binary networks that they are basing their models on.

4) L144-145 Clarify how networks are generated for analysis (1), how are networks thresholded to generate variable density? In approach (2), if original (variable) density is lower than 10% for many subjects (as shown in Fig 1C), how can be all subjects forced to have a density of 10%? The authors of course clarify this issue in Methods section lines 571-584, but perhaps it is worth introducing here the absolute thresholding using streamline count of 325 for approach (1), which is not applied to approach (2), so it is easier for the reader to interpret.

5) L150-160. I appreciate the clarification regarding the methods used to calculate tractographies, and the addition of further analyses characterising different thresholding approaches to generate different levels of sparsity. However, I still think the results presented here and in Fig1C are a bit difficult to interpret (should clarify what thresholding approach is used to define network density). Is it possible that the density presented here (variable density according to approach 1, L144?) is related to small ROIs registered to babies with PMA<32 weeks? Have the authors QCd the template/atlas registrations to PMA<32 subjects?

6) I would suggest to expand the discussion of limitations in relation to the use of Euclidean distance for GNMs and the fact that this approach won't take into account, for instance, that the distance between regions of different hemispheres (or lobes) that are next to each other is much larger than the Euclidean distance between their centroids.

(Remarks on code availability)

The code is a usable resource for the community. However, I don't see why the authors include 10 example datasets, but do not include the whole data analysed in their study, so the community can reproduce their results.

Reviewer #2

(Remarks to the Author)

The authors have addressed my comments and made appropriate revisions to the manuscript

(Remarks on code availability)

Reviewer #3

(Remarks to the Author)

I thank the authors, who have addressed some of my concerns. However, there remain many that are still unaddressed.

1. For instance, I initially made a comment that “edge-level matches [between the observed and simulated connectomes]” are not evident. The authors rightly noted that this isn’t their claim – their claim is that they match topological properties and distributions.

This is fair, but the authors are missing my point (or perhaps I didn’t explain it clearly enough). High-level network measures, or even distributions of local measures, do not uniquely specify a network. Most combinations of metrics can be satisfied by a plurality of networks. I don’t believe that this is a contentious statement.

However, (structural) brain networks are highly specific. They have a fixed set of attributes, but they arrive at these attributes through a highly conserved architecture. The variability across brains in presence/absence/weights of links is remarkably low considering the space of all networks with similar high-level properties.

I am quite confident that the synthetic networks generated by the models, though they match high-level/distributional properties of the empirical neonatal connectomes, arrive at those statistics with a wildly different configuration of edges. That is, the modeled networks only resemble the empirical networks at a coarse, summary level. Though I’m sure the discrepancy isn’t *quite* this bad, this is roughly equivalent to saying a unimodal Gaussian distribution with mean of 0 has the same mean as a bimodal distribution whose means are equally spaced on both sides of 0.

So my claim is that without matching at an edge level, the models are simply not that informative. The authors referenced Figure 4F. This figure does not inspire confidence that the models are capturing local features. There are clear mismatches (the “hubs” aren’t even in the correct locations).

2. I also raised concerns over the apparent doubling of network density. The authors countered by referring to the Ball et al 2014 paper (density was not significantly different between pre-/post-term cohorts, Song et al 2017 (the authors don’t report density, but show a large increase in strength – however, without the data it’s not easy to directly determine whether this change was driven by density or edge weights), and van den Heuvel et al (2015), which focused mostly on group-averaged matrices but also doesn’t report differences in density. In that paper, however, clustering increases and path length decreases from 30-40w, consistent with an increase in density. However, the exact change is not known given what was reported in the paper. So, as far as I can tell and unless there are other references the authors can share, it is not clear that the doubling of density is unrelated to data quality or other issues that might obscure true connections.

I also found the authors response troubling:

“given our stringent thresholding it is unlikely that the increase in density is representing brand new tracts in every case, but more likely that these tracts have become prominent enough to pass the threshold”

This suggests that whether a tract is included or not is more of a statistical question than an empirical one.

3. I also raised concerns about density driving some of the observed differences in topology. The authors responded by thresholding the matrices at 10% and carrying out their analyses on the sparsified networks. I have one follow-up question and one concern. The follow-up question is this: why 10%? The authors need to show that their effects hold over a reasonable range of thresholds. There is nothing special about the 10% value.

This analysis also raises a concern. The authors note that after controlling for density, some of effects go away or even reverse their sign (increases are now decreases). Isn’t this a problem? How can both be true? There are two views: the thresholding procedure removes noisy connections and controls for the likelihood that reported effects are purely the consequence of density. So maybe the thresholded results are closer to the truth. On the other hand, as the authors note, thresholding can be merciless and throws away meaningful individual variation. In which case, maybe the original (unthresholded) results were closer to reality. But it can’t be both. This inconsistency needs to be addressed by the authors.

4. I also asked about rich clubs, noting that brain/cranial volume surely changes (increases) during this period, so that any two points on the brain surface likely grow further apart. If the rich-club stays even remotely consistent (comprised of the same set of edges) then the expectation is that the lengths of those connections will also increase. The authors responded by saying “the distance between the parcels stays the same” (in essence). This can’t be right unless all brains were matched to some shared reference brain. If this is the case, it seems problematic in the context of generative modeling. A key component of the two-term models is the Euclidean distance matrix. If distances change, this will impact the model parameters. For example, larger/smaller brains (greater/reduced interregional distances) might need different distance penalties—the γ parameter—to match properties of the empirical network. From what the authors report, it’s not clear that this is the case.

As a side note, I examined the github repository. I have some concerns about the data itself. One of the hallmarks of structural connectomes is their distance dependence and preference for short-range connections. One way this is evident is if one were to compare the mean distance between connected nodes against the mean distance of unconnected nodes. Because connectomes make more short-range connections, the mean distance for connected nodes is always smaller than unconnected. However, when I run the script “A_run_initial_generative_models.m” to obtain the variables “edistance”

(Euclidean distance between pairs of nodes) and "example_consensus_network" and then make a boxplot of distances for connected and unconnected nodes, the distributions are not different. Maybe this is something that is true for neonatal connectomes (lack of distance dependence) or this sample is not representative in some way (or I misunderstood the README file). But this is odd to me.

In summary, I don't think this is a bad paper—I quite like many of the results. But under scrutiny, there are too many concerns related to the underlying data, processing decisions, and interpretations of the results for me to endorse its publication.

(Remarks on code availability)

Version 2:

Reviewer comments:

Reviewer #1

(Remarks to the Author)

The authors have now addressed most of my previous concerns, or at least, edited the manuscript so the limitations of their work are clear.

(Remarks on code availability)

The authors do not provide the derivative data (brain networks / adjacency matrices) used in this manuscript, so it would be very difficult to reproduce the results presented in their work.

In their response to my previous comment on data availability they mention the data is "is not theirs to release". I do not understand why would this be the case, i.e., why wouldn't they be able to share the adjacency matrices (brain networks) they calculated, and who else (if not the authors) has the right to grant access to these data.

Reviewer #3

(Remarks to the Author)

I appreciate the candid and transparent response from the authors. They are correct -- we do not see eye to eye on every matter. They have much more faith in the generative models (their ability to reveal something novel and meaningful that is not captured using other methods, e.g. graph theory) than I do.

That said, the transparency is much appreciated, and offers some balance to their submission.

I don't wish to use the review process as a negotiating table, but here's what I would suggest in the next round of revisions. At this point the authors have 1) addressed my technical concerns; I have no serious remaining technical issues; 2) they have added *some* balance to their article.

I strongly suggest being more explicit and focused in how they include this balance. Presently, their responses to my philosophical concerns related to generative models appear throughout the manuscript; they are scattered and attached to specific results/remarks.

I strongly believe that it would be beneficial to readers, and especially those unfamiliar with generative models who might not be aware of their limitations, to include a sub-section in the discussion (with a title) that includes a similarly transparent and candid discussion of the conversation that's transpired in the review process. Should we believe a 2-fold increase in network density -- are we *really* confident that this reflects a doubling of myelinated white matter? How should we interpret the results of a fit generative model? Certainly not mechanistically as a model of growth/development. If not this, then is it fair to think of the models as compressive? Perhaps, but only in the sense that the models generate networks with similar features at the level of cumulative distributions -- despite claims made here and by others using generative models, they are quite poor at faithfully capturing the precise configuration of edges white-matter networks and the topography of local network statistics (yes, I'm aware local statistics in observed and synthetic networks are often correlations, but the correlations are not strong).

Should the authors agree to this (and I would love to read the newly written paragraph, should another round of review be granted) I think it would put the article within the "recommend for publication" circle, in my opinion.

Again, I don't wish to use peer review as a negotiating table, but this is the clearest way I know to communicate the gap between the manuscript in its current form and what I still believe needs to be addressed to make it acceptable for publication.

(Remarks on code availability)

Version 3:

Reviewer comments:

Reviewer #3

(Remarks to the Author)

The authors have addressed all of my concerns. Glad to endorse this for publication.

(Remarks on code availability)

It would be useful to share as much of the data as possible; but the code itself seems reasonable.

Dear Reviewers,

Premature birth changes wiring constraints in neonatal structural brain networks: NCOMMS-23-25330

We want to thank you for your time and valuable feedback on our manuscript. We especially appreciate the encouragement and enthusiasm that you have expressed for our project. We have taken the past few weeks to undertake additional analyses and edits that we hope will address each of the insightful comments. We believe our manuscript is stronger with these revisions and have highlighted below the major changes:

Major additional analyses include:

- Analysis of multiple density networks ranging from 24-8% density.
- A completely density-controlled analysis in which each neonate has a 10% density network for clearer topological comparison.
- Additional tractographies with multiple minimum fiber length (5-30mm).
- A term-equivalent age matched analysis including only neonates from the sample that were scanned at 37 weeks PMA or later.

We have also made revisions in:

- Provided more detail on the reasoning behind thresholding procedures.
- Further clarification of the generative network modeling methods and interpretability.
- Expanded limitations section to include neurobiological interpretation and factors preceding premature birth.
- Updated term-group representative parameters for developing GNMs analysis.

We believe these changes have led to clear improvements in the current version and we hope the revisions meet your high standards. Please find below the point-by-point response to each comment and we welcome any further constructive feedback. For ease of indexing, you can find the responses to each reviewers' comment at the following pages of this response letter:

Reviewer 1 – response from Page 2 - 21

Reviewer 2 – response from Page 22 - 25

Reviewer 3 – response from Pages 26 - 32

Sincerely,

Alexa Mousley
PhD Candidate,
MRC Cognition and Brain Sciences Unit,
University of Cambridge

Response to Reviewer #1

In this manuscript Mousley and colleagues analyse a publicly available data of neonatal MRI (the Developing Human Connectome Project, dHCP). They perform a “classic” graph theory characterisation of the neonatal brain, using diffusion MRI tractography from the dHCP, preprocessed by Dr Fang-Cheng Yeh (not an author of this manuscript), and then they fit generative network models to their data, assessing optimal model parameter fits that differ between term and preterm infants.

This is a timely and interesting study, making the most of the new resources generated for the neonatal brain, and publicly available data. However, I also think the article has some flaws; I am unsure about some of the methodological decisions taken, which I think require further justification. In particular, I have some concerns about how some elements of the first part of the manuscript (graph theory analysis) are performed, and I think more details should be provided to reassure readers that structural connectivity networks for very preterm babies included in the analysis are anatomically valid. I think the generative models approach used in the second part of the manuscript is novel and interesting, providing a useful way of interpreting results and going beyond just superficial graph theory analysis.

Please find below a few suggestions and constructive criticisms to some of your methodological and study design choices. I hope these may help you improve the manuscript.

We are incredibly grateful to the reviewer for the time and effort they have put into this review. We found it very constructive and helpful in substantively improving the paper. Further clarification on methodological decisions and the validity of tractography is fundamental to the project and we have provided multiple analyses to explore this further.

Main comments

1) Networks obtained are strongly dependant on the tractography algorithm used and the arbitrary choices made (e.g., 60% consensus thresholding, minimum 30mm fibre length, minimum 325 streamlines). I am uncertain of very low densities obtained overall (less than 12% for all subjects), and in particular for subjects scanned at less than 32 weeks postconceptional age. I would suggest the authors consider an alternative tractography algorithm that is more flexible to the peculiarities of the neonatal brain, such as mrtrix’s MSMT-CSD, which has been used by previous work using this data (see for instance <https://doi.org/10.1016/j.neuroimage.2018.10.060> and <https://doi.org/10.1073/pnas.2023598118>). Tractographies shown in Figure 3A are a bit concerning. Do they correspond to neonatal brain WM anatomy? Apparently just a few central streamlines have been reconstructed.

We agree, this is an important point. Crucially, the low densities do not stem from the tractography, but from the subsequent processing steps. The sparsity is necessitated by the modeling; all known generative modeling studies use networks of 10% density or less (Akarca et al., 2021; Betzel et al., 2016; Carozza et al., 2022) or only model

one hemisphere (Arnatkevičiūtė et al., 2021; Oldham et al., 2022; Vértes et al., 2012). This is because the computational power needed to run the simulations scales linearly with connection number. Practically speaking, without sparse connectomes, we cannot do the computational modeling given that we ran 90,000 simulations for each of the 13 different value metrics and an additional 10,000 simulations per participant.

We agree entirely with the reviewer, before we introduce the modeling we need to show readers the impact of this sparsity (and indeed the decisions that impose it) on the underlying networks. In the revised manuscript, we now include a number of additional analyses with networks of differing densities, created using different decisions, to test whether the sparse networks needed for the modeling are still a fair reflection of the underlying connectivity.

i) Comparing networks across differing fiber lengths

Our first set of new analyses deployed three new tractographies, each with a different minimum fiber length of 20mm, 10mm, and 5mm. Despite different fiber lengths, our networks display similar density areas (e.g., cingulum and parietal-occipital connections) and very similar adjacency matrices (Supplementary Fig. 11C). Indeed, Spearman rank order correlations reveals that regardless of fiber length during tractography, the ordering of regions is highly similar (5mm to 10mm $r_s = 0.75$; 10mm to 20mm $r_s = 0.74$; 20mm to 30mm $r_s = 0.74$). As shown in Supplementary Figure 11A, there is a non-linear relationship between fiber length and density – actually the 30mm fiber length used in our manuscript produces denser networks than, say, a 5mm fibre length. Only a 10mm fiber length would produce a denser network. Crucially, the local topological arrangement of the networks – the thing we subsequently model – is almost identical regardless of the fiber length. This can be seen in the topological ‘fingerprints’ in the new Supplementary Figure 11B. Finally, we directly compared the tractography before and after 32 weeks PMA with our original 30mm tractography (Supplementary Figure 11D), which shows that despite differences in density, the matrices are highly similar ($r_s = 0.98$). This gives us confidence that these very early PMA scans have not been constructed in a manner that is somehow fundamentally different than expected (e.g., missing connections in a particular area).

Despite these additional analyses, it remains challenging to determine if differences we observe in early PMA tractography are truly a reflection of developmental timepoint or simply a product of tractography differences. Therefore, we have also provided a term-equivalent age analysis in which all scans taken earlier than 37 weeks PMA have been excluded. The results of this new analysis are highly consistent with the original (Supplementary Fig. 3). This additional analysis supports our original interpretation that term infants have more dense and efficient networks with more modular structure compared to preterm infants. The one effect that was significant in our original analysis but not in the term-equivalent age analysis was the change in the number of rich clubs across GA at birth. However, this is still supportive of our original interpretation as we concluded a *relative* preservation of rich club connections, whereas this analysis suggests a complete preservation of rich club connections. Therefore, even if tractography issues are present for infants scanned at early PMA, this additional analysis confirms that it has not affected our interpretation of topological patterns.

In the revised manuscript we have included these analyses and figures:

Results section 156-158: “This result was replicated in a term-equivalent age analysis in which only infants scanned at or above 37 weeks PMA were included (Supplementary Fig. 3A; $F_{density,GA\ at\ birth} = 2.71$, estimated $df = 3.02$, $p = 0.026$).”

Results section lines 175-176: “This result was replicated in the term-equivalent age analysis (Supplementary Fig. 3B; $F_{modularity,GA\ at\ birth} = 9.42$, estimated $df = 1.00$, $p = 2.25 \times 10^{-3}$).”

Results section lines 192-193: “This result was also replicated in the term-equivalent age analysis (Supplementary Fig. 3C; $F_{global\ efficiency,GA\ at\ birth} = 2.71$, estimated $df = 3.60$, $p = 0.029$).”

Results section lines 206-208: “For the term-equivalent age analysis, however, no significant relationship was found between the number of rich club connections and GA at birth (Supplementary Fig. 3D; $p = 0.412$).”

Results section lines 218-224: “These results suggest that early PMA rich club formation consists of longer-range connections compared to later PMAs. In addition, the average length of rich club connections significantly increased across GA at birth in both variable and density-controlled networks (Fig. 2B; variable: $F_{rich\ lengths,GA\ at\ birth} = 5.37$, estimated $df = 3.82$, $p = 1.06 \times 10^{-4}$; density-controlled: $F_{rich\ lengths,GA\ at\ birth} = 3.45$, estimated $df = 1.37$, $p = 0.029$) as well as in the term-equivalent age analysis (Supplementary Fig. 3E; $F_{rich\ connection\ length,GA\ at\ birth} = 8.10$, estimated $df = 2.81$, $p = 2.34 \times 10^{-5}$).”

Methods section lines 531-537: “A secondary analysis was completed with a subsample of term-equivalent age neonates ($N = 487$, preterm: $n = 40$, term: $n = 447$). This secondary analysis serves to validate the statistical controlling of PMA and GA at birth (see “Statistics” below for detail) as well as to address realistic concerns about the tractography quality of early PMA scans. Average network matrices of early PMA networks compared to late PMA networks can be found in the supplementary materials (Supplement Fig. 11D).”

Methods section lines 560-569: “As a minimum fiber length of 30mm could be too restrictive for neonates, an additional analysis was added to explore the effects of minimum fiber lengths 5mm, 10mm, 20mm and 30mm during tractography. This analysis demonstrated that the 30mm tracked networks were denser than 5mm and 20mm – indicating non-linear relationship between minimum fiber length in tractography and density (Supplementary Fig. 11A). Crucially for the present study, the local topological arrangement of the networks is similar, indicated by topological fingerprints (Supplementary Fig. 11B). In addition, Spearman rank order correlations reveals that the order of connectivity of regions is highly similar (5mm to 10mm $r_s = 0.75$; 10mm to 20mm $r_s = 0.74$; 20mm to 30mm $r_s = 0.74$). A full organizational analysis, connectivity matrices and topological fingerprints can be found in the supplementary materials (Supplementary Fig. 11).”

Supplementary Figure 3. Term-equivalent age topological analysis. (A) Density significantly increases, (B) modularity significantly decreases, and (C) global efficiency increases across GA at birth. (D) There is no significant change in the number of rich club connections, however (E) the average length of rich club connections significantly increases across GA at birth. The shaded area indicates 95% confidence intervals. The grey dotted line indicates the cut-off for term birth (37 weeks GA or later is term-born). *** indicates $p < 0.001$, ** indicates $p < 0.01$, * indicates $p < 0.05$.

Supplementary Figure 11. Exploration of the effects of minimum fiber lengths of tractography. (A) Density (5mm: $F_{density,PMA} = 11.77$, estimated $df = 8.43$, $p = 2.00 \times 10^{-16}$, $F_{density,GA\ at\ birth} = 7.41$, estimated $df = 3.29$, $p = 6.18 \times 10^{-6}$; 10mm: $F_{density,PMA} = 39.71$, estimated $df = 7.71$, $p = 2.00 \times 10^{-16}$, $F_{density,GA\ at\ birth} = 4.23$, estimated $df = 1.00$, $p = 0.040$; 20mm: $F_{density,PMA} = 15.98$, estimated $df = 10.85$, $p = 2.00 \times 10^{-16}$, $F_{density,GA\ at\ birth} = 6.22$, estimated $df = 2.80$, $p = 1.55 \times 10^{-4}$; 30mm: $F_{density,PMA} = 37.17$, estimated $df = 7.50$, $p = 2.00 \times 10^{-16}$, $GA\ at\ birth: p = 0.166$), global efficiency (5mm: $F_{global\ efficiency,PMA} = 10.89$, estimated $df = 8.69$, $p = 2.00 \times 10^{-16}$, $F_{global\ efficiency,GA\ at\ birth} = 7.06$, estimated $df = 2.93$, $p = 3.34 \times 10^{-5}$; 10mm: $F_{global\ efficiency,PMA} = 37.84$, estimated $df = 6.87$, $p = 2.00 \times 10^{-16}$, $F_{global\ efficiency,GA\ at\ birth} = 6.25$, estimated $df = 1.00$, $p = 0.013$; 20mm: $F_{global\ efficiency,PMA} = 12.10$, estimated $df = 10.16$, $p = 2.00 \times 10^{-16}$, $F_{global\ efficiency,GA\ at\ birth} = 6.56$, estimated $df = 2.72$, $p = 1.08 \times 10^{-4}$; 30mm: $F_{global\ efficiency,PMA} = 35.29$, estimated $df = 7.23$, $p = 2.00 \times 10^{-16}$, $F_{global\ efficiency,GA\ at\ birth} = 4.21$, estimated $df = 1.00$, $p = 0.041$), and modularity (5mm: $F_{modularity,PMA} = 6.99$, estimated $df = 3.51$, $p = 8.83 \times 10^{-6}$, $F_{modularity,GA\ at\ birth} = 19.98$, estimated $df = 1.00$, $p = 9.53 \times 10^{-6}$; 10mm: $F_{modularity,PMA} = 7.15$, estimated $df = 4.19$, $p = 1.43 \times 10^{-6}$, $F_{modularity,GA\ at\ birth} = 9.18$, estimated $df = 1.00$, $p = 2.53 \times 10^{-3}$; 20mm: $PMA\ p = 0.900$, $F_{modularity,GA\ at\ birth} = 10.32$, estimated $df = 1.86$, $p = 2.19 \times 10^{-5}$; 30mm: $F_{modularity,PMA} = 6.85$, estimated $df = 2.84$, $p = 7.64 \times 10^{-5}$, $F_{modularity,GA\ at\ birth} = 4.38$, estimated $df = 2.80$, $p = 3.46 \times 10^{-3}$) across GA at birth and PMA. (B) Topological fingerprints and (C) consensus networks for each tractography method. (D) Average binarized matrices of early and late PMA neonatal scans for 30mm fiber length connectomes.

ii) *Comparing networks across different thresholds*

Given the asymmetrical nature of neonatal brains (Batalle et al., 2018; Ratnarajah et al., 2013), we chose a thresholding procedure to produce sparse networks, thereby allowing us to subsequently model both hemispheres simultaneously, rather than model just one hemisphere. In a new analysis we checked whether making networks sparser changes, or preserves, the overall set of relationships between the topological characteristics.

We constructed consensus networks at progressive densities from 24% to 8% density. In the analysis, modularity and characteristic path length decreases while global efficiency increases across densities (Supplementary Fig. 2A-C). We also included an analysis of what lobes the additional connections are located for increasingly dense connectomes (Supplementary Fig. 2D). Selecting the 8% density (sparsest), 18% density connectome, and the 24% connectome (densest), we also display the topological fingerprints (Supplementary Fig. 2E) and connectivity matrices with region labels (Supplementary Fig. 2F). Additional connections present in denser connectomes (the upper half of the matrices) are evenly distributed across the connectome (Supplementary Fig. 2F). Finally, and crucially, regardless of the overall density of the networks, the underlying arrangement of local topological characteristics is unchanged.

In the revised manuscript we have included these manuscript changes and additional figure:

Results section lines 140-148: “Creating GNMs is only possible with sparse connectomes^{22,38,39,41}. This requires that we threshold our connectomes (see “Methods”; “Connection construction”), so we first explored how these sparse networks change according to PMA and GA at birth. To disentangle topological structure from network density, we conducted two analyses: (1) a variable density analysis in which the sample has *an average* density of 10% and thus represents the variability in density observed, and (2) a density-controlled analysis where each network is constrained to exactly 10% density. Finally, because the modeling requires sparse networks, we also explored the impact of that sparsity on topology across networks with 8-24% density (Supplementary Fig. 2).”

Methods section lines 571-583: “For thresholding, we employed both consensus and absolute approaches. To reduce the rate of false positive and negative detection in tractography, a 60% consensus threshold was initially applied⁶⁸. This thresholding results in streamlines between regions that are present in 60% of the sample being retained. Following this threshold, we conducted two density-based analyses. First, following previous generative network modeling methods, we performed a 325 binarization threshold to yield an average network density of 10% across the sample^{22,37}. To explore the effects of density on topology, we complete a secondary density-controlled analysis where each individual had a *proportion* binarization threshold that yielded an exactly 10% dense network. By applying these two thresholding techniques, we can complete a standard GNM analysis with varied density, followed by an exploration of how controlling for density effects topological interpretations. An additional analysis exploring how progressively more sparse

networks, ranging from 24%-8% density, relate to changes in topology, topological fingerprints and representative connectivity matrices can be found in the supplementary materials (Supplementary Fig. 2).”

Supplementary Figure 2. Multiple density consensus networks. (A) Modularity decreases, **(B)** Global efficiency increases, and **(C)** characteristic path length decrease with increasingly dense connectomes. **(D)** Comparisons of consecutive connectomes, indicating the distribution of new connections across 7 lobes (indicated as a % change from the sparser to dense network). **(E)** Topological fingerprints demonstrating the correlation between local organizational measures for the 8% (sparsest), 18% (middle) and 24% (densest) density networks. **(F)** Connectome matrices representing the 8%, 18% and 24% networks. Here, the lower triangle is the matrix and for 18% and 24% networks the upper triangle is only comprised of connections that are present in that network that weren't present in the previous network (8-18% and 18-24% comparisons).

iii) *Do the tractographies in Figure 3 actually correspond to neonatal anatomy?* Figure 3 is solely a schematic for explaining the generative network modeling theory and procedure. Panel A was constructed to illustrate the concept of increasing connectivity – it is not reflective of tractography used in the analysis. We recognize how this was confusing and apologize. In the revised manuscript, we have removed the ‘Week’ labels, which could be misleading, and replaced this with ‘Generative

Network Modeling'. We also further clarified in the figure description that this is merely illustrative.

Figure 3 caption lines 305-307: “(A) Diagram depicting how models form over time by adding connections in the network, based on a trade-off between parameterized cost ($d_{i,j}$) and value ($k_{i,j}$), to mimic network growth.”

In summary, in the revised manuscript we have included a number of new analyses that formally test for the impact of tractography choices and sparsity on the resulting networks. We have made major changes to the manuscript to clarify why sparsity was the goal and clarifying what the impact of sparsity is on the resulting networks.

2) In the data presented by the authors, there is a strong link between network density and age at birth and age at scan. Since network density is, by definition, massively driving measures of integration and segregation, it is unclear whether measures presented in Figure 2 are adding much.

We agree, network density will be intrinsically related to other network measures. Therefore, we have included a density-controlled analysis of the graph measures in which each participant's connectome is restricted to a density of exactly 10%. Whilst this will likely remove important individual differences, it does allow us to make direct topological comparisons between participants, unaffected by differences in density. Many measures of segregation and integration are indeed related to density; the relationships between some measures and PMA, or GA at birth, change when controlling for density, as we would expect. For instance, the networks of those born preterm have no significant difference in efficiency than infants born at term, when you control for overall density (Fig. 2A) However, certain key topological properties are unaffected by density, for instance the impact of PMA and GA at birth on modularity (Fig. 2). We have incorporated this with the revised Methods and Results and have updated Figure 2 to include the density-controlled analyses.

Results section lines 143-148: “To disentangle topological structure from network density, we conducted two analyses: (1) a variable density analysis in which the sample has an average density of 10% and thus represents the variability in density observed, and (2) a density-controlled analysis where each network is constrained to exactly 10% density. Finally, because the modeling requires sparse networks, we also explored the impact of that sparsity on topology across networks with 8-24% density (Supplementary Fig. 2).”

Results section lines 168-178: “However, the density-controlled analysis revealed that modularity significantly increased across PMA, particularly rapidly before about 34 PMA (Fig. 2A; $F_{modularity,PMA} = 8.15$, estimated $df = 5.35$, $p = 2.00 \times 10^{-16}$). This suggests that while the overall increase in density is inherently related to a less modular network, the type of connections that are forming at older PMAs are driving an increase in modular structure. In contrast, both variable and density-controlled analyses demonstrated that modularity significantly decreases across GA at birth (Fig. 2A; variable: $F_{modularity,GA\ at\ birth} = 7.96$, estimated $df = 3.08$, $p = 6.74 \times 10^{-6}$; controlled: $F_{modularity,GA\ at\ birth} = 8.93$, estimated $df = 1.00$, $p = 2.90 \times 10^{-4}$). This result was replicated in the term-equivalent age analysis (Supplementary Fig. 3B; $F_{modularity,GA\ at\ birth} = 9.42$,

estimated $df = 1.00$, $p = 2.25 \times 10^{-3}$). These findings suggest that in addition to less dense, more modular networks overall, infants born at early GA have a fundamentally more modular topology.”

Results section lines 184-197: “However, when density is controlled, global efficiency significantly decreases while characteristic path length increases (Fig 2A; $F_{global\ efficiency,PMA} = 10.99$, estimated $df = 3.91$, $p = 2.00 \times 10^{-16}$; $F_{characteristic\ path\ length,PMA} = 10.17$, estimated $df = 3.66$, $p = 2.00 \times 10^{-16}$). Together, these results suggest that increasing network efficiency across PMA can be explained by increases in density – when density is controlled for, the topological structure of networks becomes *less* efficient across PMA. Similar to PMA, with variable density networks global efficiency significantly increased while characteristic path length decreased across GA at birth (Fig. 2A; $F_{global\ efficiency,GA\ at\ birth} = 6.12$, estimated $df = 3.66$, $p = 3.82 \times 10^{-5}$; $F_{characteristic\ path\ length,GA\ at\ birth} = 5.69$, estimated $df = 3.16$, $p = 1.79 \times 10^{-4}$). This result was also replicated in the term-equivalent age analysis (Supplementary Fig. 3C; $F_{global\ efficiency,GA\ at\ birth} = 2.71$, estimated $df = 3.60$, $p = 0.029$). When controlling for density, however, no significant relationships were found between GA at birth and global efficiency (Fig. 2A; $p = 0.462$) or characteristic path length ($p = 0.918$). In other words, the less efficient networks for infants born at early GA can primarily be explained by less dense networks, not a fundamental change in underlying topology.”

Results section lines 208-213: “Indeed, the density-controlled network analysis demonstrated that while the total number of rich club connections still significantly increased across PMA (Fig. 2B; $F_{rich\ connections,PMA} = 4.05$, estimated $df = 3.42$, $p = 2.31 \times 10^{-3}$), no significant effect was found across GA at birth (Fig. 2B; $p = 0.059$). Put simply, the increase in rich club connections across PMA occurred regardless of changes in density, but differences in rich club connections across GA at birth were density dependent.”

Results section lines 215-226: “Across PMA, average length of rich club connections significantly decreased in both variable and density-controlled networks (Fig. 2B; variable: $F_{rich\ lengths,PMA} = 11.92$, estimated $df = 3.04$, $p = 2.00 \times 10^{-16}$; density-controlled: $F_{rich\ lengths,PMA} = 15.58$, estimated $df = 3.12$, $p = 2.00 \times 10^{-16}$). These results suggest that early PMA rich club formation consists of longer-range connections compared to later PMAs. In addition, the average length of rich club connections significantly increased across GA at birth in both variable and density-controlled networks (Fig. 2B; variable: $F_{rich\ lengths,GA\ at\ birth} = 5.37$, estimated $df = 3.82$, $p = 1.06 \times 10^{-4}$; density-controlled: $F_{rich\ lengths,GA\ at\ birth} = 3.45$, estimated $df = 1.37$, $p = 0.029$) as well as in the term-equivalent age analysis (Supplementary Fig. 3E; $F_{rich\ connection\ length,GA\ at\ birth} = 8.10$, estimated $df = 2.81$, $p = 2.34 \times 10^{-5}$). Contrasting that of PMA, these results indicate that early GA at birth is related to shorter rich club connections than infants born at later GA, unrelated to total connectivity differences.”

Results section lines 248-252: “This process fosters increased integration and efficiency across PMA which masks the fundamental topological trends towards less integrated and efficient networks. Changes across GA at birth are more nuanced, rapid increases in density relates to increasing integration, though

when density is controlled, we observed decreasing modularity but no differences in efficiency.”

Results section lines 353-355: “The density-controlled analysis indicates a significant improvement of model fit across PMA but no significant difference in model fit across GA at birth (Supplementary Fig. 10).”

Results section lines 382: “These results were all replicated in our density-controlled analysis (Supplementary Fig. 10D).”

Discussion section lines 439-440: “Secondly, previous work has not included density-controlled analyses, making it hard to isolate topological modularity during this early developmental period.”

Discussion section lines 452-458: “Crucially, this increase was primarily explained by increases in density across GA at birth, but not so across PMA. Put simply, preterm infants do not have inherently fewer rich club connections compared to term infants with density-controlled networks. This aligns with the theory that premature infants have conserved connectivity in core connections but reduced connectivity in peripheral regions^{21,27}. However, our results also demonstrate that regardless of density, while the length of rich club connections significantly decreases across PMA, the opposite effect is found across GA at birth.”

Methods section lines 576-580: “To explore the effects of density on topology, we complete a secondary density-controlled analysis where each individual had a *proportion* binarization threshold that yielded an exactly 10% dense network. By applying these two thresholding techniques, we can complete a standard GNM analysis with varied density, followed by an exploration of how controlling for density effects topological interpretations.”

Figure 2. Organizational differences across PMA and GA at birth. (A) Modularity significantly decreased across PMA and GA at birth in variable density networks, but modularity significantly increased across PMA and decreased across GA at birth in controlled-density networks. Global efficiency significantly increased across both PMA and GA at birth in variable density networks. However, with density-controlled networks global efficiency significantly decreased across PMA but no significant effect was found across GA at birth. (B) The number of rich club connections significantly increased across PMA in both variable density and density-controlled networks. The number of rich club connections also significantly increased across GA at birth in variable density networks, but no significant effect was found with density-controlled networks. In addition, average length of rich club connections (Euclidean distance) significantly decreased across PMA in both variable density and density-controlled networks. The average length of rich club connections significantly increased across GA at birth in both variable density and density-controlled networks. The shaded area indicates 95% confidence intervals. The grey dotted line indicates the cut-off for term birth (37 weeks GA or later is term-born). *** indicates $p < 0.001$, ** indicates $p < 0.01$, * indicates $p < 0.05$.

3) Postconceptional age at scan and age at birth are strongly related in these data, but this relationship is particularly true for subjects scanned at <37 weeks postconceptional age, making very difficult to disentangle one factor from the other.

We appreciate this comment and agree that it is not possible to disentangle completely premature birth from < 37 weeks PMA. However, we believe that the right statistical approach to doing this is with generalized additive models (GAMs). By statistically controlling for one measure while exploring the other, these types of models give us the best chance of testing how networks are related to each metric. We have clarified this for the reader:

Methods section lines 681-683: “PMA, sex, head circumference, translation, and rotation were controlled for in GAMs on observed data when exploring GA at birth. When looking at PMA, the same variables were controlled for, in addition to GA at birth.”

In addition to statistically controlling for PMA, we have added a term-equivalent age analysis (please see Main Comments, response 1i).

4) At >37 weeks postconceptional age the authors have a mixture of term and preterm born babies. Any conclusion about brain maturation (postconceptional age at scan) will be confounded by the effect of preterm birth. I think the authors should be very careful with their statistical design in order to address these issues. For points 3 and 4, perhaps they can address specific hypotheses by using the appropriate sample, for instance, typical development with term subjects only; effect of preterm birth with term and preterm babies scanned at term equivalent age (so they are comparable in terms of postconceptional age at scan); and early network development (<37 weeks) with babies scanned <37 weeks only.

This is an important point, and a great suggested solution. As noted above, the GAMs are designed to disentangle PMA at scan from GA at birth, but we have also now included an additional analysis that includes only infants that were scanned at a term-equivalent age (please see Main Comments, response 1i).

Detailed Comments:

L118-119: Is the sample size corresponding to the number of participants or the number of sessions included? Note that most preterm babies were scanned twice as part of the dHCP. In the methods section the authors mention they are excluding repeated scans, but it is unclear what sessions were excluded (i.e., preterm scanned soon after birth, or preterm at term equivalent age)? Why exclude sessions, instead of include all subjects and perform sub-analyses? Or use LME to deal with repeated measures.

The sample size refers to both the number of participants and the number of sessions included, because we have selected a cross-sectional sample. Due to the data utilized in this study being accessed from DSI Studio (https://brain.labsolver.org/hcp_d2.html), we had access to 724 scans, 82 of which were identified as repeat scans. As the main

purpose of this project was to perform generative network modeling, we chose to select a cross-sectional sample to avoid potential confounding factors in the simulation analysis. We clarified this choice in the revised Results and Methods sections:

Results section lines 119-120: “Our cross-sectional sample consisted of neonates ($N = 630$) scanned at 26 to 45 weeks PMA ($n = 183$ preterm, $n = 447$ term; see “Methods”; “Participants” for detail; Supplementary Fig. 1).”

Methods section lines 528-529: “To obtain a cross-sectional sample, we only included the first scan for each neonate.”

L135-136: “This is to be expected; children born premature have to wait longer before they can be scanned.” -> I find this is a strange statement, what do the authors mean by “wait”? Preterm babies in the dHCP are scanned twice, soon after birth and at term equivalent age. Preterm babies can be scanned in a larger range due to being ex utero for longer. Due to dHCP design (scanning up to 44 weeks PMA), preterm babies can be scanned between 24-44 weeks PMA, while term can only be scanned between 37-44 weeks PMA, causing a negative correlation between age at birth and age at scan. This may also be driven by their exclusion criteria.

We thank the reviewer for pointing out of this confusing statement. We have clarified that the more weeks between birth and scan is simply due to the longer time preterm infants are ex utero before 44 PMA. Also important to note, as mentioned in the immediately previous response, we are just using the first available scan for each participant.

Results section lines 136-138: “As scanning was completed at a maximum from 44 PMA, infants born at term must be scanned within a short time frame, whereas infants born early have a larger age range in which scanning can be conducted.”

L141-150: I am unconvinced about what can actually be concluded from the authors’ assessment of network density. This is extremely dependant of tractography algorithm selected, and ad-hoc parameters chosen. Were the same number of streamlines generated for all subjects? With the same number of seeds? Where were seeds placed? Tractography is particularly tricky for subjects scanned at less than 32 weeks postconceptional age. It would be reassuring to see a couple of examples of such young infants’ tractographies, perhaps it could be added to Figure 1A.

We appreciate that tractography of young infants is tricky. For neonates scanned at or above 32 weeks PMA had an average of 1.65×10^6 streamlines ($\pm 9.77 \times 10^3$), whereas less than 32 weeks PMA neonates have a mean of 2.20×10^6 ($\pm 4.31 \times 10^4$). Tractography methods were the same across the whole sample. We used a random seed, and the maximum fiber count was set to 5000000. We have expanded upon our methodological choices as well as including multiple early infant tractographies in Figure 1B.

Methods section lines 550-554: “Following previous neonatal studies, fiber tracking was performed in DSI Studio with a turning angle of 35°, step size of 1mm, maximum fiber count of 5,000,000, minimum fiber length of 30mm, maximum fiber length of 250mm, and a random initial seed propagation direction⁴⁹. Connectivity type was count-end, indicating that streamlines will be counted as present between two regions when the streamline ends within those regions.”

In addition to clarifying our tractography algorithm and design choices, we agree it is important to account for potentially lower quality tractography of very young infant networks. Our additional term-equivalent age analysis excludes infant scans < 37 weeks PMA and is consistent with our original findings (please see Main Comments response 1i). We have also included multiple sensitivity analyses to determine the impact of decisions like fiber length and threshold (please see Main Comments response 1i and 1ii).

L138-150: It is unclear if association between density and postconceptional age / GA at birth are corrected by one another statistically. Here and elsewhere, the effect of preterm-birth may be confounding the results associated to postconceptional age, i.e., the authors are assessing the combined effect of preterm-birth and post-conceptional age, rather than postconceptional age in typically developing individuals.

We apologize that the statistical correction was unclear in this section. We have adapted this section to clarify this and added in a term-equivalent age analysis (see Main Comments, response 1i) to further delineate between GA at birth and PMA. First, we clarified our statistical procedure:

Results section lines 114-117: “We considered both age metrics’ effects on macroscopic neural connectivity by statistically controlling for PMA when exploring GA at birth and vice versa using generalized additive models (see “Methods”; “Statistics” for detail).”

Methods section lines 681-683: “PMA, sex, head circumference, translation, and rotation were controlled for in GAMs on observed data when exploring GA at birth. When looking at PMA, the same variables were controlled for, in addition to GA at birth.”

Secondly, we changed our methods and results to include the added term-equivalent age analysis: Please see response 1i for the figure as well as our in-text changes.

L138-150: In a similar note, I am not sure whether the authors can conclude much about the topology of typically developing babies when mixing term and preterm individuals. It is expected that they will find a shallow U around 36-37 weeks, since this is the age where they are starting to see term- (or nearly term) born babies. I would suggest the authors perhaps could look at associations with postconceptional age and gestational age at birth independently for term and preterm babies, and make clearer whether they are correcting for the appropriate confounders.

We agree that this needs clarifying. We have added the clarifying statements:

Results section lines 114-117: “We considered both age metrics’ effects on macroscopic neural connectivity by statistically controlling for PMA when exploring GA at birth and vice versa using generalized additive models (see “Methods”; “Statistics” for detail).”

Methods section lines 681-683: “PMA, sex, head circumference, translation, and rotation were controlled for in GAMs on observed data when exploring GA at birth. When looking at PMA, the same variables were controlled for, in addition to GA at birth.”

We have also conducted a term-equivalent age analysis in which we only include term infants and preterm infants scanned at or above 37 weeks PMA. Please see Main Comment response 1i for the figure as well as our in-text changes.

L152-202: Please clarify if assessing binary or weighted graph theory metrics.

We used binarized networks. We apologize that this was unclear and have made following changes:

Results section lines 104-105: “We explored the relationship between binarized structural brain networks and two age-related metrics.”

Results section lines 162-164: “We found global organization of the binarized networks with variable density demonstrated a shallow U curve trend for network segregation (for definitions of network measures see “Methods”; “Graph theory”).”

L152-202: Authors have previously shown massive variability in density, which, by definition, massively influence graph theory metrics of integration and segregation. I wonder whether the results shown would hold if comparing brain topology independently of differences in density, for instance with a proportional thresholding approach.

We agree and have included a density-controlled analysis of the graph theory metrics. Please see response 2 in the Main Comments.

Figure 3A: I am not sure how to interpret tractographies at 30 and 35 weeks shown here. These tractographies look a bit strange, not corresponding to perinatal WM anatomy.

Figure 3A are fake tractographies aimed to communicate the theory behind generative network modeling. We see how this is misleading and have altered the figure and caption to clarify this. Please see Main Comments response 1i.

Figure 4E: I wonder how much of global efficiency’s variance is explained just by network density, and hence, how much is added by the other topological aspects of the generative model. In L311-315 and Supp Fig 7 the authors suggest that there are no differences in model fit due to density. If I understand correctly

in methods section, it is stated that a proportional threshold at 10% is applied in order to control for differences in network density. But in Figure 1C it is clear that network density is below 10% for many subjects already, so a 10% proportional threshold won't make a difference. I would suggest to try a 5% threshold instead.

In the original analysis, there was an average of 10% density across the entire sample. This density was obtained by a consensus and binarization thresholding (please see response 1 in Main Comments for further explanation). The variation in density was not originally controlled for, but the additional density-controlled analysis suggests that global efficiency, but not modularity, is reflective of varied density in the sample (please see response 2 in Main Comments). A quick note on thresholding, because it is very relevant here: the main analysis uses a 60% *consensus* threshold. This thresholding procedure keeps streamlines between regions that have streamlines present for 60% or more of the sample. The second thresholding, binarization thresholding, converts areas that have 325 or more streamlines present into '1'. The end result is an average density across participants of 10%, but density can vary from participant to participant (as the reviewer says, being under 10% for some participants). The density-controlled analysis uses a *proportion* threshold of 10%, meaning that every participant's connectome is guaranteed to have a density of precisely 10% derived from selecting the binarization threshold specific to that individual. We apologize this was not clearer in our original manuscript and have made alterations to the methods section to make this more explicit.

Methods section lines 571-584: “For thresholding, we employed both consensus and absolute approaches. To reduce the rate of false positive and negative detection in tractography, a 60% consensus threshold was initially applied⁶⁸. This thresholding results in streamlines between regions that are present in 60% of the sample being retained. Following this threshold, we conducted two density-based analyses. First, following previous generative network modeling methods, we performed a 325 binarization threshold to yield an average network density of 10% across the sample^{22,39}. To explore the effects of density on topology, we complete a secondary density-controlled analysis where each individual had a *proportion* binarization threshold that yielded an exactly 10% dense network. By applying these two thresholding techniques, we can complete a standard GNM analysis with varied density, followed by an exploration of how controlling for density effects topological interpretations. An additional analysis exploring how progressively more sparse networks by increasing the absolute threshold, resulting in networks ranging from 24%-8% density. The analysis explores how density relates to changes in topology, topological fingerprints and representative connectivity matrices can be found in the supplementary materials (Supplementary Fig. 2).”

Perhaps it would be good if it could be clarified that binary networks are being modelled (so readers can interpret the results without having to read methods section in detail). Also, please clarify how are long-/short-range connection defined; and clarify how distance is calculated, e.g., Euclidian distance between ROIs, or streamline (WM) distance between ROIs? If the former, how do you take into account that the distance between regions of different hemispheres (or

lobes) that are next to each other is much larger than the Euclidean distance between their centroids?

We agree. In the revised manuscript, we have clarified early in the results that we are using binary networks and that we use Euclidean distance. Within the modeling framework, it is indeed interesting to consider which method of distance we should use. Generally speaking, fiber length and Euclidean distance are correlated – as strongly as $r = 0.773$ (Akarca et al., 2023) – but the problem of using fiber length as the distance is that it heavily constrains the modeling. We cannot have a fiber length between nodes that are unconnected. The modeling requires a full distance matrix because otherwise it will only be able to simulate connection formation between nodes we already know are connected in the observed data, and thus it would be highly unsurprising when the model fits are so good. The only alternative is to use fiber lengths and attempt to interpolate for the missing values, but this is not widely done because it is unclear whether the interpolation is good enough. This is why Euclidean distance is the preferred metric. We have clarified this in the revised manuscript:

Results section lines 266-267: “In these models, binary networks are generated by a probabilistic equation that determines if any pair of unconnected nodes (i and j) will form a connection (Fig. 3):”

Results section lines 273-275: “Euclidean distance, which is highly correlated with fiber length³⁸, is used because a model requires a full distance matrix, including regions that have not been connected yet.”

Figure 6: Is the preterm model representative of preterm at term equivalent age, or preterm during preterm period?

The preterm model is derived from the economic wiring constraints of the entire preterm infant group. These wiring constraints were calculated by controlling for PMA, so the preterm model is representative of wiring constraints required to simulate preterm infants regardless of PMA. We added this to clarify:

Methods section lines 629-632: “For the ‘term’ and ‘preterm’ representative parameters were calculated by the group average of GA at birth-predicted parameters while controlling for PMA in GAMs. Group representative models were run with 1,000 simulations with a single parameter combination.”

L505-516: Please provide more details about the number of streamlines generated, where were the seeds placed? How was WM and GM parcellated? Please provide more information about the 60% consensus thresholding performed, and its justification. How was the “325” streamlines threshold selected?

For neonates scanned at or above 32 weeks PMA had an average of 1.65×10^6 streamlines ($\pm 9.77 \times 10^3$), whereas less than 32 weeks PMA neonates have a mean of 2.20×10^6 ($\pm 4.31 \times 10^4$). We decided to include a consensus threshold as it has been previously demonstrated to reduce the rate false positive and negative detection of connections (Reus & van den Heuvel 2013). Specifically, a 60% consensus threshold has been identified as the most effective balance of using group-level

connections to threshold networks and thus is why we chose this as our first thresholding step (Reus & van den Heuvel 2013). After this, our binarization threshold was decided based on the streamline threshold that would achieve an average 10% density across the whole sample. This is to achieve an average density that is both consistent with past generative network modeling work (Akarca et al., 2021; Betzel et al., 2016) and pragmatically speaking this means we can run in total 7,470,000 simulations. We have clarified these decisions and the rationale behind them in the revised manuscript:

Methods section lines 550-558: “Following previous neonatal studies, fiber tracking was performed in DSI Studio with a turning angle of 35°, step size of 1mm, maximum fiber count of 5,000,000, minimum fiber length of 30mm, maximum fiber length of 250mm, and a random initial seed propagation direction⁴⁹. Connectivity type was count-end, indicating that streamlines will be counted as present between two regions when the streamline ends within those regions. The scans were registered to a neonatal template⁶⁶ and parcellated using the Automated Anatomical Labeling 90 (AAL90) atlas for neonates⁶⁷. The AAL90 adult atlas was constructed with anatomical localization and from which neonatal templates, tissue probability maps (grey matter, white matter, and cerebral spinal fluid), and brain parcellation were created⁶⁷.”

Methods section lines 571-583: “For thresholding, we employed both consensus and absolute approaches. To reduce the rate of false positive and negative detection in tractography, a 60% consensus threshold was initially applied⁶⁸. This thresholding results in streamlines between regions that are present in 60% of the sample being retained. Following this threshold, we conducted two density-based analyses. First, following previous generative network modeling methods, we performed a 325 binarization threshold to yield an average network density of 10% across the sample^{22,37}. To explore the effects of density on topology, we complete a secondary density-controlled analysis where each individual had a *proportion* binarization threshold that yielded an exactly 10% dense network. By applying these two thresholding techniques, we can complete a standard GNM analysis with varied density, followed by an exploration of how controlling for density effects topological interpretations. An additional analysis exploring how progressively more sparse networks, ranging from 24%-8% density, relate to changes in topology, topological fingerprints and representative connectivity matrices can be found in the supplementary materials (Supplementary Fig. 2).”

L524: Please clarify if using weighted or binary graph theory metrics.

Revised sentence lines 592-593: “All graph theory network measures were calculated from binarized networks using the Brain Connectivity Toolbox (BCT) in MATLAB 2020b¹⁰.”

L510: Isn't 30mm minimum fibre length too restrictive for preterm babies?

We appreciate the concern with the 30mm minimum fiber length. We have provided an additional analysis to address this – see Main Comment, response 1i.

Abstract: why is in utero imaging “poor”? Perhaps this is a bit too ambiguous and unspecific. Fetal MRI has advanced a lot recently, and ultrasound is fit for purpose.

We agree and have changed this to:

Abstract lines 32-33: “Due to practical limitations, such as technological advancements and data availability of fetal MRI, there is still much we do not know about early topological development.”

The term “birth age” is a bit unusual in the neonatal literature. I would suggest to change “birth age” for “gestational age at birth” or “GA at birth”?

We agree and have changed all instances of ‘birth age’ to ‘gestational age (GA) at birth’ and clarified this:

Results section lines 105-108: “Postmenstrual age (PMA) refers to the time since the mother’s last menstrual period (including both gestation and time since birth), and gestational age (GA) at birth refers to the gestational week of the infant when they were born. GA at birth defines their term: an infant is *preterm* if born less than 37 weeks gestation and *term* if born 37 weeks or later.”

In Figure 1C it is a bit unclear what is significant at $p < 0.001$. Please include in the legend the statistical model tested.

We have changed the legend to include lines 128-129: “(C) Connectome density significantly increased across PMA and GA at birth (assessed with a generalized additive model).”

Figure 2A: why association with postconceptional age only starts at ~32 weeks in this panel?

The youngest PMA in this data is 26 weeks, however, the GA at birth line covered the PMA line in this graph. We have edited all graphs to make the PMA line more visible.

L205: No fetal data assessed in this manuscript, so please be careful referring to “prenatal” development.

We apologize for this misuse of “prenatal”. We have changed the sentence:

Results section lines 247-248: “During the early stages of neonatal development (approximately before 35 weeks PMA), structural brain networks rapidly undergo rapid formation of new connections.”

L208: It is unclear if differences are due to ex utero exposure, or if there is something different in the brain of preterm born individuals, which may be associated to the reason why they are born preterm.

We agree and have pondered this very much ourselves. We have clarified language in the manuscript to avoid implying the results observed are purely related to ex-utero exposure:

Results section lines 361-363: “However, early GA at birth is related to a significant acceleration of this effect, resulting in weaker penalization of long-range connections for infants born early.”

Results section lines 366-368: “GA at birth appears to be associated with an acceleration in this process, demonstrated by a higher reliance on topological value for infants born early than at term.”

We have also chosen to discuss this point in depth in the “Limitations” section.

Limitations section Lines 516-522: “Lastly, it cannot be overlooked that the differences observed in this study are potentially related to factors that precede the event of a preterm birth. Indeed, recent advancements in fetal fMRI have led to the ability to identify infants that will be born preterm from reduced fetal fMRI signal⁵⁸. These results, along with the high number of environmental factors associated with preterm birth (e.g., low socio-economic status and maternal history of premature birth)⁵⁹ suggests the presence of differences for those born early that may precede the premature birth itself. Thus, it is important to not interpret the present study as causal.”

L512: The authors refer to Fitzgibbon et al for the neonatal space. Do they mean to reference Schuh et al. template instead?

We apologize for this error. The neonatal template used was the default template provided in DSI studio (for source, please see: https://github.com/frankyeh/DSI-Studio-atlas/blob/main/dHCP_neonate/dHCP_neonate.README.txt) which is indeed the Schuh et al. (2018) template. We have changed the citation.

Response to Reviewer 2:

In this paper, the authors present a network analysis of structural connectivity data acquired from a large cohort of neonates (dHCP). Through a series of descriptive analyses performed on binarized brain networks, they demonstrate an increase in network density with consequent decreases in modularity, path length and increases in efficiency. Using a simple generative procedure, the authors generate simulated networks that share similar properties with the empirical connectomes and use the parameters of these models to infer putative constraints on early network development. Overall, the observed network changes replicate previous findings, and the generative network modelling approach adds some interesting novel aspects, however, as written, the paper is difficult to follow and I have some methodological concerns that limit my enthusiasm for the study. I have listed some specific comments below.

Thank you for this incredibly helpful review. We appreciate the time an in-depth review like this takes and found the comments very useful for making improvements to the manuscript.

1) Using a consensus thresholding approach, the authors find that network density is lower in younger infants, with a preservation of longer connections. The Methods state that a minimum of 30mm cutoff was used for tractography. This strikes me as very high, considering at 25 weeks the occipital-frontal diameter is only around 75-80mm. Presumably, in the youngest infants, many shorter fibres would thus be rejected prior to construction of the network matrices? The authors state that head circumference was considered in some statistical analyses but this would not account for the absence of connections prior to analysis.

This is a great point. Before we proceed to any modeling we need to convince readers that the tractography, and subsequent design choices, result in connectomes that are a good reflection of the underlying neonatal neuroanatomy. To address this, we have provided an additional analysis which includes tractography using a minimum fiber length of 5-30mm. Despite different fiber lengths, our networks display similar density areas (e.g., cingulum and parietal-occipital connections) and very similar adjacency matrices (Supplementary Fig. 11C). Indeed, Spearman rank order correlations reveals that regardless of fiber length during tractography, the ordering of regions is highly similar (5mm to 10mm $r_s = 0.75$; 10mm to 20mm $r_s = 0.74$; 20mm to 30mm $r_s = 0.74$). As shown in Supplementary Figure 11A, there is a non-linear relationship between fiber length and density – actually the 30mm fiber length used in our manuscript produces denser networks than, say, a 5mm fibre length. Only a 10mm fiber length would produce a denser network. Crucially, the local topological arrangement of the networks – the thing we subsequently model – is almost identical regardless of the fiber length. This can be seen in the topological ‘fingerprints’ in the new Supplementary Figure 11B. Finally, we directly compared the tractography before and after 32 weeks PMA with our original 30mm tractography (Supplementary Figure 11D), which shows that despite differences in density, the matrices are highly similar ($r_s = 0.98$). This gives us confidence that these very early PMA scans have not been constructed in a manner that is somehow fundamentally different than expected (e.g., missing connections in a particular area).

Please see Response to Reviewer 1 Main Comments response 1i for added figure and in-text changes.

2) Network models are necessarily an abstraction from the neurobiology of the brain, however, no commentary is provided on how these network-based parameters may relate to the well-described differences in white matter development following preterm birth. Significant differences in white matter anisotropy have been observed across most of the developing white matter, largely driven by disruptions to (pre)myelination processes. There is extensive literature on the establishment, timing and location of white matter tracts across cortical areas during this period – see work by Emi Takahashi amongst others. The authors should consider their findings in light of this large body of existing work.

We agree and have included a substantial new section in the revised Limitations section (Page 20, Lines 502-514):

“Macroscale binarized connectomes are compressed representations of neurobiology. A connection being present or absent between two regions could reflect the strength or developmental stage of that connection – not the actual existence of such a connection. This is extremely important to consider during interpretation, as premature birth is associated not just with reduced connectivity, but also an impairment to the migration and maturation of neural cells⁵⁵. The preterm period overlaps with the pre-myelination stage, during which there is rapid movement and maturation of glial cells⁵⁶. Disruption during initial stages of glial cell development become more apparent in infancy in the myelination process⁵⁷. However, topological analysis of macroscale networks is likely influenced by these early cell-level difference, as it relates to decreasing anisotropy and associated metrics during the preterm period⁵⁶. Thus, our topological analysis of the macroscale neonatal network cannot be interpreted as solely connection-level differences and is likely, if not definitely, encompassing the disruption of pre-myelination development in infants born at early GA.”

3) In the Results section (para 2.) the sample is described as scanned from 20 to 44 weeks and through the Results, trends are described from '20 weeks to..'. No infants were scanned at 20 weeks, the youngest age MRI was acquired in the dHCP is 26 weeks, with the youngest age at birth at 23 weeks. Please correct.

We apologise for the error – this is corrected in the revised manuscript.

4) On this, 'postconceptional' age is a poorly defined term. In this context, postmenstrual age is preferred as a more precise estimation of age– see <https://publications.aap.org/pediatrics/article/114/5/1362/67715/Age-Terminology-During-the-Perinatal-Period>. Similarly, 'ex-uterine' is typically 'extra-uterine'.

We agree and have adopted the reviewer's recommended nomenclature of postmenstrual age (PMA) throughout the revision and have changed 'ex-uterine' to 'extra-uterine' throughout.

Throughout, results are split based on 'post-conception' and 'birth' age. I find this makes the results difficult to follow. Digging into the methods, it appears that postconceptional age were controlled for in GAM models and although this isn't stated anywhere in the results section, I assume that all the 'birth age' results are corrected for age at scan? No information is provided on how this correction was performed, was postceptional age included as an additional smooth term in the birth age models? Were the postconceptional age models also corrected for birth age? These two parameters are obviously closely related, it is unclear why, when using the flexible GAM framework, a unified model wasn't presented including additive smooth terms for both age at birth and age at scan?

We agree that the statistical control of age metrics was unclear – the reviewer is correct we do use a single GAM model with additive smooth terms for both PMA and GA at birth. We have added more explicit detail to both the Methods and Results sections, to highlight that indeed the GAM models control for each metric when examining the other.

Results section lines 114-117: "We considered both age metrics' effects on macroscopic neural connectivity by statistically controlling for PMA when exploring GA at birth and vice versa using generalized additive models (see "Methods"; "Statistics" for detail)."

Methods section lines 681-683: "PMA, sex, head circumference, translation, and rotation were controlled for in GAMs on observed data when exploring GA at birth. When looking at PMA, the same variables were controlled for, in addition to GA at birth."

We have also added a term-equivalent age analysis (see response to Reviewer 1 Main comment response 1i).

How was the significance of the spatial embedding results determined? There are several associations with highly significant results ($p=10^{-4}$ – 10^{-6}) that do not appear well-represented by the accompanying plots. Figure S6, for example, where highly variable positive and negative correlations between empirical and predicted clustering coefficients are displayed with no apparent average effect and little correspondence between surface plots are presented as $p<0.001$? Are the statistics suitable considering spatially correlated datapoints?

We apologize for the lack of clarity regarding spatial embedding. Generally, spatial embedding is significant replication of regional organization in the simulated networks, relative to the observed networks. However, we statistically tested for this in two different ways. First, for the consensus network modeling, we simply used Pearson Correlation between the simulated and observed network's local measures (Supplementary Figure 8D). For this analysis, we can use a correlation because we are only comparing the simulations to a single consensus network. However, for

individual generative network models, we are comparing the similarity between each individual's simulation and their observed network, across the entire sample. For this analysis, we chose to use linear mixed effects models (LMEM) with region and participants as random effects. We decided to use LMEM because while it's likely that spatial embedding varies across brain regions and between participants, and in this second test we were only interested in the success of spatial embedding *across the entire sample*.

We chose to illustrate these results with an average surface plot, cumulative density function, and correlations because we believe they convey the accuracy of the spatial embedding beyond the LMEM. For example, clustering coefficient was significant as described by the LMEM, but at an individual-level simulations may not be replicating clustering coefficient accurately (Supplementary Fig. 6D; estimate = 0.02, $t(53.37) = 2.85$, $p = 6.13 \times 10^{-3}$). Therefore, we believe that these plots provide important context beyond the statistical methods as spatial embedding does not appear to be present in all individuals across all measures assessed.

We apologize that the spatial embedding methods and results were not cleared in the original manuscript. We have provided a more detailed explanation in the Methods as well as addressing the sample vs. individual interpretation of spatial embedding in the Results section:

Figure 4 legend lines 332-333: "The surface plot depicts the average degree in the left hemisphere for the observed and simulated networks in all regions."

Results section lines 302-345: "For individually fit GNMs, spatial embedding was assessed with linear mixed effects models (LMEM) with node and participant as random effects. This analysis showed that local organization of the simulated networks was significantly related to that of the observed networks for edge length (estimate = 3.24×10^{-2} , $t(59.98) = 3.52$, $p = 8.42 \times 10^{-4}$), degree (Fig. 4F; estimate = 0.04, $t(78.46) = 4.30$, $p = 4.96 \times 10^{-5}$), local efficiency (estimate = 0.02, $t(60.56) = 3.17$, $p = 2.40 \times 10^{-3}$), clustering coefficient (estimate = 0.02, $t(53.37) = 2.85$, $p = 6.13 \times 10^{-3}$), and matching (estimate = 0.06, $t(89.26) = 4.61$, $p = 1.33 \times 10^{-5}$) but not betweenness ($p = 0.500$) (Supplementary Fig. 9). It is important to note that while we see significant spatial embedding across the sample with LMEM, individual-level spatial embedding appears variable given correlational plots⁴³ (Fig. 4F; Supplementary Fig. 9)."

Methods section lines 684-688: "Spatial embedding, which is replication of local organization, was calculated in 80 cortical regions. For consensus network GNMs, spatial embedding was assessed with Pearson correlations. For individually-fit GNMS, spatial embedding was assessed with linear mixed effects models with participants and nodes as random effects with the *lme4* package in R⁷⁵."

Response to Reviewer 3:

In this submission, the authors fit generative models to neonatal structural connectomes. They report changes in network properties of connectomes and that their models fit the data reasonably well.

The results are interesting, but I have a suspicion that much of the findings can be explained by age-related changes in density (effects that the authors describe) but also data-quality issues associated with the acquisition and subsequent processing of neonate imaging data.

There's also a tendency to treat the models as "mechanistic". That is, any changes in estimated parameters are interpreted as changes in the trajectory of the real network. "We see an increase in η with age? Well, older brains must be subjected to stronger spatial constraints." The generative models are simply compressive. They take data and estimate the best fit given the model. But even great fits do not imply that the model is a plausible explanation of the network's trajectory.

We would like to thank the reviewer for taking the time to provide these constructive comments We broadly agree with both main points here: (i) we need to carefully explore the role of density, and the quality of tractography for the neonates; (ii) we must be careful in what we conclude from the modeling.

Detailed comment Line 64: Why the different citation style for the reviews?

We have changed this line to match the rest of the revised manuscript.

Detailed comment Line 78: Generative models are useful but they sure are not mechanistic. The models used here and by others are simply *not* developmental models. They are compressive. They identify a set of parameterized wiring rules that create synthetic networks. The process by which the network is generated has *nothing*, not even stylistically, in common with the process by which connectomes develop.

We agree, these models are not mechanistic in that they do not tell us how connections form, and to put it bluntly, the brain clearly does not form binary connections one at a time. We have tried throughout the revised manuscript to make sure that we are not giving that impression.

We think there are two important things we can learn from the compressive quality of these models. The first is the direct comparison of different wiring rules. That homophily based models achieve far better compression than alternative models tells us something important about the underlying trade-off a good model needs to approximate to produce a realistic network. The second is the changing parameter combinations needed to achieve the best compression of each participant's individual connectome. This tells us that to realistically approximate each person's connectome this trade-off needs to be differently tuned.

From both take-aways, we learn about an economic wiring constraint window to create realistic topology at the entire sample or group level. These models are therefore providing contextual information - not mechanistic explanations. We did not intend to imply in the original manuscript that these are developmental models and have clarified this in the revised manuscript:

Introduction section lines 76-78: “Generative network modeling (GNM) is a computational framework that can provide insight into the principles of topological organization because these models achieve a compressed representation of the principle constraints upon brain networks³³⁻³⁶.”

Results section lines 260-262: “Now, we take a step further to test whether or how these organizational differences are shaped by economic wiring constraints by simulating the formation of those networks.”

Results section lines 384-387: “To explore how these altered wiring constraints drive the emergence of organizational characteristics, we ran two GNMs with selected parameters determined by the mean of the optimal GA at birth-predicted parameters for preterm ($\eta = -1.85$; $\gamma = 0.34$) and term neonates ($\eta = -1.74$; $\gamma = 0.32$) (Fig. 6A).”

Discussion section lines 484-487: “Of course, the GNM does not provide mechanistic insight into *how* these changes in wiring constraint occur, but this shift in trade off, resulting the formation of different types of connections, provides a compressed account of why changing trade-offs shape the emerging topology.”

Discussion section lines 499-500: “While we see substantial replication of topological metrics, GNMs are not identical to observed connectomes and thus cannot be interpreted as exact replications.”

Major comment: Showing that the model parameters vary with age or are correlated with different demographic/behavioral measures is an interesting exercise, but it doesn't imply different mechanisms of brain growth.

Most damning, the generated networks do not look like connectomes. They match a bunch of high-level distributional traits – e.g. degree, clustering, betweenness, edge length distribution. But the edge-level matches are just not there. That is, the actual wiring diagram, despite “baking into the model” the spatial locations of nodes, does not match what we measure in reality.

We completely agree that GNMs do not perfectly match reality, however, we need to make our presentation of the similarities between the simulated and observed networks clearer because they do in fact look a lot like connectomes. In our analysis we explored this in two ways – the global distributions network properties (degree, clustering etc.), and in terms of local organisation (both at a group level and across participants). Global network distributions of the generated networks closely mirror those of the observed networks. *But the local network organization is replicated too.* Regions that have a high degree in the observed network are significantly more likely to have a high degree in the simulations (Figure 4F). Another way of testing whether

the simulations capture the local organizational properties of the observed networks is with the ‘topological fingerprint’. This metric tests whether the within-network local arrangement of topological properties (such as areas with high centrality having low modularity) are captured by the simulation. We show that indeed the homophily-based models do an incredibly good job of capturing these local properties (Figure 4D). We believe that a fair interpretation of these results is that the generated networks are replicating both global and local topological features of the observed connectomes, even if they do not do so perfectly.

We did not claim that there were edge-level matches, as our intent with these models was not to describe mechanistic brain growth. Our analysis is strictly at the level of topological characteristics rather than topography. One reason this is important is, because like all published generative models, we are using binary networks and generative models which could never precisely capture the nuanced growth and strengthening of individual connections. We made changes in the manuscript to clarify this:

Introduction section lines 76-78: “Generative network modeling (GNM) is a computational framework that can provide insight into the principles of topological organization because these models achieve a compressed representation of the principle constraints upon brain networks³³⁻³⁶.”

Discussion section lines 484-487: “Of course, the GNM does not provide mechanistic insight into *how* these changes in wiring constraint occur, but this shift in trade off, resulting the formation of different types of connections, provides a compressed account of why changing trade-offs shape the emerging topology.”

Discussion section lines 499-500: “While we see substantial replication of topological metrics, GNMs are not identical to observed connectomes and thus cannot be interpreted as exact replications.”

Detailed comment Line 103: what sort of data quality assessments are performed on the MRI data? It’s not unreasonable to imagine that data quality for smaller brains in which white-matter is still developing have poorer signal than, say, an adult brain for which most dMRI acquisitions are optimized. It seems possible that many of the reported developmental effects could, in fact, reflect changes in data quality with age.

We agree that poorer signal in smaller and younger brains is a concern in this analysis. We do describe poor scan quality assessment in the Methods lines 528-530: “Neighboring DWI correlation identified $n = 34$ subjects with poor scan quality who were removed from the sample⁶¹”. However, beyond scan quality there is reasonable concern around tractography validity. We have provided an additional term-equivalent age analysis where we have excluded all infants scanned < 37 weeks PMA. This analysis is highly consistent with or original analysis. This suggests that the relationship between topology and GA at birth is not related solely to protentional quality issues with early scanned infants (please see Reviewer 1 Main Comment response 1i for the figure and in-text changes).

Detailed comment Figure 1, panel D: The matrices are not particularly informative w/o node/region labels. For the average binarized network, the colorbar label is missing.

We have updated the figure to include region and colorbar label.

Detailed comment Line 142: Does density really double during this period? That is, does the number of white-matter links actually increase that much? I know there's a period of myelin refinement, but the idea that there are *brand new* tracts forming does not seem plausible. It seems more likely to me that there are issues with dMRI and the acquisition (and maybe tractography) that make reconstructing some of these tracts difficult and in a way that the difficulty of reconstruction varies systematically with age.

The whole brain volume increases from 22.27 +/-0.80 to 367.3 +/- 15.7 ml from 19 gestational weeks to just after birth (Huang et al., 2006). White matter connectivity specifically increases drastically during this period (Ball et al., 2014; Song et al., 2017; Van Den Heuvel et al., 2015). Indeed, reports from 20 – 40 PMA suggest that there is a 10-fold increase in network strength (Song et al., 2017). Therefore, we are confident that the increases in density we observed are a reflection of expected developmental patterns.

However, given our stringent thresholding it is unlikely that the increase in density is representing brand new tracts in every case, but more likely that these tracts have become prominent enough to pass the threshold. We agree with the reviewer that before we proceed with the modeling it is first important to test, as far as possible, that the tractography and processing decisions are not systematically changing the quality of the reconstruction in those born earlier. There are multiple sensitivity analyses in the revised manuscript that now add three additional tractographies, each with a minimum fiber length threshold (Reviewer 1, Main Comment response 1i), and with different density thresholds (Reviewer 1, Main Comment response 1ii). However, it is incredibly difficult to disentangle genuine differences in underlying neurobiology from differences in the ease with which algorithms can resolve the tractography. For that reason, the most useful additional analysis for the current purposes is a term-equivalent analysis. The revised manuscript now includes a term-equivalent age analysis in which all scans taken earlier than 37 weeks PMA have been excluded. The results of this new analysis are highly consistent with the original (please see Reviewer 1, Main Comment response 1i). Therefore, even if tractography issues are present for infants scanned at early PMA, this additional analysis confirms that it has not affected our interpretation of topological patterns.

Detailed comment Line 145: How do you estimate a fractional degree of freedom? Not a serious complain, but it's not obvious to me.

Estimated degrees of freedom (EDF) are a standard output of the generalized additive models (GAMs). EDF is capturing the complexity of the smoothing function, or in other words represents the amount 'wiggle' or curves in the data that are required to fit the data well. If the EDF is close to 1, it indicates a linear relationship because few curves are required to fit the data. The higher the EDF is, the more curves are required in the model fit, which indicates a more flexible smoothing function is used to fit the data.

Essentially, EDF can be interpreted as a representation of how complex the patterns in the data are and as a check of potential over-fitting because extremely high EDF may indicate an over-fit model.

Detailed comment Line 152: Of course network modularity decreases, it's density dependent. If you create random networks with gradually ramped up density, their modularity will decrease monotonically. Careful comparison against null models is absolutely needed to support these claims. It's not obvious that the reported effects have to do with density or true changes in organization of the networks. Frankly, the same is expected for the other network metrics; density biases their values. I haven't seen it demonstrated, but I'm sure there's a similar density effect for rich clubs.

We agree – this is a great point. Modularity, and indeed multiple other network properties, are inherently related to density. In the revised manuscript we include a density-controlled analysis in which we use a proportion threshold of 10%, meaning that each connectome has precisely the same density. While this will likely remove important individual differences, it does allow us to make direct topological comparisons between participants, unaffected by differences in density. Many measures of segregation and integration are indeed related to density; the relationships between some measures and PMA, or GA at birth, change when controlling for density, as we might expect. For instance, the networks of those born preterm are actually *more*, and not *less*, efficient when you control for overall density. However, certain key topological properties are unaffected by density, for instance the impact of PMA and GA at birth on the number and length of rich-club connections. We have incorporated this with the revised Methods and Results and have updated Figure 2 to include the density-controlled analyses (please see Reviewer 1, Main Comment response 1 for more details).

Another sensitivity analysis we incorporated within the revised manuscript was varying overall network density between 24% and 8%. One thing we wanted to check is whether the topological arrangement of network properties is changed or preserved as the density changes. This is particularly important with regard to the subsequent modeling. Importantly, regardless of the differences in density, the topological relationships (a.k.a. the 'topological fingerprint') remains unchanged (please see Reviewer 1, Main Comment response 1 for more details).

Detailed comment Figure 2: It's not surprising that mean RC edge length (in ED) increased with age, right? Even if the exact same edges and RC members are preserved at every age, growing brains/volumes mean that edges will tend to grow.

Rich clubs are defined by a specific set of parcellated nodes which are equal distance apart for all infants. However, we found that even when we control for PMA, which indicates the developmental age the infant is at, we still see significant increases in RC lengths across GA at birth. Therefore, there is a relationship between RC length and the timing of birth separate from age. We replicated this effect in our term-equivalent age analysis where we only included infants 37 weeks PMA age or older, finding that there is indeed a significant relationship between GA at birth and RC length

(please see Reviewer 1, Main Comment response 1i). In the revised manuscript we have clarified this in the 'Methods' section:

Methods sections lines 607-611: “The 27 rich club nodes from the consensus network were used to define connection types for all participants as rich (rich node to rich node connection), feeder (rich node to non-rich node connection), and local (non-rich node to non-rich node). Average length of connections was also explored across each connection type, defined by the Euclidean distances between connected nodes.”

Detailed comment Line 271: I'm genuinely curious if the authors can shed some light on why homophily does so well as the “value” term. The explanations in earlier generative modeling papers were never fully satisfying. I also feel like homophily is kind of the same thing as the distance penalty. If two nodes are near one another, we'll connect them together. Therefore, nearby nodes also have similar connectivity profiles, making it even more likely that they connect. How does the homophily rule promote long connections? I'm still curious about that and have been since our first generative modeling papers. But it must be real, there's a dozen or so papers that have all replicated the initial findings.

This is a great question – and one we keep finding ourselves returning to. A helpful starting point is to first consider what would a model look like if it only made connections based on cost (i.e. distance) minimizing? Our study includes precisely this model – the Spatial model contains only the distance-minimizing constraint. This model performed the worst out of all models tested (Figure 4A). This suggests that simply making the shortest connections does not replicate real connectomes well. When we look at clustering and degree-based value rules, we find that homophily outperforms them all (Figure 4A). This suggests that the organization of the regions (at least for clustering and degree metrics) is less relevant to the formation of connections than the *similarly of connections* between regions. This is interesting, particularly because forming a connection with high-degree regions is a common theory of how networks grow.

So what is it that makes homophily so important? We believe homophily is less about local knowledge of spatially close nodes – which is the only consideration for 'cost' – but likely has more to do with communicability. Homophily is a locally computable heuristic for network communicability, in other words, it is a locally 'knowable' measure of the diffusion of signals across the network. Thus, we believe homophily is likely captures a signal diffusion process that, while local connectivity is relevant, is compressing more general network communicability. A good test of this would be to constrain networks to optimise communicability within a distance-penalising regime, and test whether it can be simulated with a homophily GNM. This is precisely what spatially-embedded Recurrent Neural Networks (seRNNs) do, and if you fit a generative model to the resulting network you do indeed need a homophily wiring rule to get a good fit (Achterberg et al., 2023). With this perspective, homophily is likely to drive local connections but opens the possibility of long-range connections to form if the regions have sufficient connectivity via other regions. This may also function as a general cost-saving process – as two regions that are well connected through many short connections between regions may be more efficient with a direct connection.

As the reviewer can perhaps tell, we find this issue really interesting and are happy to provide an extended section on this in the revised Discussion:

Discussion section lines 466-477: “*Why homophily?* The homophily modeling principle, emerges consistently as a necessary ingredient when simulating networks across different scales and species^{22,34,38-41}. This consistency speaks to its capacity to capture some fundamental, locally computable, network property that simulates the formation of topologically desirable characteristics. One emerging theory is that homophily is a heuristic for efficient communicability – the traversability of information through a network between two points. In a recent paper utilizing spatially-embedded recurrent neural networks, researchers used a regularization term that incorporated both distance and communicability. The researchers found that a homophily rule is required to fit the generative model to the resulting networks⁵³. Thus, one possibility is that homophily reflects a locally computable metric that allows a network to form connections based on minimizing redundant communicability and that this in turn drives the characteristic topology of biological networks.”

Response to Reviewer # 1

In this revised version of their manuscript, Mousley and colleagues have clarified or addressed many of my initial concerns. They have performed tractographies with different parameters (including minimum streamline length), and assessed and clarified the effect of network density on their results, which I think greatly improves their study.

1) My main remaining concern is that given that one of the aims of the study is to characterise the effect of preterm birth in brain development, I do not understand the decision to exclude all the valuable scans of preterm babies scanned at term equivalent age (i.e. GA at birth < 37 with PMA>37 weeks) which are publicly available as part of the dHCP dataset.

In their response to my previous comments they justify this decision based on the idea of avoiding confounding factors: “As the main purpose of this project was to perform generative network modelling, we chose to select a cross-sectional sample to avoid potential confounding factors in the simulation analysis”. However, I have the opposite impression, by not including preterm at term scans they are massively impairing the capacity of their models to deal with the confounding effect of GA at birth and PMA at scan. Preterm at term scans would help them to better tune GAM models allowing the authors to actually disentangle the associations with preterm birth from the associations with brain maturation. Without the preterm at term scans I am not sure the authors can hold any interpretation regarding the effect of preterm birth in brain development, and as far as I understand, they are “only” characterising different aspects of brain maturation. Since they are not including comparable data (e.g. term babies vs preterm babies scanned at term equivalent age), I don’t see how their GAMs can model what they are not trained to model. In other words, the authors cannot be sure to what extent the effects that they attribute to preterm birth (including their GNM “preterm” model) would also be observed at the same GA in utero, in babies that will then be born at term (if that data was available, which of course it isn’t). If they included preterm at term scans, they could directly compare babies with the same postconceptional/PMA age, where the only difference between the two groups (term at term vs preterm at term) is GA at birth, effectively disentangling the effect of GA at birth from PMA at scan.

Thank you for this point. The original reason we didn’t include all of those (preterm and term) scans is because they are the *second* scans for those individuals. This means that you can’t use a GAM to model them alongside all the other scans, you’d need to use an alternative mixed model to allow for two scans in a few cases.

That all said, we do agree with the reviewer’s position. Given the point we wish to make – preterm birth shapes network and model parameters – it is a shame to have just $n = 40$ preterm babies in the term-equivalent analysis we included in the revised manuscript.

As such, we’ve put some careful thought into the best way of incorporating those extra scans. We have taken $n = 120$ scans from babies born preterm and scanned at

term-equivalent age (including the scans the reviewer mentions). We then conducted a propensity-matched analysis, using a group perfectly matched for postmenstrual age and sex ($n = 120$). In essence, this gives us two groups of *identical* size, with scans taken at an *identical* PMA, but with one group being born preterm and the other born at term. We think that comparing these two groups directly is the clearest and simplest way of testing whether the key results – preterm effects on network modularity, efficiency and model parameters – are genuinely reflective of being born preterm.

We tested the key topological measures (i.e., modularity and global efficiency) and GNM parameters between groups. These results replicate our original analysis, and the term-equivalent age analysis included in the previous revision. This third approach, the propensity matched analysis, also shows that preterm infants have more modular, less efficient networks, compared to term infants (see below: Supplementary Fig. 3B,C). In addition, this new analysis confirms that to simulate the connectomes of infants born preterm requires a significantly stronger η parameters compared to term infants. The gamma is also higher, as in the other two analyses, but this is not significant in the new analysis (see below: Supplementary Fig. 10D).

The new analysis and results are incorporated within the latest revision of the manuscript. These updating findings strongly indicate that – even when infants are scanned at *identical* postmenstrual age, preterm infants still have more modular and less efficient networks, and a stronger cost penalty is needed to simulate them.

Updated sections:

Results lines 175-78: “In addition, preterm infants had significantly more modular networks compared to term infants ($t(216.96) = 12.16, p < 2.200 \times 10^{-16}$) in the propensity-matched analysis where the two groups are *identically* matched on sex and PMA (Supplementary Fig. 3B; see “Methods”; “Participants”).”

Results lines 193-96: “This result was replicated in the term-equivalent age analysis (Supplementary Fig. 3C; $F_{global\ efficiency, GA\ at\ birth} = 2.71$, estimated $df = 3.60, p = 0.029$) and is consistent with the propensity-matched analysis in which preterm infants had significantly lower global efficiency compared to term infants ($t(203.35) = -7.24, p = 9.193 \times 10^{-12}$).”

Results lines 382-84: “These results are confirmed through density-controlled, term-equivalent age, and propensity-matched analyses. However, in the propensity-matched analysis, γ is smaller in the preterm group, though the difference is not significant (Supplementary Fig. 10).”

Methods lines 556-61: “To provide further confidence in term-equivalent age interpretations, we have also included a propensity-matched analysis with additional preterm infants scanned at term-equivalent age ($n = 80$). With this data, we create two groups (each $n = 120$) matched in sex and PMA – one group with infants born preterm and the other with infants born at term. There is no significant difference in head circumference between these groups ($p = 0.282$).”

Methods lines 603-05: “This thresholding technique was repeated for the propensity-matched analysis ($n = 240$) with a 390 binarization threshold to yield an average network density of 10%.”

Methods lines 613-18: “The propensity-matched sample was included in this exploration of multiple densities and showed that global efficiency and modularity differ significantly between groups across 5-24% average density networks (Supplementary Fig. 2E). Together these additional analyses demonstrated that while more dense networks would yield overall more efficient, less modular topology, that this does not change significant effects between term and preterm groups.”

Methods lines 722-24: “For the propensity-matched analysis, Welch two sample t-Tests were performed in R to examine group differences in modulatory, global efficiency, and GNM parameters.”

Supplementary Figure 3. Term-equivalent age topological analysis. (A) Density significantly increases across GA at birth. (B) Modularity significantly decreases across GA at birth and propensity-matched analysis shows that preterm infants have significantly higher modularity compared to term infants. (C) Global efficiency significantly increases across GA at birth and propensity-matched shows preterm infants have significantly lower global efficiency compared to term infants. (D) There is no significant change in the number of rich club connections, however (E) the average length of rich club connections significantly increases across GA at birth. The shaded area indicates 95% confidence intervals. The grey dotted line indicates the cut-off for term birth (37 weeks GA or later is term-born). *** indicates $p < 0.001$, ** indicates $p < 0.01$, * indicates $p < 0.05$.

Supplementary Figure 10. Density-controlled analysis and parameter changes. (A) Energies from 90,000 simulations in $-8 \leq \eta \leq 0$ and $-8 \leq \gamma \leq 8$ sampling space fit to the density-controlled consensus network. The violin plot displays the lowest 1,000 energies for all 13 'value' rules. 'Matching' had the lowest overall energy (0.08) followed by 'neighbors' (0.10). (B) The average energy landscape of the density-controlled 'matching' models. The white dots indicate the location of the best-fit models for all participants. The violin plot demonstrates the disruption of energies of the best-fit models (maximum = 0.28, minimum = 0.06, $M = 0.09$, $SD = 0.02$). (C) Energy significantly decreased across PMA ($F_{energy,PMA} = 32.88$, estimated $df = 7.46$, $p = 2.00 \times 10^{-16}$) but not birth age ($p = 0.380$). (D) The original analysis (repeated from Figure 5) demonstrates that η significantly decreases across PMA but increases across GA at birth. The opposite relationship is present for γ , which significantly increases across PMA and decreases across GA at birth. The density-controlled analysis replicates these results ($F_{\eta,PMA} = 39.56$, estimated $df = 5.51$, $p = 2.00 \times 10^{-16}$; $F_{\eta,GA\ at\ birth} = 5.14$, estimated $df = 3.19$, $p = 4.06 \times 10^{-4}$; $F_{\gamma,PMA} = 11.34$, estimated $df = 5.18$, $p = 2.00 \times 10^{-16}$; $F_{\gamma,GA\ at\ birth} = 14.67$, estimated $df = 1.12$, $p = 5.50 \times 10^{-5}$). The term-equivalent age analysis is also consistent ($F_{\eta,GA\ at\ birth} = 7.29$, estimated $df = 3.06$, $p = 1.92 \times 10^{-5}$; $F_{\gamma,GA\ at\ birth} = 5.67$, estimated $df = 2.30$, $p = 8.42 \times 10^{-4}$). The propensity-matched analysis shows significantly lower eta ($t(180.83) = -11.80$, $p = 2.200 \times 10^{-16}$) but not significantly higher gamma for the preterm group compared to the term group ($p = 0.792$). This suggests that differences in gamma at early PMA for preterm groups may resolve by term-equivalent age. The shaded area indicates 95% confidence intervals. The grey dotted line indicates the cut-off of term-born infants (37 weeks GA or later is term-born). *** indicates $p < 0.001$, ** indicates $p < 0.01$, * indicates $p < 0.05$.

I also have other suggestions and/or issues that I think the authors should consider addressing:

Thank you for these suggestions. They are incredibly helpful and we have incorporated them in the latest revision.

2) Figure 1C: I am not sure the authors have appropriately described what is

plotted here. Since the authors are plotting PMA at scan and GA at birth simultaneously in the x-axis, there should be two datapoints for each given density (one blue dot for GA at birth and one orange/yellow dot for PMA at scan). In other words, each density existing in the y-axis should always have at least two corresponding data points in the x-axis, as all scans (and therefore connectomes) have one age at birth and one age at scan. I struggle to see for instance the GA at birth (blue dots) corresponding to data with PMA at scan between 25 and 30 weeks PMA (with low densities, below 8%, yellow dots). These data don't seem to be taken into account when plotting GAMs model for GA at birth (with much fewer data points below 8% density).

Is the y-axis perhaps not really representing "density" but a corrected version of density given specific GA at birth / PMA at scan (and then labelling and legend are misleading)? Or are there missing datapoints? Similar effect can be seen in Figure 2 and Figure 5 for other variables, please clarify.

We apologize for the lack of clarity with these plots. Every infant does have two datapoints – one for GA at birth and one for postmenstrual age – however, there are no (or few) direct density alignments for two points because we are using the GAM-predicted scores. These plots are created with the `visreg()` function in R shows how the predictor variable (GA at birth or PMA) effects the outcome (density). Therefore, these are not the raw networks densities associated with age but density *after controlling for other variables*. We have changed the figure legend to make this clearer to readers.

Figure 1C, lines 128-29: “Connectome density significantly increased across PMA and GA at birth (represented as predicted density from generalized additive model).”

3) Figure 1D: Clarify what is edge weight representing (average number of streamlines?). Perhaps the authors want to consider whether streamline count is a valuable metric to use at all (see for instance the “6th sin” in <https://doi.org/10.3390/diagnostics9030115>), and consider the effect of using such an unreliable measure as input to create the binary networks that they are basing their models on.

The reviewer is right, Figure 1D shows the weighted matrix as the average number of streamlines. We have clarified this in the revision. We have updated both the figure caption and the figure title/colorbar label to reflect this more accurately.

Figure 1. Neonatal connectivity across PMA and GA at birth. (A) Tractograms highlighting the reduced connectivity between preterm and term infants who were both scanned at 42 weeks PMA. Below the tractograms, a graphic depicts the difference between PMA (orange) and GA at birth (blue) for these infants. The red bars indicate the time infants were exposed to the *extra-uterine* environment before being scanned. (B) Representative tractograms of four neonates born and scanned at different PMA and GA at birth. (C) Connectome density significantly increased across PMA and GA at birth (represented as predicted density from generalized additive model). The grey dotted line indicates the cut-off for term birth (37 weeks GA or later is term-born). (D) Average connectivity weighted matrix (average number of streamlines) and binarized matrix (consistency matrix), where the value indicates that proportion of participants that have that edge. *** indicates $p < 0.001$, ** indicates $p < 0.01$, * indicates $p < 0.05$.

We agree that there are important limitations when using binary networks and streamline counts. Relatedly, in other work, we are exploring a range of different input variables to the connectivity calculation. Using a Connectom scanner based in Cardiff (CUBRIC) we're able to calculate multiple microstructural properties of tracts,

a notable one of which is intra-axonal signal fraction (IASF). IASF is the closest proxy we are aware of for the (%) diffusion signal arising from neurites and axons, and hence, it quantifies the volume fraction occupied by axons. As you will see below, streamline counts correlate strongly with this microstructural measure (and the various derivatives of it) – across individuals the mean within-participant correlation between this metric and the number of streamlines being $r = 0.87$. To us, this suggests that while streamline counts have their clear and known limitations, these do not inherently make it an unreliable measure.

The within-participant correlation between IASF and streamline counts across 80 individuals scanned on the Connectom scanner at Cardiff University

We agree with the review paper’s main argument that weighted connectomes are more accurate representation of network topology than binarized connectomes. To use GNMs, however, we must binarize the networks. As the review paper states, generalizing binary and weighted connectomes should be done with caution given the differences between the two methods. This reason why we explore the binary networks only, and so extensively, is because these are the networks that we ultimately use for simulations. When constructing or pipeline, however, we did aim to mitigate the effects of this limitation. For example, we chose deterministic fiber tracking method which, as the review states, is known to reduce the rate of false positives. Crucially, we have also tested various thresholding approaches and explored how topology changes across different conditions (Supplementary Fig. 2) which most notably shows that our propensity-matched analysis (as outlined above), are entirely robust to thresholding choices.

In addition to our discussion about the limitation of using binary GNMs, we have added a point about using binarized networks in general and cited the reference provided above:

Discussion lines 497-500: “First, we are simulating binarized networks without the capability to increase or decrease connection strength or prune connections. This not only limits the biological accuracy of simulations but also requires the observed networks to be binary, which is considered less reliable for representing topology than weighted networks networks⁵⁴.”

4) L144-145 Clarify how networks are generated for analysis (1), how are

networks thresholded to generate variable density? In approach (2), if original (variable) density is lower than 10% for many subjects (as shown in Fig 1C), how can be all subjects forced to have a density of 10%? The authors of course clarify this issue in Methods section lines 571-584, but perhaps it is worth introducing here the absolute thresholding using streamline count of 325 for approach (1), which is not applied to approach (2), so it is easier for the reader to interpret.

We recognize how this is not clear without diving into the methods section and have revised to include more details about how these analyses were conducted.

Results lines 141-45: “To disentangle topological structure from network density, we conducted two analyses: (1) a variable density analysis for which an absolute threshold was applied (325 streamlines) that yielded an average density of 10% across the sample, and (2) a density-controlled analysis for which each network thresholded individually so that every network in the sample was *exactly* 10% density.”

5) L150-160. I appreciate the clarification regarding the methods used to calculate tractographies, and the addition of further analyses characterising different thresholding approaches to generate different levels of sparsity. However, I still think the results presented here and in Fig1C are a bit difficult to interpret (should clarify what thresholding approach is used to define network density). Is it possible that the density presented here (variable density according to approach 1, L144?) is related to small ROIs registered to babies with PMA<32 weeks? Have the authors QCd the template/atlas registrations to PMA<32 subjects?

We agree that the very early scans (PMA<32) are challenging to interpret. We included translation/rotation quality control metrics from the preprocessing pipeline in all our analyses (lines 715-16). We believe it is possible that lower densities are related to smaller ROIs in younger infants, however, we have not received quality-controlled metrics for template/atlas registration from DSI Studio from our fiber tracking. However, we think the best way of testing for certain whether our preterm versus term effects are genuine, or an artefact of a tricky registration process, is the further analyses that remove these early scans. The GAM analysis that just includes scans conducted at term-equivalent, included in the previous revision, and the propensity matched analysis included in this revision (Supplementary Fig. 3), both show that the effects of preterm birth are present even when just looking at scans conducted at post-term ages. These differences cannot be due to potential poor tracking of early scans. Since, however, our main analysis includes these early scans we have added additional reference to the caution which should be taken when interpreting these early scans:

Discussion lines 535-37: “Furthermore, caution must be taken when interpreting extremely early PMA scans and the effects of preterm birth. Firstly, early PMA fiber tracking is challenging and thus likely includes more errors than tracking of infants closer to term age.”

6) I would suggest to expand the discussion of limitations in relation to the use

of Euclidean distance for GNMs and the fact that this approach won't take into account, for instance, that the distance between regions of different hemispheres (or lobes) that are next to each other is much larger than the Euclidean distance between their centroids.

We appreciate the focus on the limitation of Euclidean distances for GNMs as we believe it to be one of the most important limitations of this work. While constrained by word count, we have elaborated on this in our discussion:

Discussion lines 508-14: “Thirdly, GNMs use Euclidean distance between regions as the ‘cost’ metric. A limitation of this metric, as compared to fiber lengths, is that it does not consider distances between regions in different hemispheres being larger than those within the same hemisphere. For GNMs, however, fiber lengths cannot be used as the ‘cost’ metric because they don't exist between all regions, and thus must be interpolated. It is of note, however, that fiber length and Euclidean distance are correlated – as strongly as $r = 0.773^{55}$.”

Remarks on code availability:

The code is a usable resource for the community. However, I don't see why the authors include 10 example datasets, but do not include the whole data analysed in their study, so the community can reproduce their results.

We used example data because the data are not ours to release. We are in the process of checking how much we are allowed to make publicly available outside of the portal via which anyone can access these publicly available data. If we receive permission to make the data freely available, we will immediately do so.

Response to Reviewer #2

The authors have addressed my comments and made appropriate revisions to the manuscript

Response to Reviewer #3

I thank the authors, who have addressed some of my concerns. However, there remain many that are still unaddressed.

Thank you for taking the time to review our manuscript and for the thoughtful and constructive comments. There is a lot we agree on. For instance, we agree that the current generation of GNMs do not capture edge-level information. But where we disagree is that this means these models are not informative. On the contrary we think they add important additional information beyond our novel analysis of the observed networks. That is why we wish to retain them.

In the response we have worked through each point and explained where we agree or disagree and our reasoning. We very much appreciate your perspective even

where we respectfully disagree. Our aim in the revision is to make sure that we are transparent as possible about our design choices and their impact, data quality, the value (and limitations) of the models, and to make sure the reader has all the information they need to make their own evaluation.

1. For instance, I initially made a comment that “edge-level matches [between the observed and simulated connectomes]” are not evident. The authors rightly noted that this isn’t their claim – their claim is that they match topological properties and distributions.

This is fair, but the authors are missing my point (or perhaps I didn’t explain it clearly enough). High-level network measures, or even distributions of local measures, do not uniquely specify a network. Most combinations of metrics can be satisfied by a plurality of networks. I don’t believe that this is a contentious statement.

We agree, a range of possible networks can generate some combination of local/global network measures together. But crucially, there are clear limits to this – the vast majority of networks *do not* satisfy the model distributions. In the manuscript we compare 13 different generative models, each with 90,000 simulations. The homophily models provide a more realistic simulation than all other models (Supplementary Fig. 8) and provide a highly accurate approximation of the statistical network properties. Moreover, the tuning of that particular model variant provides radically different networks, only a tiny minority of which can capture the higher-order network properties. The same is true of the topological fingerprint – the local arrangement of network properties across nodes – not all models can equally satisfy the spatial patterning of the network properties.

We do agree that these models are not designed to reproduce the exact location of edges, but we disagree that high-level network properties, or their local distributions, can be easily captured by a plurality of networks. Exploring and understanding a model that can do this, relative to those that cannot, is an informative addition to our novel analysis of the observed networks. To make this clear to the reader and therefore we have added this section to the discussion:

Discussion lines 468-472: “Despite not being aimed to provide edge-level fits, these homophily GNMs with narrow parameters uniquely recapitulate realistic neonatal network properties. These properties are not captured by a plurality of networks and therefore the exploration of how homophily GNMs generate networks could provide new theories into economic constraints of networks.
Why homophily?”

However, (structural) brain networks are highly specific. They have a fixed set of attributes, but they arrive at these attributes through a highly conserved architecture. The variability across brains in presence/absence/weights of links is remarkably low considering the space of all networks with similar high-level properties.

I am quite confident that the synthetic networks generated by the models, though they match high-level/distributional properties of the empirical

neonatal connectomes, arrive at those statistics with a wildly different configuration of edges. That is, the modeled networks only resemble the empirical networks at a coarse, summary level. Though I'm sure the discrepancy isn't *quite* this bad, this is roughly equivalent to saying a unimodal Gaussian distribution with mean of 0 has the same mean as a bimodal distribution whose means are equally spaced on both sides of 0.

So my claim is that without matching at an edge level, the models are simply not that informative. The authors referenced Figure 4F. This figure does not inspire confidence that the models are capturing local features. There are clear mismatches (the "hubs" aren't even in the correct locations).

We appreciate this critique of the level of interpretability and use of generative network modeling methods. Again, we agree with much of your perspective – the networks do not replicate edge-level matches and the networks likely replicate high-level properties through somewhat different edge locations. Where we disagree is that this makes these models uninformative. As the other reviewers have stated, these models allow us to explore something that cannot be seen within the graph theory analyses of observed data alone. Binarized GNMs are *not intended* to replicate networks at the edge-level, indeed binarized networks themselves do not capture the true complexity of real networks. Instead, these models are designed to test the underlying trade-offs needed in order to reproduce networks with high-level properties of the observed networks, and the local distributions of those properties. In essence, what economic conditions are required to yield networks with similar properties to those observed? As noted above, establishing a model that can use this trade-off to approximate those properties is not trivial.

We and others in the field are currently working on a new generation of GNMs designed to capture other properties of networks, including the weights of edges (Akarca et al., 2023). The current crop of GNMs, as applied here, were simply not designed to replicate the location of edges. But we remain convinced that these models provide valuable insight into the underlying constraints on network formation, beyond the novel analyses of the observed networks.

We have added to the introduction to clarify for the reader what these models are designed to do:

Introduction lines 76-79: “The purpose of these models is not to recapitulate edge locations, instead binary GNMs aim to replicate topological patterns of real networks and thus allow for exploration of what conditions are required to yield these properties.”

2. I also raised concerns over the apparent doubling of network density. The authors countered by referring to the Ball et al 2014 paper (density was not significantly different between pre-/post-term cohorts, Song et al 2017 (the authors don't report density, but show a large increase in strength – however, without the data it's not easy to directly determine whether this change was driven by density or edge weights), and van den Heuvel et al (2015), which focused mostly on group-averaged matrices but also doesn't report differences in density. In that paper, however, clustering increases and path

length decreases from 30-40w, consistent with an increase in density. However, the exact change is not known given what was reported in the paper. So, as far as I can tell and unless there are other references the authors can share, it is not clear that the doubling of density is unrelated to data quality or other issues that might obscure true connections.

We appreciate the reviewer's summary of past literature. In fact, we have not found a past paper that explores density differences in the way as we have. The papers we referenced were aimed at providing a summary of past work examining topology, to provide context as to why our density results are plausible. We apologize that this was unclear in our previous response. Together, increasing rich-club connections from 28-34 gestational weeks (Ball et al., 2014), increasing strength from 20-40 PMW (Song et al., 2017) and other topological results (van den Heuvel et al., 2015), all suggest that increasing density during this period of development is very reasonable. Generally, the concept that density is increasing drastically across this period is accepted by neonatal researchers (Bethlehem et al., 2022)

However, we do nonetheless agree with the central point the reviewer is making. There is a perennial concern about the quality of those very early PMA (low density) scans. That is why in the previous revision we included an additional term-equivalent age analysis. We wanted to exclude those very early low-density scans and test whether we replicate our main findings, which we did (Supplementary Fig. 3). To go further, the latest revision includes a propensity-matched analysis. We created two groups (both $n = 120$) matched on PMA and sex, one group with prematurely born infants and the other with term born infants. In other words, we have two groups, both scanned at term, scanned at an *identical* post-menstrual age, but one born preterm and the other born at term. Again, this replicates our original main findings.

In short, we agree that it can be very hard to know whether the sparse connectomes from those very early PMA scans reflect issues of data quality (despite the QC steps) or a genuine sparsity of connections. However, we think that sparsity is plausible given other explorations of topology (Ball et al., 2014; Song et al., 2017; van den Heuvel et al., 2015). Moreover, this cannot explain our key findings, which are present when those early scans are excluded and even when we match participants for age at scan.

Please find below the manuscript changes for the propensity-matched analysis:

Results lines 175-78: "In addition, preterm infants had significantly more modular networks compared to term infants ($t(216.96) = 12.16, p < 2.200 \times 10^{-16}$) in the propensity-matched analysis where the two groups are *identically* matched on sex and PMA (Supplementary Fig. 3B; see "Methods"; "Participants")."

Results lines 193-96: "This result was replicated in the term-equivalent age analysis (Supplementary Fig. 3C; $F_{global\ efficiency, GA\ at\ birth} = 2.71$, estimated $df = 3.60, p = 0.029$) and is consistent with the propensity-matched analysis in which preterm infants had significantly lower global efficiency compared to term infants ($t(203.35) = -7.24, p = 9.193 \times 10^{-12}$)."

Results lines 382-84: “These results are confirmed through density-controlled, term-equivalent age, and propensity-matched analyses. However, in the propensity-matched analysis, γ is smaller in the preterm group, though the difference is not significant (Supplementary Fig. 10).”

Methods lines 556-61: “To provide further confidence in term-equivalent age interpretations, we have also included a propensity-matched analysis with additional preterm infants scanned at term-equivalent age ($n = 80$). With this data, we create two groups (each $n = 120$) matched in sex and PMA – one group with infants born preterm and the other with infants born at term. There is no significant difference in head circumference between these groups ($p = 0.282$).”

Methods lines 603-05: “This thresholding technique was repeated for the propensity-matched analysis ($n = 240$) with a 390 binarization threshold to yield an average network density of 10%.”

Methods lines 613-18: “The propensity-matched sample was included in this exploration of multiple densities and showed that global efficiency and modularity differ significantly between groups across 5-24% average density networks (Supplementary Fig. 2E). Together these additional analyses demonstrated that while more dense networks would yield overall more efficient, less modular topology, that this does not change significant effects between term and preterm groups.”

Methods lines 722-24: “For the propensity-matched analysis, Welch two sample t-Tests were performed in R to examine group differences in modulatory, global efficiency, and GNM parameters.”

Supplementary Figure 3. Term-equivalent age topological analysis. (A) Density significantly increases across GA at birth. (B) Modularity significantly decreases across GA at birth and propensity-matched analysis shows that preterm infants have significantly higher modularity compared to term infants. (C) Global efficiency significantly increases across GA at birth and propensity-matched shows preterm infants have significantly lower global efficiency compared to term infants. (D) There is no significant change in the number of rich club connections, however (E) the average length of rich club connections significantly increases across GA at birth. The shaded area indicates 95% confidence intervals. The grey dotted line indicates the cut-off for term birth (37 weeks GA or later is term-born). *** indicates $p < 0.001$, ** indicates $p < 0.01$, * indicates $p < 0.05$.

I also found the authors response troubling:

“given our stringent thresholding it is unlikely that the increase in density is representing brand new tracts in every case, but more likely that these tracts have become prominent enough to pass the threshold”

This suggests that whether a tract is included or not is more of a statistical question than an empirical one.

All thresholding methods have pros and cons, and in our revision, we aim to be as transparent as possible regarding the benefits and limitations of our choices. There

is important discourse around the use of binary networks (please see our response to Reviewer 1, comment 3). Our perspective is that using strict thresholding – while losing some potential connections in the process – affords much higher confidence in the presence of connections that survive thresholding. With the rate of false positives in fiber tracking, our aim was to be strict – assuming that some excluded connections may be false – at the cost of excluding some ‘real’ connections that are weak. We appreciate the reviewer’s focus on the limitation of this choice; however, we believe this is still an empirical method which is commonly applied (Bullmore & Sporns, 2009).

To highlight the impact of our thresholding methods we have added to our methods:

Methods lines 596-598: “As we are utilizing binary networks, we chose a strict thresholding approach to allow for high levels of confidence in the surviving connections in addition to conducting many sensitivity analyses to explore the effects of this thresholding.”

3. I also raised concerns about density driving some of the observed differences in topology. The authors responded by thresholding the matrices at 10% and carrying out their analyses on the sparsified networks. I have one follow-up question and one concern. The follow-up question is this: why 10%? The authors need to show that their effects hold over a reasonable range of thresholds. There is nothing special about the 10% value.

We completely agree, there is nothing particularly special about 10%. In line with your suggestion, to show the effects hold over a reasonable range of thresholds, we have included additional topological analyses with progressive density from 8-24% showing how topology changes relative to density, as well as 5%-24% density networks in the propensity-matched analysis highlighting that group-effects hold across all levels of thresholding tested (Supplementary Fig. 2).

Ten percent was used in the main manuscript because this is the benchmark used by past GNM work, allowing the results to be more readily compared to prior literature (Akarca et al., 2021; Betzel et al., 2016; Carozza et al., 2022). Previous studies using GNMs with denser networks have modelled only one hemisphere, due to the computational power needed to run the simulations scaling super-linearly with connection number (Arnatkevičiūtė et al., 2021; Oldham et al., 2022; Vértes et al., 2012). Thus, the choice was informed by past GNM work that demonstrated 10% is feasible to model both hemispheres. In our multiple density analysis, we show that increasing density makes networks less modular and more efficient, however, the significant topological differences between term and preterm groups remains across 5-24% density (Supplementary Fig. 2). Thus, choosing 10% for the main manuscript does not change our main results.

This analysis also raises a concern. The authors note that after controlling for density, some of effects go away or even reverse their sign (increases are now decreases). Isn't this a problem? How can both be true? There are two views: the thresholding procedure removes noisy connections and controls for the likelihood that reported effects are purely the consequence of density. So maybe the thresholded results are closer to the truth. On the other hand, as

the authors note, thresholding can be merciless and throws away meaningful individual variation. In which case, maybe the original (unthresholded) results were closer to reality. But it can't be both. This inconsistency needs to be addressed by the authors.

Reverse signs, or lack of effects, in a fully density-controlled analysis is not a problem – indeed it should be expected. An analysis in which density is strictly controlled is asking a fundamentally different question, relative to the variable density analysis. Allowing for variable density across individuals, while in some senses being a truer picture of the underlying networks, is inherently biased because networks with more connections will inevitably be more integrated. Put simply, it is hard to disentangle *topological* and *topographical* differences without controlling density. Thus, a density-controlled analysis asks a different question – regardless of the *number* of connections, are these networks *organized* differently? When we control for density, we enable direct topological comparison without bias. Taken together, these analyses provide extremely interesting context to our topological results. For example, it appears that increased modularity in preterm infants is not solely due to few connections, however, less efficient networks in preterm infants is *uniquely explained* by the number of connections (i.e., this effect goes away when we control the number of connections). That is why it is useful to have both.

Furthermore, in the latest revision we have included another density analysis with the propensity-matched sample. Here, we demonstrate that significant group effects remain unchanged with varying density (Supplementary Fig. 2E). We have added a section in the Discussion to point out to the reader the importance of considering a variety of thresholding methods:

Discussion lines 528-33: “In addition, we used thresholded networks which has important limitations to consider. By thresholding, we remove potential false positive connections from tracking errors⁵⁹ but also likely remove real, weak connections. While we have provided variable density, controlled density (Fig. 2), and multiple density analyses (Supplementary Fig. 2) to demonstrate the effects of these choices, it is important to note that there is no ‘right’ thresholding method, and each has important benefits and limitations.”

Supplementary Figure 2. Exploration of multiple density networks. Across consensus networks, **(A)** modularity decreases, **(B)** global efficiency increases, and **(C)** characteristic path length decreases with increasingly dense connectomes. **(D)** Comparisons of consecutive consensus networks, indicating the distribution of new connections across 7 lobes (indicated as a % change from the sparser to dense network). **(E)** In the propensity-matched analysis, preterm infants at term-equivalent age have significantly less efficient, more modular networks in compared to term infants across 5-24% average density networks (global efficiency: 5% $t(189.15) = -10.76, p < 2.200 \times 10^{-16}$; 10% $t(190.91) = -8.28, p = 2.065 \times 10^{-14}$; 15% $t(237.49) = -9.66, p < 2.200 \times 10^{-16}$; 20% $t(220.23) = -5.76, p = 2.748 \times 10^{-8}$; 24% $t(237.75) = -4.01, p = 8.134 \times 10^{-5}$; modularity: 5% $t(221.97) = 11.60, p < 2.200 \times 10^{-16}$; 10% $t(224.80) = 14.31, p < 2.200 \times 10^{-16}$; 15% $t(193.12) = 12.06, p < 2.200 \times 10^{-16}$; 20% $t(235.52) = 9.35, p < 2.200 \times 10^{-16}$; 24% $t(197.76) = 8.65, p = 1.805 \times 10^{-15}$). **(F)** Topological fingerprints demonstrating the correlation between local organizational measures for the 8%, 18% and 24% density networks. **(G)** Connectome matrices representing the 8%, 18% and 24% networks. Here, the lower triangle is the matrix and for 18% and 24% networks the upper triangle is only comprised of connections that are present in that network that weren't present in the previous network (8-18% and 18-24% comparisons).

4. I also asked about rich clubs, noting that brain/cranial volume surely changes (increases) during this period, so that any two points on the brain surface likely grow further apart. If the rich-club stays even remotely consistent (comprised of the same set of edges) then the expectation is that the lengths of those connections will also increase. The authors responded by saying “the distance between the parcels stays the same” (in essence). This can't be right unless all brains were matched to some shared reference brain. If this is the case, it seems problematic in the context of generative modeling. A key component of the two-term models is the Euclidean distance matrix. If

distances change, this will impact the model parameters. For example, larger/smaller brains (greater/reduced interregional distances) might need different distance penalties—the γ parameter—to match properties of the empirical network. From what the authors report, it's not clear that this is the case.

We agree – this is a great point. There has yet to be a version of this particular modeling technique that can truly accommodate changes in brain size per se. This is because the approach considers the *relative* distance between regions, rather than their absolute distance. If this changes, 'costly' connections could become 'cheap' or vice versa which would drastically change the interpretation of the model. Previously, Oldham et al. (2022) utilized HCP data to explore distance changes across gestational age. By using the modal surface matching (MSM) algorithm, Oldham et al. (2022) mapped node locations across GA weeks. This work demonstrates that despite overall volume changes, the *relative* distances between nodes does not drastically change (Oldham et al., 2022). Since GNMs 'cost' metric relies on *relative* distances, different gestational ages will have minimal impact.

Our propensity-matched analysis also addresses this issue. In this analysis we have an equal group of term-born, and a group of prematurely born infants, who have been matched for sex and PMA (i.e. age at scan). Due to the matched PMA, the brain/cranial volume are at the same developmental timepoint. In our original analysis, head circumference significantly differs between preterm and term born infants ($p < 2.200 \times 10^{-16}$; term mean = 35.01, preterm mean = 30.39), however, in the propensity-matched analysis there is no significant difference in the head circumference of preterm and term groups ($p = 0.282$). Thus, in the propensity-matched analysis the sizes differences are eliminated. Here, we show significant key topological differences between term and preterm infants across multiple density networks (Supplementary Fig. 2E) and a significant difference in the η parameter (Supplementary Fig. 10D). Given these analyses, our main result *cannot* be attributed problems concerning changes in brain volume.

To make it clear to the reader that this propensity-matched analysis results in matched head sizes, we have included the head circumference result:

Methods lines 556-61: “To provide further confidence in term-equivalent age interpretations, we have also included a propensity-matched analysis with additional preterm infants scanned at term-equivalent age ($n = 80$). With this data, we create two groups (each $n = 120$) matched in sex and PMA – one group with infants born preterm and the other with infants born at term. There is no significant difference in head circumference between these groups ($p = 0.282$).”

As a side note, I examined the github repository. I have some concerns about the data itself. One of the hallmarks of structural connectomes is their distance dependence and preference for short-range connections. One way this is evident is if one were to compare the mean distance between connected nodes against the mean distance of unconnected nodes. Because connectomes make more short-range connections, the mean distance for connected nodes is always smaller than unconnected. However, when I run

the script "A_run_initial_generative_models.m" to obtain the variables "edistance" (Euclidean distance between pairs of nodes) and "example_consensus_network" and then make a boxplot of distances for connected and unconnected nodes, the distributions are not different. Maybe this is something that is true for neonatal connectomes (lack of distance dependence) or this sample is not representative in some way (or I misunderstood the README file). But this is odd to me.

We apologize for the confusion. The example networks provided are completely randomized and are simply there so the code will run. They do not carry properties that represent neonatal data – it is simply a placeholder. Sorry – we should have made that clear! As this is not our data, we have requested permission from dHCP to publish the data from this project. If allowed, we will make our data immediately public so anyone can replicate our findings.

We have updated the README file to clarify this: “This folder contains random **placeholder** data (which hold no properties representative of real networks) for all variables necessary to run the scripts.”

In summary, I don't think this is a bad paper—I quite like many of the results. But under scrutiny, there are too many concerns related to the underlying data, processing decisions, and interpretations of the results for me to endorse its publication.

We sincerely appreciate the time and effort you have dedicated to reviewing our manuscript. It has been invaluable in improving the manuscript. While we may not see eye-to-eye on everything, we believe the latest revision showcases that our findings are robust to a whole series of processing decisions (e.g., density changes), robust to the removal of early scans or even matching participants for PMA at scan and contains everything that readers need to accurately understand the work and make their own evaluation. Finally, the connectomes in the Github repository are currently dummy placeholders but the actual data will be made public upon the permissions being granted to us.

Response to Reviewer # 1

Reviewer #1 (Remarks to the Author):

The authors have now addressed most of my previous concerns, or at least, edited the manuscript so the limitations of their work are clear.

Reviewer #1 (Remarks on code availability):

The authors do not provide the derivative data (brain networks / adjacency matrices) used in this manuscript, so it would be very difficult to reproduce the results presented in their work.

In their response to my previous comment on data availability they mention the data is "is not theirs to release". I do not understand why would this be the case, i.e., why wouldn't they be able to share the adjacency matrices (brain networks) they calculated, and who else (if not the authors) has the right to grant access to these data.

We apologize for the delay with publishing the data. We were hoping to do our due-diligence and get direct approval from dHCP before making any data public. Though we have not received a response from dHCP, we have published all the derived data used in the manuscript.

The data is now publicly available: <https://osf.io/ng43c/>

Response to Reviewer # 3

Reviewer #3 (Remarks to the Author):

I appreciate the candid and transparent response from the authors. They are correct -- we do not see eye to eye on every matter. They have much more faith in the generative models (their ability to reveal something novel and meaningful that is not captured using other methods, e.g. graph theory) than I do.

That said, the transparency is much appreciated, and offers some balance to their submission.

I don't wish to use the review process as a negotiating table, but here's what I would suggest in the next round of revisions. At this point the authors have 1) addressed my technical concerns; I have no serious remaining technical issues; 2) they have added *some* balance to their article.

I strongly suggest being more explicit and focused in how they include this

balance. Presently, their responses to my philosophical concerns related to generative models appear throughout the manuscript; they are scattered and attached to specific results/remarks.

I strongly believe that it would be beneficial to readers, and especially those unfamiliar with generative models who might not be aware of their limitations, to include a sub-section in the discussion (with a title) that includes a similarly transparent and candid discussion of the conversation that's transpired in the review process. Should we believe a 2-fold increase in network density -- are we *really* confident that this reflects a doubling of myelinated white matter? How should we interpret the results of a fit generative model? Certainly not mechanistically as a model of growth/development. If not this, then is it fair to think of the models as compressive? Perhaps, but only in the sense that the models generate networks with similar features at the level of cumulative distributions -- despite claims made here and by others using generative models, they are quite poor at faithfully capturing the precise configuration of edges white-matter networks and the topography of local network statistics (yes, I'm aware local statistics in observed and synthetic networks are often correlated, but the correlations are not strong).

Should the authors agree to this (and I would love to read the newly written paragraph, should another round of review be granted) I think it would put the article within the "recommend for publication" circle, in my opinion.

Again, I don't wish to use peer review as a negotiating table, but this is the clearest way I know to communicate the gap between the manuscript in its current form and what I still believe needs to be addressed to make it acceptable for publication.

We are very happy to agree to this recommendation. We have greatly appreciated the discourse over these revisions and agree that consolidating the philosophical concerns and interpretability of the generative models into a single paragraph would greatly help readers. We have broken the limitations section of our discussion into two parts to provide clear titles and included key aspects of our discourse for the reader. In a couple of places we have also incorporated existing limitations that were already included in the Discussion, but which are now consolidated into a single place:

Lines 496-531: **“Generative network modeling interpretability and limitations.** When interpreting the GNM parameters, it is important to consider what these models do and *do not* tell us. A well-fit generative model tells us about the global trade-off between the influence of connection cost (i.e. distance) and connection value (i.e. homophily) needed to recreate the high-level distributional properties of the observed networks. If a GNM does a good job of recapitulating these distributional properties then not only should the global distributions of simulated network properties be very close to those of

the observed networks, but so too should the local spatial arrangements of those properties (i.e. the ‘topological fingerprint’) – a node with high betweenness centrality should have a low clustering coefficient, and so on. There is something specific about the characteristics of GNMs that can do this. Replace the homophily term with a different property (e.g. nodal strength), or change the parameters outside relatively narrow bounds, and the GNM’s ability to replicate these properties of networks is much reduced. We included this type of modeling alongside our graph theory analyses because we wanted to test with the *economic constraints* needed to recapitulate these network properties change across the preterm to term timeframe. It is in this sense that these models are compressive: with just two calibrated global parameters, networks can be simulated that capture many of the complex distributional hallmarks of observed networks. These contextual details about the conditions necessary to best replicate network properties are only ascertainable through generative analyses. But it is important to consider what these models *do not* tell us.

Firstly, these models do poor job of placing edges in the right locations. The models are tuned to achieve distributions of high-level network properties, but the location of the connections themselves will vary and are often poorly matched to those in the observed networks. Likewise, these models produce *some* spatial embedding – high degree nodes in the simulations are also significantly more likely to be high degree nodes in observed networks – but this embedding is far from perfect. There are many likely reasons for this, including that we are working with very sparse binarized networks. Needless to say, the current application of GNMs offer insights into the trade-offs needed to yield realistic network features. They are not designed for, and should not be used for, edge-level exploration of network formation.

The second obvious way in which these models fall short is that they are not biologically mechanistic. These models are not attempting to simulate a biological process, so whilst they can predict the *types* of connections that form under different conditions, they do not explain *why* a given connection has formed. In other words, they are not biologically compressive. There are current methodological barriers that this type of modeling that must be addressed before the field can move in this direction. We are currently simulating binarized networks without the capability to increase or decrease connection strength or prune connections. This not only limits the biological accuracy of simulations but also requires the observed networks to be binary, which is considered less reliable for representing topology than weighted networks⁵⁴. Weighted GNMs that update connection strength over time, in addition to connection formation, have recently been established although are not currently placed to simulate large samples⁵⁵. Another methodological limitation is that during the pre- and neonatal phases, the actual volume of the brain changes rapidly; however, we are constrained to calculating the distance between regions with a single atlas. Recent work suggests that differences in cortical geometry may influence connectome topology⁴³. Thus, the addition of a biological informed measure into the GNMs likely would improve the accuracy of simulations. While we see substantial replication of topological metrics, GNMs are not identical to observed connectomes and thus cannot be interpreted as exact replications. A final obvious methodological limitation is the way in which connection ‘cost’ is

calculated. GNMs use Euclidean distance between regions as the 'cost' metric. A limitation of this metric, as compared to fiber lengths, is that it does not consider distances between regions in different hemispheres being larger than those within the same hemisphere. For GNMs, however, fiber lengths cannot be used as the 'cost' metric because they don't exist between all regions, and thus must be interpolated. It is of note, however, that fiber length and Euclidean distance are correlated – as strongly as $r = 0.77$."

In a separate section of the Discussion we have also highlighted the point the reviewer highlights about changes in network density. We did this in a separate section because this is not specific to the GNMs, but to the connectomes themselves.

Lines 549-568: **“Project design and neurobiological limitations.** Macroscale binarized connectomes are compressed representations of neurobiology. A connection being present or absent between two regions could reflect the strength or developmental stage of that connection – not the actual existence of such a connection. Thus, the rapid increase in network density across postmenstrual ages should be interpreted within binarization limitations, specifically in consideration of increasing density. Premature birth is associated not just with reduced connectivity, but also an impairment to the migration and maturation of neural cells⁵⁶. The preterm period overlaps with the pre-myelination stage, during which there is rapid movement and maturation of glial cells⁵⁷. Disruption during initial stages of glial cell development become more apparent in infancy in the myelination process⁵⁸. However, topological analysis of macroscale networks is likely influenced by these early cell-level difference, as it relates to decreasing anisotropy and associated metrics during the preterm period⁵⁷. Thus, our topological analysis of the macroscale neonatal network cannot be interpreted as solely connection-level differences and is likely, if not definitely, encompassing the disruption of pre-myelination development in infants born at early GA. In addition, we used thresholded networks which has important limitations to consider. By thresholding, we remove potential false positive connections from tracking errors⁵⁹ but also likely remove real, weak connections. While we have provided variable density, controlled density (Fig. 2), and multiple density analyses (Supplementary Fig. 2) to demonstrate the effects of these choices, it is important to note that there is no 'right' thresholding method, and each has important benefits and limitations.”

Finally, again, thank you for the time you have invested in helping revise the manuscript, and for the even-handed approach you have taken.

Response to Reviewer # 3

Reviewer #3 (Remarks to the Author):

The authors have addressed all of my concerns. Glad to endorse this for publication.

We really appreciate your thoughtful and thorough review!

Reviewer #3 (Remarks on code availability):

It would be useful to share as much of the data as possible; but the code itself seems reasonable.

We agree that sharing data is important! We have made all our data public on OSF: <https://osf.io/ng43c/>. Here you can find observed networks (raw and thresholded), derived data (graph theory measures and GNM model fit data) and the simulated networks which can all be used to replicate our findings with our code. We have also provided Source Data for all our figures.